# Catalytic Role Of Noise And Necessity Of Inductive Biases In The Emergence Of Compositional Communication

**Łukasz Kuciński**\*
Polish Academy of Sciences
`lkucinski@impan.pl`

**Tomasz Korbak**
University of Sussex
`tomasz.korbak@gmail.com`

**Paweł Kołodziej**
Polish Academy of Sciences[†]
`p.kolodziej@gmail.com`

**Piotr Miłoś**
Polish Academy of Sciences,
deepsense.ai
`pmilos@impan.pl`

## Abstract

Communication is compositional if complex signals can be represented as a combination of simpler subparts. In this paper, we theoretically show that inductive biases on both the training framework and the data are needed to develop a compositional communication. Moreover, we prove that compositionality spontaneously arises in the signaling games, where agents communicate over a *noisy channel*. We experimentally confirm that a range of noise levels, which depends on the model and the data, indeed promotes compositionality. Finally, we provide a comprehensive study of this dependence and report results in terms of recently studied compositionality metrics: topographical similarity, conflict count, and context independence.

## 1 Introduction

In emergent communication studies, one often considers agents who can share information about a set of objects described by the common features. Such a situation is common in multi-agent systems with partial observation (Foerster et al. (2016), Lazaridou et al. (2016), Jaques et al. (2018), Raczaszek-Leonardi et al. (2018)) and it is the major theme in signaling games (Fudenberg et al. (1991), Lewis (1969), Skyrms (2010), Lazaridou et al. (2018)). In a signaling game, one agent (a sender) conveys information about an object to another agent (a receiver), which then has to infer the object's features. Typically, agents are rewarded if some of the features are correctly identified. During this process, the agents develop a communication protocol. A recent line of work has studied conditions under which compositionality emerges (Batali (1998); Kottur et al. (2017); Choi et al. (2018); Korbak et al. (2019); Li and Bowling (2019); Słowik et al. (2020b,a); Guo et al. (2020)).

Compositionality is a crucial feature of natural languages and it has been investigated extensively in cognitive science (see e.g. Chomsky (1957) Fodor and Pylyshyn (1988)). It is often measured using dedicated metrics such as topographic similarity (Brighton and Kirby (2006); Lazaridou et al. (2018); Kriegeskorte (2008); Bouchacourt and Baroni (2018)), context independence Bogin et al. (2018), conflict count Kuciński et al. (2020), or positional disentanglement (Chaabouni et al. (2020)). In signaling games it bears a strong resemblance to the concept of disentangled representations, see (Higgins et al. (2017), Kim and Mnih (2018), Locatello et al. (2019)). In machine learning context,

---

\*Corresponding author.
[†]Now at Google.

35th Conference on Neural Information Processing Systems (NeurIPS 2021).

compositionality is perceived as a generalization mechanism (Lake et al. (2016)) and has been used e.g. for goal composition (Jiang et al. (2019)) or knowledge transfer (Li and Bowling (2019)).

In this paper, we theoretically show that inductive biases on both the training framework and the data are needed for compositionality to emerge. A similar observation has been made by Kottur et al. (2017); however, our result is more fundamental and points out a common misconception that compositionality can be learned in a purely unsupervised way. Such a result can be perceived as a discrete analog of Locatello et al. (2019), applicable in the communication context.

We then prove that adding an inductive bias in the loss function coupled with communication over a noisy channel leads to the spontaneous emergence of compositionality. This shows the catalytic role of noise in this process. Intuitively, this can be attributed to the (partial) robustness of compositional language with respect to message corruption caused by a noisy channel.

We experimentally verify that a certain range of noise levels, dependent on the model and the data, promotes compositionality. We provide a wide range of experiments that illustrate the influence of different priors. For the inductive biases in the training framework, we look into the impact of the network architecture as well as implementation and temporal variation in noise. On the data side, we study the effect of scrambling visual input or its description. We also study the generalization properties of the proposed training framework.

## 2    Related work

The topic of communication is actively studied in multi-agent RL, see Hernandez-Leal et al. (2020, Table 2) for a recent survey. Compositionality is often investigated in the context of signaling games (Fudenberg et al. (1991), Lewis (1969), Skyrms (2010), Lazaridou et al. (2018)). Recent research has shown that strong inductive biases or grounding of communication protocols are necessary for the protocol to be compositional (see e.g. Kottur et al. (2017), Słowik et al. (2020b)). The inductive bias can be imposed into the architecture of the agents or the training procedure. For instance, Das et al. (2017) place pressure on agents, to use symbols consistently across varying contexts, by a frequent reset of the agent's memory. A model-based approach was proposed by Choi et al. (2018) and Bogin et al. (2018), who build upon the obverter algorithm (Oliphant and Batali (1997), Batali (1998)). Słowik et al. (2020a) explore games with hierarchical inputs and shows how agents implemented as graph convolutional networks obtain good generalization. Korbak et al. (2019) implemented the idea of template transfer (Barrett and Skyrms, 2017) by pre-training the agents on simpler subtasks before the target task. Kirby (2001) studied the iterative learning paradigm, where each generation of agents learns the language spoken by the previous generation before starting to communicate. In the machine learning literature, this idea was explored by Li and Bowling (2019), Cogswell et al. (2019) and Ren et al. (2020) with the generation transfer typically implemented as reinitializing the weights of agents' neural networks. Such an approach inevitably introduces noise into the learning process. This naturally leads to a question of whether the noise itself may be a sufficient mechanism of compositionality, which we will try to address in this paper. Guo et al. (2020) have shown that the choice of a game has a large impact on the properties of a communication protocol emerging in that game, foreshadowing what we call grounding. Kim and Oh (2021) study emergent language in the group setting with varying population sizes and connectivity. Liu et al. (2021) cast constructing structured neural architectures as a communication problem.

The noisy channel model of communication was famously introduced by Shannon (1948). The idea of noise as a driving force in the emergence of communication was first proposed by Nowak and Krakauer (1999), who showed that word-level compositionality is the optimal solution to the problem of communication in a noisy environment under a particular fitness function. Tucker et al. (2021) consider a noisy channel communication, where the agents learn to communicate via discrete tokens. Noise is also used in deep learning, e.g. as a regularizer (see e.g. dropout (Srivastava et al., 2014)) or a mechanism allowing backpropagation through a discrete latent (see e.g. Salakhutdinov and Hinton (2009), Kaiser and Bengio (2018)). Noise in the latter context was used in Foerster et al. (2016) in order to learn to communicate. The authors observed that it is essential for successful training.

## 3    Noisy channel method

We discuss the language and compositionality in Section 3.1. The impossibility result and the need for biases in emergent compositionality is the content of Section 3.2. The communication task considered

in this paper as well as the catalytic role of noise is described in Section 3.3. Theoretical results from this section hold in a somewhat idealized situation, but experiments in Section 5 are performed in a more realistic setup. The difference is described in Section 4.1.

## 3.1 Language and compositionality

Consider a set of objects described by some features, and let a set $\mathcal{F}$ contain a combination of these features' values. For example, the features could represent shape, say `squares` and `circles`, and color, say `red` and `green`, in which case $\mathcal{F} = \{$`red square`, `red circle`, `green square`, `green circle`$\}$. The features could be identified with partitions of $\mathcal{F}$. More formally, each feature can be defined via equivalence relation, with features values corresponding to equivalence classes of this relation. In our example, the color corresponds to the partition $\{$`red square`, `red circle`$\}$ ('red') and $\{$`green square`, `green circle`$\}$ ('green'), while for the shape corresponds to the partition $\{$`red square`, `green square`$\}$ ('square') and $\{$`red circle`, `green circle`$\}$ ('circle').

We assume that objects with feature space $\mathcal{F}$ can be defined in a space $\mathcal{X}$ and are generated by a two-stage process: first, the feature values are sampled from $f \in \mathcal{F}$, then an element of $\mathcal{X}$ is sampled according to a distribution conditioned on $f$.

In general, a language is defined as a mapping from objects to strings over some finite alphabet $\mathcal{A}$ (sometimes called messages), $\ell : \mathcal{X} \rightarrow \mathcal{A}^*$. In this paper, we will study a subset of languages $\ell$, where the range of the language has a fixed length, equal to the number of features. We say that $\ell$ is compositional with respect to a given feature if a change in $i$-th feature only impacts a corresponding $j$-th index of the message. Continuing the previous example, let $\mathcal{A} = \{$`a`, `b`$\}$ and consider a language $\ell$ mapping red squares, red circles, green squares, and green circles to `aa`, `ba`, `ab`, and `bb`, respectively. Then $\ell$ is compositional with respect to color and shape features since the change of color only impacts the first index in the message (and analogously for the shape), see Figure 1(a).

Consider a permutation $\pi \colon \mathcal{F} \rightarrow \mathcal{F}$. In our running example, suppose that $\pi$ is an identity, except that it swaps `red circle` with `green circle`, i.e. $\pi($`red circle`$) = $ `green circle` and $\pi($`green circle`$) = $ `red circle`. The resulting language $\ell_\pi$ would then map red squares, red circles, green squares, and green circles to `aa`, `bb`, `ab`, and `ba`, respectively. This language is not compositional with respect to color and shape features, since if we change a shape value in `red circle`, both symbols in the message will change (from `bb` to `aa`), see Figure 1(b). However, $\ell_\pi$ is compositional with respect to a different set of features (shape and 'different color-shape'), see Figure 1(c). Consequently, compositionality should be defined together with features, with respect to which it holds. In the next section, we show that this observation has significant implications for learning.

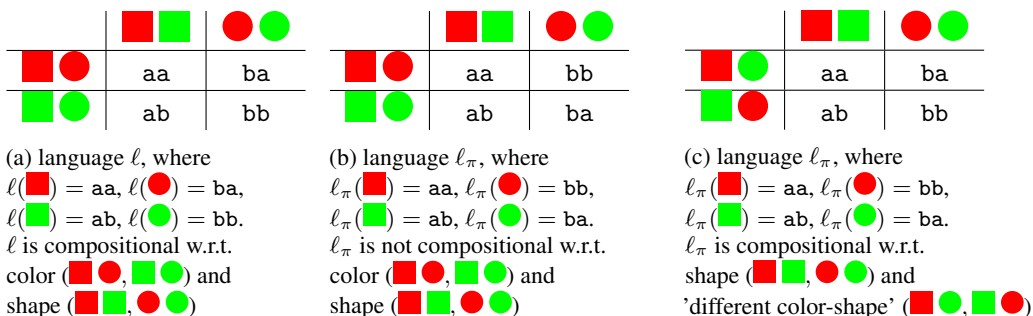

(a) language $\ell$, where
$\ell($■$) = $ `aa`, $\ell($●$) = $ `ba`,
$\ell($■$) = $ `ab`, $\ell($●$) = $ `bb`.
$\ell$ is compositional w.r.t.
color (■ ●, ■ ●) and
shape (■ ■, ● ●)

(b) language $\ell_\pi$, where
$\ell_\pi($■$) = $ `aa`, $\ell_\pi($●$) = $ `bb`,
$\ell_\pi($■$) = $ `ab`, $\ell_\pi($●$) = $ `ba`.
$\ell_\pi$ is not compositional w.r.t.
color (■ ●, ■ ●) and
shape (■ ■, ● ●)

(c) language $\ell_\pi$, where
$\ell_\pi($■$) = $ `aa`, $\ell_\pi($●$) = $ `bb`,
$\ell_\pi($■$) = $ `ab`, $\ell_\pi($●$) = $ `ba`.
$\ell_\pi$ is compositional w.r.t.
shape (■ ■, ● ●) and
'different color-shape' (■ ●, ■ ●)

Figure 1: Language, features, and compositionality.

## 3.2 Inductive biases and compositionality

This section investigates whether compositionality can spontaneously emerge in an unsupervised way. The answer to this question is negative since the underlying features cannot be inferred from the data.

**Theorem 1.** *For a uniform distribution, $\mu$ on $\mathcal{F}$ and a permutation $\pi \colon \mathcal{F} \rightarrow \mathcal{F}$, the distribution $\mu \circ \pi^{-1}$ is also uniform.*

While this result is elementary to prove (see Appendix F) it has deep implications for the emergence of compositionality in the learning process. Suppose that data represent a balanced set of all features

(i.e. features are sampled from $\mu$) and recall the two-stage generation process described in Section 3.1. If $f \sim \mu \circ \pi^{-1}$ has the same distribution as $f \sim \mu$, then the distribution of the observed data does not depend on $\pi$. Consequently, any emergent compositional language $\ell$ with respect to some features is not compositional with respect to other (permuted) features. We arrive at the following conclusion:

*Any learning process which hopes to achieve compositionality must involve some priors related to the data and learning framework.*

Theorem 1 can be viewed as a discrete version of Locatello et al. (2019, Theorem 1). There are multiple ways of grounding the learning process, including imposing inductive biases on the agents and designing loss functions to disentangle features. We study these, together with a new mechanism: injecting noise into the communication channel. In the next section, we show that this mechanism provably achieves compositionality.

### 3.3 Compositionality and communication over a noisy channel

Another major conceptual finding is that compositional communication spontaneously emerges when introducing a relatively simple mechanism – a noisy channel. This is proved in Theorem 2, provided that the loss function penalizes agents' mistakes, but also rewards for (partially) correct guesses.

Recall, that we consider a signaling game, where agents cooperate and develop a communication protocol (a language $\ell$) in order to maximize their joint reward. Here a sender observes a certain object with features $f \in \mathcal{F}$ and sends a message to the receiver to allow him to infer $f$. The agents are rewarded if some of the features are identified correctly, see equation 1.

The above setup is standard and now we augment it with a noisy channel. The noisy channel is located between the sender and the receiver and may scramble messages. A message $\mathbf{s}$ is transformed into a corrupted message $\mathbf{s}'$, by replacing each symbol, independently and with probability $\epsilon \in (0, 1)$, with a different, uniformly sampled, symbol.

For the sake of this section and Theorem 2 below, we will make the following series of assumptions. Let $\mathcal{F} = \mathcal{F}_1 \times \ldots \times \mathcal{F}_K$, that is the feature space is factorized into $K$ features. Furthermore, assume that each feature has the same number of values. We will assume that $\mathcal{X} = \mathcal{F}$ and define a language used by the sender as a mapping $\ell : \mathcal{F}_1 \times \ldots \times \mathcal{F}_K \to \mathcal{A}^K$, where $\mathcal{A}$ is an alphabet and $|\mathcal{A}| = |\mathcal{F}_i|$. We will further assume that the message $\mathbf{s} = \ell(f)$ is decoded as $\ell^{-1}(\mathbf{s})$. The corrupted message corresponding to $f \in \mathcal{F}$ is denoted by $\ell(f)'$ and the inferred features corresponding to this corrupted message are given by $f' = \ell^{-1}(\ell(f)') \in \mathcal{F}$. The setup for this section and for our experiments (Section 4 and Section 5) differ, see Section 4.1.

Recall that the definition of compositionality (with respect to given features) connects the change in feature values with the change in message symbols. It is thus reasonable to look for loss functions that are somehow factorized in terms of individual symbols. Consider the following loss function:

$$J(\ell, f) = \mathbb{E}[H(\rho(f', f))], \tag{1}$$

where $H$ is a non-negative, strictly increasing function and $\rho$ is the Hamming distance[3]. Intuitively, $\rho$ measures the number of changed features in the corrupted message and $H$ controls the degree by which we penalize this quantity.

**Theorem 2.** *Assume that $K \geq 2$, $\mathcal{F} = \mathcal{F}_1 \times \ldots \times \mathcal{F}_K$, $|\mathcal{A}| = |\mathcal{F}_i| \geq 2$, and $\mathcal{X} = \mathcal{F}$. Suppose additionally that $\epsilon < (|\mathcal{A}| - 1)/|\mathcal{A}|$. Then a language $\ell^*$ minimizes $J$ over all languages $\ell$ which are one-to-one mappings if and only if $\ell^*$ is compositional (with respect to features given by $\mathcal{F}_i$).*

Informally, Theorem 2 states that optimization of loss function $J$ promotes compositionality (assumption on the one-to-one property is technical). What makes this possible is the factorized nature of the losses and the introduction of a noisy channel. Interestingly, there exist other loss functions with similar properties. We postpone the analysis of this point and the proof of Theorem 2 to Appendix F.

It is instructive to discuss some of Theorem 2 assumptions. Assumption $\mathcal{F} = \mathcal{X}$ means that $\ell$ takes semantically meaningful symbolic input. This does not cover many interesting cases, where $\mathcal{X}$ has representation entangled in terms of features (e.g. an image). Furthermore, the assertion of Theorem 2 holds for a rather wide spectrum of $\epsilon$ values (up to $0.8$ for $|\mathcal{A}| = 5$, which is what we use in the main experiment in Section 5). This stands in contrast to our experimental results, where we

---

[3]The Hamming distance between two vectors $v, w \in \mathbb{R}^K$ is defined as $\rho(v, w) = \sum_{i=1}^{K} \mathbf{1}(v_i \neq w_i)$.

observe an interaction between different noise levels and compositionality, see Section 5. There is no contradiction here since in Section 5 we study more realistic setup that extends beyond relatively strict assumptions of Theorem 2, see Section 4.1.

# 4 Experiments setup

## 4.1 Differences between experimental and theoretical setups

The setup of our experiments is more realistic (and common in this area of research) and extends beyond the assumptions of Theorem 2. We assume that a dataset $\mathcal{X}$ contains images, consequently making features entangled in a visual representation (i.e. $\mathcal{F} \neq \mathcal{X}$). We also allow the alphabet size to differ from the number of feature values, $|\mathcal{A}| \neq |\mathcal{F}|$. Additionally, the receiver only sees the messages sent by the sender and has to learn to decode the messages. A further gap stems from the implementation details and the use of neural networks, which may converge to suboptimal solutions. We do, however, assume that the feature space $\mathcal{F}$ is a Cartesian product (see Section 3.3).

## 4.2 Training pipeline

In this section, we sketch the training pipeline and postpone the details to Appendix C. The dataset of images observed by the sender is denoted by $\mathcal{D}$. Each element of $\mathcal{D}$ has $K$ independent features $f_1, \ldots, f_K$ (here we consider $K = 2$). Both the sender and the receiver are modeled as neural networks (for details see Appendix B). The sender network takes an image from $\mathcal{D}$ as input and returns a distribution over the space of messages of length $L$ (here we assume $L = K = 2$). We assume that conditionally on the image, the symbols in the message are independent and take values in a finite alphabet $\mathcal{A}_s = \{1, \ldots, d_s\}$. This distribution is then distorted by the noisy channel, which is a function that maps probability vectors into $d_s$-dimensional logits. Unless otherwise stated, we assume that noise is be defined as a dense layer with a specific choice of (not learnable) weights matrix and a $\log$ activation function:

$$\texttt{noise}(x) = \log(Wx). \tag{2}$$

Here we assume that $W \in \mathbb{R}^{d_s \times d_s}$ is a fixed matrix, which takes a probability vector to a probability vector with positive entries. An example of such $W$ is a stochastic matrix. In this paper we use

$$W_{ij} = \begin{cases} 1 - \varepsilon, & i = j, \\ \frac{\varepsilon}{d_s - 1}, & i \neq j. \end{cases} \tag{3}$$

Alternative implementations of noise architecture are possible, see Appendix C. The noisy logits are then used to sample a message and pass it to the receiver. To make this operation differentiable we use Gumbel-Softmax (Jang et al. (2016)) with Straight-Through mode (Kaiser and Bengio (2018)).[4] Upon receiving the message, the receiver outputs a distribution over possible values of the features, using an alphabet $\mathcal{A}_r = \{1, \ldots, d_r\}$. Finally, the neural networks are trained using a linear combination of cross-entropy loss for the receiver, cross-entropy loss for the sender, and $L_2$-regularization. The loss term for the sender incentives the language to be a one-to-one mapping.

## 4.3 Datasets

We use two datasets: shapes3d (used in Burgess and Kim (2018) and included in the TensorFlow datasets package) and a dataset used by Choi et al. (2018)[5] and Korbak et al. (2019), which we will refer to as the obverter dataset, see Figure 2. The datasets are similar but offer different forms of visual variability. Shapes3d dataset includes images of 3D shapes. Each element is a $(64, 64, 3)$ RGB image, and is characterized by multiple features, such as shape or object hue. We choose images with values for both features ranging in $\{0, 1, 2, 3\}$. The obverter dataset contains images of four shapes (box, cylinder, ellipsoid, sphere) in four colors (blue, cyan, gray, green). Each image has dimensions $(128, 128, 3)$, and we use 1000 images for each shape–color pair. For details see Appendix A.

---

[4]We believe that Gumbel-Softmax approach is now an established choice in emergent communication research, see e.g. Lee et al. (2017), Mordatch and Abbeel (2017). Chaabouni et al. (2020) report that Gumbel-Softmax converges to similar solutions as REINFORCE but faster and is more stable.

[5]The dataset is available at `https://github.com/benbogin/obverter`. We used the code provided in the repository to generate 1000 images for each color-shape pair.

### 4.4 Training details

The sender and the receiver are implemented as feed-forward neural networks. To ensure the diversity of random initializations we run each experiment with 100 random seeds. For each seed, there were 100 evaluation runs, once every 2000 network updates. We report metrics averaged over the last 20 evaluation runs. For details on architecture and hyperparameters' choice, see Appendix B.

### 4.5 Compositionality measures

We measure compositionality in terms of four metrics used in emergent communication literature: topographic similarity, conflict count, context independence, and positional disentanglement. For all metrics, the higher values the better, except for conflict count, for which the reverse is

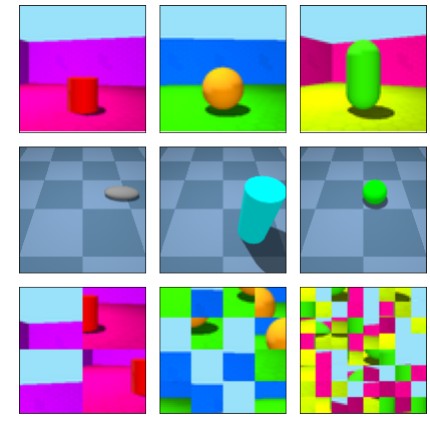

Figure 2: Top: shapes3d. Middle: obverter. Bottom: Scrambled shapes3d (32, 16, 8).

true. The results presented in Section 5 indicate that the metrics agree on the assessment of our results. Parallel to compositionality metrics, we also report accuracy, which we often refer to as *acc*.

**Topographic similarity** Topographic similarity (Brighton and Kirby, 2006; Lazaridou et al., 2018), or *topo* for short, is a popular measure of structural similarity between messages and features. Let $L_f : \mathcal{F} \times \mathcal{F} \to \mathbb{R}_+$ be a distance over features and $L_m : \mathcal{A}_s^* \times \mathcal{A}_s^* \to \mathbb{R}_+$ be a distance over messages. The topographical similarity is the Spearman $\rho$ correlation of $L_f$ and $L_m$ measured over a joint uniform distribution over features and symbols. We choose $L_m$ to be the Levenshtein (1966) distance and treat features $f \in \mathcal{F}$ as ordered pairs or features so we can choose $L_f$ to be the Hamming distance. We use topo as the main metric.

Figure 3 shows the expected value of topographic similarity for a random bijective language with a message length equal to 2. For 5-symbol alphabet, this value does not exceed 0.2 which sets a point of reference for compositionality results. For derivation see Appendix E.

**Conflict count** Let $\phi : \{1, \ldots, K\} \to \{1, \ldots, K\}$ be a permutation. The principal meaning of a symbol $s$ at position $j$ is defined as $\mathtt{m}(s, j; \phi) = \arg\max_v \mathtt{count}(s, j, v; \phi)$, where $\mathtt{count}(s, j, v; \phi)$ is defined as $\sum_{(\mathtt{img}, f) \in \mathcal{D}} \mathbf{1}(\ell(\mathtt{img})_j = s, f_{\phi(j)} = v)$, $v$ runs over all values of all features, and ties in $\arg\max$ are broken arbitrarily. Then, conflict count metric, *conf* for short, is defined as $\mathtt{conf} = \min_\phi \sum_{s,j} \mathtt{score}(s, j; \phi)$, where $\mathtt{score}(s, j; \phi) = \sum_{v \neq \mathtt{m}(s,j;\phi)} \mathtt{count}(s, j, v; \phi)$. Intuitively, $\mathtt{score}$ measures how many times the feature assigned to a symbol $s$ at a position $j$ diverts from its principal meaning $\mathtt{m}(s, j; \phi)$. $\mathtt{conf}$ sums these errors and takes min over possible orderings $\phi$. This metric was introduced in Kuciński et al. (2020).

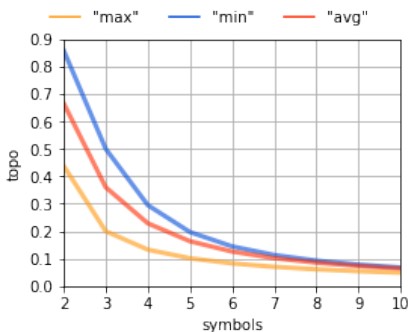

Figure 3: Expected value of topographic similarity for a random bijective language with message length 2, as a function of the alphabet size. "min", "max", and "avg" stand for different ways of computing ranks.

**Context independence** Context independence (Bogin et al. (2018)), abbreviated here as *cont*, measures the alignment between symbols forming a message and features $f_1, \ldots, f_K$. By $p(s|f)$, we mean the probability that the sender maps a feature $f$ to a message containing symbol $s \in \Sigma$. We define the inverse probability $p(f|s)$ similarly. Finally, we define $s^f = \arg\max_s p(f|s)$; $s^f$ is the symbol most often sent in presence of a feature $f$. Then, context independence is $\mathbb{E}(p(v^k|k) \cdot p(k|v^k))$; the expectation is taken with respect to the joint uniform distribution over features and symbols.

**Positional disentanglement** Let $s_j$ denote the $j$-th symbol of a message $f(d)$, and $c_1^j$ the feature with the highest mutual information with $s_j$, and $c_2^j$ with the second highest mutual information: $c_1^j = \arg\max_c \mathcal{I}(s_j; c)$, $c_2^j = \arg\max_{c \neq c_1^j} \mathcal{I}(s_j; c)$ where $\mathcal{I}(\cdot; \cdot)$ is mutual information and $c$ is a feature value. Then, positional disentanglement (Chaabouni et al., 2020) is defined as

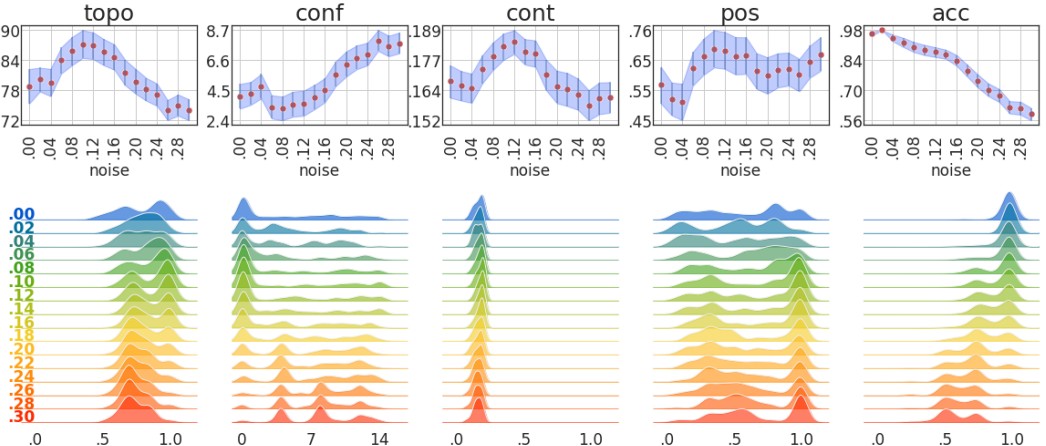

Figure 4: Results of the main experiment on shapes3d dataset. Top panel: average value of metrics for various noise levels. The shaded area corresponds to bootstrapped $95\%$-confidence intervals. Bottom panel: kernel density estimators for metrics and noise levels across seeds. Here *topo* stands for topographic similarity, *conf* for conflict count, *cont* for context independence, *pos* for positional disentanglement, and *acc* for accuracy. $\frac{1}{L} \sum_{j=1}^{L} \left( \mathcal{I}(s_j; c_1^j) - \mathcal{I}(s_j; c_2^j) \right) / \mathcal{H}(s_j)$, where $L$ is the maximum message length and $\mathcal{H}(s_j)$ is entropy over the distribution of symbols at $j$-th place in messages for each feature. We ignore positions with zero entropy.[6] We will call this measure *pos*, for short.

## 5 Experiment results

In this section, we study how different inductive biases for the model and the data influence compositionality. The main experiment is presented in Section 5.1 and the experiments in Section 5.2-5.4 are its variation. In the main experiment we assume that the message length is $K = 2$, $|\mathcal{A}_s| = 5$, and $|\mathcal{A}_r| = 8$ (see Section 4.2). Notice that both $\mathcal{A}_s$ and $\mathcal{A}_r$ differ from one another and from the number of possible feature values (which equals 4). It turns out that the qualitative results are similar for both the shapes3d and obverter datasets, and in the interest of brevity we only report the results for both in the main experiment. Supplementary material for this section can be found in Appendix D.

Our main findings include the fact that the noise indeed catalyzes the emergence of compositionality. There is an interesting dependence on the noise level: too high noise may impede learning, while too small noise vanishes within other sources of noise. Importantly, this phenomenon appears consistently across different biases. Using two head output of the network is the strongest bias toward compositionality (amongst studied). Finally, we show that compositionality can generalize to unseen cases when fine-tuning is allowed.

### 5.1 Main experiment: emergence of compositionaliy

The results, presented in Figure 4, illustrate how noise catalyzes the emergence of compositionality. The top panel of Figure 4 shows the important patterns for metrics: they improve until an extremal point is reached, and decline afterward. Topographic similarity achieves extremum for noise level $0.1$, reaching value $0.87$, which is a significant improvement upon $0.79$ for the lack of noise. The accuracy drops down with an increase of the noise level, as expected, however the speed of the decline increases. This shows that there is an interesting compositionality-accuracy trade-off. The bottom panel of Figure 4 complements the overall picture with a visualization of metrics' distribution. We see interesting dynamics in topographic similarity distribution with respect to change in the noise levels. Namely, it starts by accumulating mass at the higher spectrum of its values, reaching a peak for noise $0.1$, after which it transitions to a bimodal distribution, finally shifting its mass more towards the mediocre end of the spectrum. An interesting observation is that the results for experiments with achieved high accuracy, are not only better but also the effect of noise is more pronounced (for

---

[6]Positional disentanglement is related to residual entropy proposed by Resnick et al. (2020). Chaabouni et al. (2020) also proposed bag-of-words disentanglement, which assumes order-invariance of messages. Due to our architecture choice, this assumption is not met, hence we decided not to report this metric.

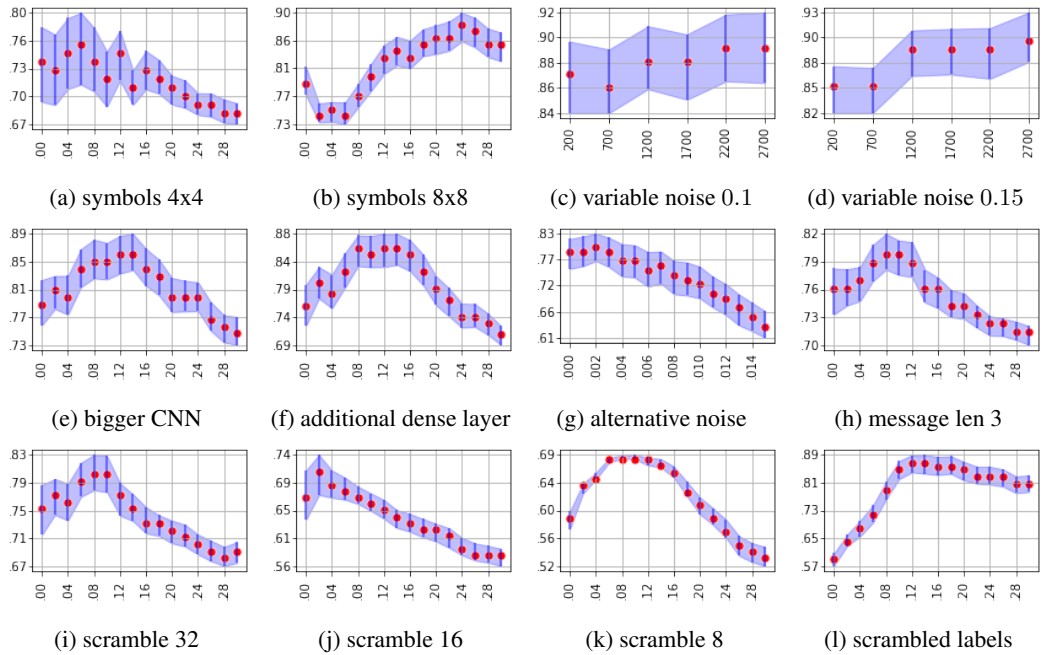

| (a) symbols 4x4 | (b) symbols 8x8 | (c) variable noise 0.1 | (d) variable noise 0.15 |
| (e) bigger CNN | (f) additional dense layer | (g) alternative noise | (h) message len 3 |
| (i) scramble 32 | (j) scramble 16 | (k) scramble 8 | (l) scrambled labels |

Figure 5: Topographic similarity for experiments in Sections 5.2 and Section 5.3. The shaded areas correspond to bootstrapped $95\%$-confidence intervals for average topo.

instance there are 47 seeds with accuracy exceeding $0.9$, for which noise $0.16$ yields $0.96$ topo). For the sake of brevity, we defer the discussion of this phenomenon to Appendix D.1. The accuracy undergoes a similar transformation as a topographic similarity. The detailed numerical analysis, as well as corresponding results for the obverter dataset, can be found in Appendix D.1.

### 5.2 Influence of model inductive bias

**Different number of symbols** Here we study the impact of a different number of symbols on compositionality. In our experiments we used the communication channel with $|\mathcal{A}_s| = 5$, giving a total of $25 = (5 \times 5)$ possible messages. This is slightly redundant since only $16 = (4 \times 4)$ is required, so this is the first case that we study here. It turns out, that allowing for only 16 messages makes the training less stable. For topographic similarity, see Figure 5(a), small to medium values of noise exhibit wide confidence intervals and it is statistically hard to distinguish between the metric values (this might be attributed to a bimodal distribution of topo in this noise range, see Figure 15 in Appendix D.2). For larger values of noise (greater than $0.16$), topo starts to visibly decline. As the second experiment, we considered the total of $64 = (8 \times 8)$ messages. Interestingly, the topographic similarity values for the small noise regime (up to $0.08$) do not improve over the baseline value ($0.79$), see Figure 5(b). This behavior changes for medium to large values of noise, where we can observe a visible increase in topo, peaking at $0.88$ with a noise level of $0.24$. For details see Appendix D.2.

**Variable noise** Understanding what happens under varying noise is an interesting and subtle problem. It would most probably arise in more complex situations when a communication channel is a part of a bigger system. Additionally, it might be similar to a phenomenon observed in supervised learning, indicating that training can benefit from learning rate warmup (see e.g. Goyal et al. (2017), Frankle et al. (2020)). In this paragraph, we discuss a simple experiment with increasing noise and leave a more nuanced study for further work. More precisely, in the initial stage of training the noise is kept at the level $\epsilon_0$, and after $T \in \{200, 700, 1200, 1700, 2200, 2700\}$ network updates it is switched to a different value, $\epsilon_T$, and kept there for the rest of the training. The results for topographic similarity are presented in Figures 5(c)-(d), where $\epsilon_0 = 0$ and $\epsilon_T = 0.1$ and $\epsilon_T = 0.15$, respectively. In Figure 5(d) we see that topo increases from $0.85$ for $T = 200$, to $0.9$ for $T = 2700$. The effect in Figure 5(c) is weaker and the variance in the results is quite high. The details can be found in Appendix D.3.

**Sensitivity with respect to small architecture changes** This set of experiments aims to check the impact of changing the parameters of CNN and the dense layers in the agents' network. In the former experiment, we change the number of filters in the sender's CNN architecture from two layers with 8 filters, to two layers with 16 filters. This results in the slight change of topographic similarity

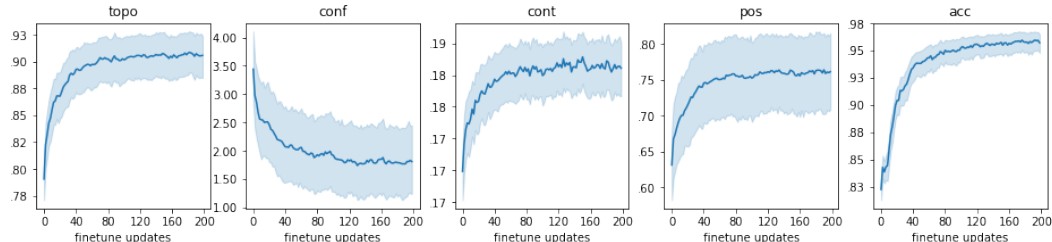

Figure 6: Fine-tuning for a baseline agent trained with noise 0.1 and with several features removed from the training set (the diagonal from the shape-by-color matrix). On the x-axis is the number of finetuning updates.

profile, see Figure 5(e), with the highest average value of $0.86$ for noise range $0.14$ significantly outperforming the zero-noise case ($0.79$). For the second experiment, we added a dense layer (with 64 neurons) to the receiver's architecture, see Figure 5(f). This again shows improvement of compositionality due to noise, although the noise in the range $[0.08, 0.16]$ performs roughly the same (see also Appendix D.4).

**Sensitivity to noisy channel implementation** Here we consider an alternative noisy channel implementation, where the noise permutes the sampled symbols, as opposed to distorting the distribution of the symbols (see Appendix C). It turns out that with this change, the scale of noise where the interesting things happen changes as well, see Figure 5(g). Namely, when compared with the main experiment, noise values are in the range smaller by the order of magnitude (we report results for noise levels in $\{0.000, 0.001, \ldots, 0.015\}$). The results suggest that the small values of noise can help, while the larger noise levels lead to a decline in compositionality, see also Appendix D.5.

**Longer message** In this section, we discuss the experiment with an additional feature (with floor color acting as the third feature) and a message of length 3. The result for topographic similarity can be seen in Figure 5(h), see also Appendix D.6. The overall topographic similarity level is lower than in the main experiment, however, the distinctive peak is visible, here for noise level $0.08$.

## 5.3 Data inductive biases

**Visual priors** This experiment was intended to check how much the CNN-backed input is relevant in the compositionality context. We could conjecture that CNN may facilitate shape recognition and therefore be the driving force in the emergence of languages compositional with respect to the canonical shape, color split. To check this we impair the prior by scrambling images, as depicted in the bottom panel Figure 2. An image is scrambled by splitting it into $(64/x)^2$ disjoint tiles of height and width equal to $x$, and randomly reshuffling them. This procedure significantly distorts the accuracy profile of the method (see Appendix D.7). In particular, each transition from coarser to finer tiles, the accuracy decreases significantly: for zero-noise it drops from $0.97$ for no tiling, to $0.95$ for $x = 32$, to $0.79$ for $x = 16$ and $0.53$ for $x = 8$. The overall compositionality metrics decrease as well, but the characteristic peak for some positive noise level is still present, see Figure 5(i)-(k). Having said that, the metrics incur a significant boost, when computed for a subset of experiments with high accuracy. We conjecture that the explanation is that the CNN prior is indeed relevant but is not the only one (the output considered in the next section is another).

**Scrambled labels** In this experiment, we aimed to understand how much the overall architecture output is important in the emergence of compositionality. The receiver's network has a two-headed output and the training framework uses a factorized loss function. In the standard setting, we compare the heads' outputs with 'color' and 'shape' respectively, therefore we reflect human priors from the data. In this experiment, we distort this setting, permuting the set of (color, shape) and factorizing them into new labels (see Appendix D.8). We use a random permutation, so the new labels are abstract and correspond to some joint color-shape concepts. In our experiments, we show that languages, which emerge are compositional with respect to these new concepts, see Figure 5(l). Consequently, they are *not* compositional in the standard color-shape framework. This is in line with the claims of Section 3.2 and further highlights that the output inductive bias is essential.

## 5.4 Generalization

**Network features** In this paragraph, we present visualizations of the sender network, to gain some insights into what the network is doing, and whether we can observe any obvious overfit. In the

upper panel of Figure 7, a t-SNE (Van der Maaten and Hinton (2008)) visualization is shown for both the raw shapes3d dataset and the last dense layer of a trained agent with noise 0.1. We observe that the network successfully disentangles features of the data, which in the raw dataset appears to be entangled. This could be a good sign in the context of generalization[7]. Furthermore, Figure 7 (bottom panel) shows occlusion and saliency maps for a sample image from shapes3d dataset. The network seems to pay attention to parts of the image that could be relevant for the task.

**Zero-shot and fine-tuning** In this experiment, we study the generalization properties of the training protocol. We trained the baseline agent with noise 0.1 with several features removed from the training set (a diagonal from the shape-by-color matrix) and used the removed set to analyze the out-of-distribution performance. In the zero-shot setup, the messages on these observations were not in line with the compositional structure acquired on the training set, which is reflected by rather unimpressive outcomes for each metric (initial values for each metric presented in Figure 6). This may suggest that for a zero-shot generalization stronger biases might be required (see e.g. Słowik et al. (2020a,b)). However, after a few network updates on the whole dataset (without the removed diagonal) we observed a quite significant increase in compositionality metrics, see Figure 6. The highest improvement can be seen roughly in the first 50 additional network updates.

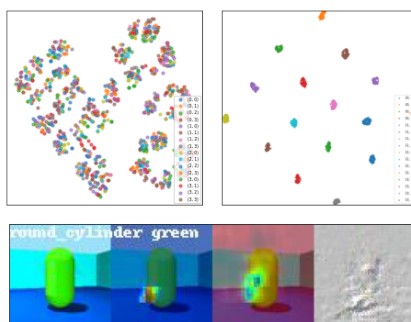

Figure 7: t-SNE visualization (with perplexity 50 and 10000 steps). Top left: shapes3d, top right: features of the last dense layer. Bottom: occlusion and saliency maps.

## 6  Limitations of the method

Here we summarize a few limitations of the method, that we hope to overcome in future research.

**Theory** Theorem 2 uses several restrictive assumptions, e.g. fixed message length or alphabet size equal to the range of feature values. In a general setting, it could be the case that noise would promote non-compositional, error-correcting, communication protocols. Stating the general conditions for the emergence of compositionality is an open problem.

**Choice of datasets** Confirmation on non-synthetic datasets is needed to fully underpin our method.

**Communication protocol** We assume a simple communication protocol with messages of length two and two features, each taking four values (except for experiments in Section 5.2).

**Compositionality model** We assume a rather simple compositionality model based of independent features. Exploring non-trivial compositionality (Steinert-Threlkeld, 2020; Korbak et al., 2020) might be an important conceptual development.

**Architecture** Our architecture might implicitly exhibit some unknown biases that increase compositionality. On the other hand, the architecture might be too simple to achieve strong results in tasks such as zero- or few- shot generalization.

**Multi-agent interactions** We used supervised training setup, it would be natural to test the influence of noise in reinforcement learning scenarios.

## 7  Conclusions

In this paper, we theoretically show that inductive biases on both the training framework and the data are needed for the compositionality to emerge spontaneously in signaling games. We then formulate inductive biases in the loss function and prove that they are sufficient to achieve compositionality when coupled with communication over a noisy channel. Consequently, we highlight the catalytic role of noise in the emergence of compositionality. We perform a series of experiments in order to understand different aspects of the proposed framework better. We empirically validate that, indeed, a certain range of noise levels, dependent on the model and the data, promotes compositionality. Our work is foundational research and does not lead to any direct negative applications.

---

[7]It is known that t-SNE does not necessarily represent cluster sizes, distances, and respective positions (see Wattenberg et al. (2016)). Hence, t-SNE cannot be expected to serve as a compositionality metric.

## Acknowledgments and Disclosure of Funding

The work of Piotr Miłoś was supported by the Polish National Science Center grant UMO-2017/26/E/ST6/00622. The work of Tomasz Korbak was supported by the Leverhulme Doctoral Scholarship. This research was supported by the PL-Grid Infrastructure. Our experiments were managed using `https://neptune.ai`. We would like to thank the Neptune team for providing us access to the team version and technical support.

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
