# A  Datasets

## A.1  Shapes3d

Shapes3d is a dataset (see Burgess and Kim (2018) and the Tensorflow Datasets package) consisting of $64 \times 64 \times 3$ RGB images of objects having six independent features (floor color, wall color, object color, scale, shape, and orientation), see Figure 2. In this paper, we use four shapes (cube, cylinder, sphere, and rounded cylinder) and four object colors (red, orange, yellow, and green), totaling 192000 images.

## A.2  Obverter

The obverter dataset (Bogin et al. (2018)) is available at the following address: `https://github.com/benbogin/obverter`. The original dataset consists of $128 \times 128 \times 3$ RGB images of objects having eight colors and five shapes. In this paper, we used four colors (blue, cyan, gray, green) and four shapes (box, cylinder, ellipsoid, sphere), see Figure 2. We have generated 1000 samples for each color-shape combination using a generation script available at the dataset repository, hence the total number of images is 160000. Since the qualitative results were similar, and in the interest of brevity, we report results for the obverter only for the main experiment (Appendix D.1).

# B  Experimental setup

## B.1  Architecture

The network consists of three main parts: the sender, the receiver, and the noisy discrete channel between them see Figure 8. The sender network consists of two convolutional layers (with 8 filters, kernel $3 \times 3$, stride 1, and elu activation function), each coupled with a $2 \times 2$ max pool layer with stride 2. The last max pool layer's output is passed through two dense layers (with 64 neurons and elu activation) and a linear classifier with softmax for each symbol. The noisy channel layer consists of a dense layer with $|A_s|$ neurons, a fixed weights matrix, and a $\log$ activation function. This is followed by a Gumbel softmax layer. The receiver network takes two encoded symbols as input and concatenates them to obtain one input vector $s$. Consequently, $s$ and $1 - s$ are passed to the dense layers, the result is summed up and processed by the elu activation function and two dense layers (similarly to Kaiser and Bengio (2018)). There are two linear classifiers with a softmax layer at the output: one for the shape and one for the color. Each dense layer in the receiver has 64 neurons.

## B.2  Hyperparameters and training

For training, we used $\lambda_{KL} = 0.01$, $\lambda_{l_2} = 0.0003$, an Adam optimizer (with $\beta_1 = 0.9$, $\beta_2 = 0.999$), learning rate 0.0001, and a batch size of 64. The same set of hyperparameters was used for all the experiments. The hyperparameters were chosen on the original obverter dataset available at the repository referenced in Appendix A.2. We used a grid search over parameters: learning rate ($1e-2, 1e-3, 1e-4, 3e-4$), kl regularization coefficient ($1e-1, 1e-2, 2e-2, 3e-2, 1e-3, 3e-3, 5e-3, 1e-4$), the number of CNN's filters ($8, 16$), the CNN's filter sizes ($3 \times 3, 5 \times 5$), the sender's embedding size ($32, 64$), $l_2$ regularizer weight ($1e-2, 1e-3, 3e-3, 1e-4, 3e-4, 1e-6$), and the number of neurons in receiver's dense layers ($32, 64$).

Each experiment was run on 100 seeds and had 200000 network updates. The dataset was split into the training set ($90\%$ of the total) and on the test set (remaining $10\%$ of the dataset). The evaluation was done every 2000 updates on the test set.

## B.3  Infrastructure used

The typical configuration of a computational node used in our experiments was: the Intel Xeon E5-2697 2.60GHz processor with 128GB memory. On a single node, we ran 4 or 5 experiments at the same time. A single experiment (single seed) takes about 5 hours. We did not use GPUs; we found that with the relatively small size of the network (see Appendix B.1) it offers only slight wall-time improvement while generating substantial additional costs.

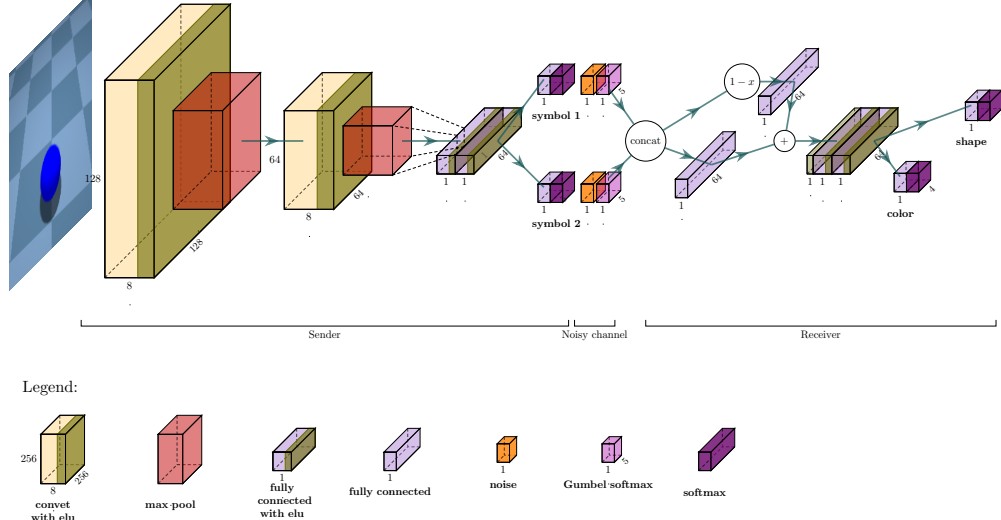

Figure 8: The architecture of the neural network. Here $|A_s| = 5$.

# C  Training pipeline

## C.1  Detailed derivation

The dataset of images observed by the sender is denoted by $\mathcal{D}$. Each element of $\mathcal{D}$ has $K$ independent features $f_1, \ldots, f_K$ (here we consider $K = 2$). Both the sender and the receiver are modeled as neural networks ($s_\theta$ and $r_\psi$, respectively; for details see Appendix B). The sender network takes an image from $\mathcal{D}$ as input and returns a distribution over the space of messages of length $L$ (here we assume $L = K = 2$). We assume that conditionally on the image, the symbols in the message are independent and take values in a finite alphabet $\mathcal{A}_s = \{1, \ldots, d_s\}$. We furthermore assume, that features are enumerated with $A_f = \{1, \ldots, d_f\}$ and the receiver's alphabet is $\mathcal{A}_r = \{1, \ldots, d_r\}$. Formally, $s_\theta(\texttt{img}) = (s_\theta^i(\texttt{img}))_{i=1}^K$, where $s_\theta^i(\texttt{img}) = (s_{j,\theta}^i(\texttt{img}))_{j=1}^{d_s} \in \mathcal{P}(A_s)^8$ represents the probability distribution corresponding to the $i$th symbol. Define a function $\texttt{noise} \colon \mathcal{P}(A_s) \to \mathbb{R}^{d_s}$ as follows:

$$\texttt{noise}(x) = \log(Wx), \tag{4}$$

where $W \in \mathbb{R}^{d_s \times d_s}$ is a fixed matrix, such that $Wx > 0$ and $Wx \in \mathcal{P}(A_s)$, for any $x \in \mathcal{P}(A_s)^9$.
 The second condition on $W$ is satisfied, for instance, by a family of stochastic matrices; several examples are also given at the end of this section. In this paper, we use $W$ defined as

$$W_{ij} = \begin{cases} 1 - \varepsilon, & i = j, \\ \frac{\varepsilon}{d_s - 1}, & i \neq j. \end{cases}$$

Let $\widehat{s}_\theta^i(\texttt{img})$ denote the logits of $i$th symbol distribution which passes through the noisy channel:

$$\widehat{s}_\theta^i(\texttt{img}) = \texttt{noise}(s_\theta^i(\texttt{img})).$$

Suppose further that $g^i = (g_1^i, \ldots, g_{d_s}^i)$ is a vector of i.i.d. Gumbel$(0, 1)$ random variables and define the following functions:

$$\texttt{gumbel\_sample}(x; g) = \arg\max_i (x_i + g_i),$$

$$\texttt{gumbel\_softmax}(x; \tau, g)_i = \frac{\exp((x_i + g_i)/\tau)}{\sum_{j=1}^k \exp((x_j + g_j)/\tau)}.$$

---

[8] $\mathcal{P}(A) = \{p \in \mathbb{R}^{|A|} : p_i \geq 0, \sum_{i \in A} p_i = 1\}$.

[9] We could also define $\texttt{noise}$ for all $x \in \mathbb{R}^m$, for some $m$, by first applying $\texttt{softmax}$ to $x$, and then using equation 4.

Let
$$\widehat{\mathfrak{m}}_i = \texttt{gumbel\_softmax}(\widehat{s}_\theta^i(\texttt{img}); \tau, g^i) \in \mathbb{R}^{d_s}.$$

The receiver neural network is denoted as $r_\psi(\mathfrak{m}) = (r_\psi^i(\mathfrak{m}))_{i=1}^K$, where $r_\psi^i(\mathfrak{m}) = (r_{j,\psi}^i(\mathfrak{m}))_{j=1}^d \in \mathcal{P}(A_r)$ represents the probability distribution on $A_r$, corresponding to the $i$th feature.

In the Straight-Through mode (see Jang et al. (2016)), $r_\psi$ takes $\widehat{\mathfrak{m}}$ as input half of the time, and the remaining half of the time, it takes $\widetilde{\mathfrak{m}}$. Here

$$\widetilde{\mathfrak{m}} = \texttt{stop\_gradient}(\overline{\mathfrak{m}} - \widehat{\mathfrak{m}}) + \widehat{\mathfrak{m}},$$
$$\overline{\mathfrak{m}}_i = \texttt{one\_hot}(\omega_i) \in \mathbb{R}^{d_s},$$
$$\omega_i = \texttt{gumbel\_sample}(s_\theta^i(\texttt{img}); g^i) \in A^{d_s},$$

i.e. $(\omega_1, \ldots, \omega_K)$ is a sampled noisy message. The neural networks are trained using the following loss function:
$$\mathcal{L} = \mathcal{L}_{xent} + \lambda_{KL}\mathcal{L}_{KL} + \lambda_{l_2}\mathcal{L}_{l_2}.$$
The cross-entropy loss is defined as

$$\mathcal{L}_{xent} = -\mathbb{E}_{(\texttt{img}, f_1, \ldots, f_K) \sim \mathcal{D}} \left[ \sum_{i=1}^K \log r_{f_i, \psi}(\widetilde{\mathfrak{m}}(\texttt{img})) \right].$$

Furthermore, $\mathcal{L}_{KL} = \mathbb{E}_{x \sim \mathcal{D}} \left[ \sum_{i=1}^K \text{KL}(U(\mathcal{A}_s) || s_\theta^i(x)) \right]$ and $\mathcal{L}_{l_2} = ||\theta||_2 + ||\psi||_2$, where $U(\mathcal{A}_s)$ denotes the uniform distribution of $\mathcal{A}_s$. The KL loss incentives the language to be a one-to-one mapping.

### C.2   Alternative noise architecture

The above implementation of noise is not the only one possible. We can apply noise after the message is formed. Denote the uncorrupted sender's message

$$\mathfrak{m}_i = \texttt{gumbel\_softmax}(\log s_\theta^i(\texttt{img}); \tau, g^i) \in \mathbb{R}^{d_s}.$$

Further, let $\rho_i$ be uniformly sampled random permutations of $A_s$ and $\epsilon_i$ are Bernoulli random variables such that $\mathbb{P}(\epsilon_i = 1) = \epsilon = 1 - \mathbb{P}(\epsilon_i = 0)$, with $\epsilon > 0$. We define the noise matrices

$$N_i = \epsilon_i P^{\rho_i} + (1 - \epsilon_i)\mathbb{I},$$

where $P^{\rho_i}$ is the permutation matrix corresponding to $\rho_i$ and $\mathbb{I}$ is the identity matrix. The corrupted message is then given by
$$\widehat{m}_i = N_i \mathfrak{m}_i.$$
Above we assume that $\widehat{m}_i, \mathfrak{m}_i$ is encoded in the one-hot vector form $\in \mathbb{R}^{d_s}$. We note that this particular implementation of the noise has the advantage of being differentiable using the standard autograd methods.[10]

## D   Detailed results

Each experiment was run on 100 seeds. When presenting results we give $95\%$-confidence intervals, bootstrapped using 4000 resamples.

### D.1   Main experiment

The results for the shapes3d dataset are is visualized in Figure 9 (which is the copy of Figure 4 placed here for convenience), and numerical results are summarized in Table 1. Similarly, for the obverter dataset, the results can be found in Figure 10 and in Table 2. From the qualitative perspective, the results for the obverter dataset are similar to the ones obtained for the shapes3d dataset. Having said

---

[10]We cannot differentiate the random sampling of $\rho_i, \epsilon_i$ but we can differentiate multiplication with respect to $N_i$.

that, we can see that overall the metrics are more stable (resulting in narrower confidence intervals), the best performing noise level is slightly different (0.12), and the density evolution of topo across different noise levels appears to be smoother.

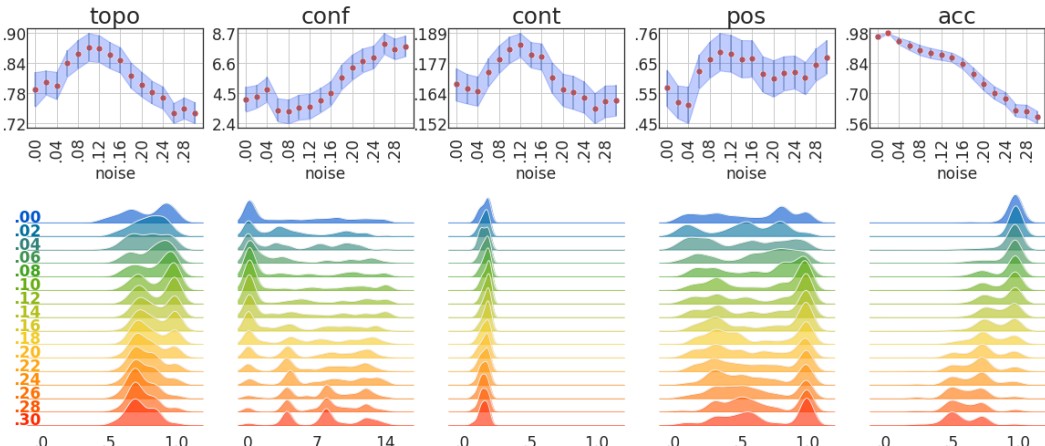

Figure 9: Shapes3d dataset. Top panel: average value of metrics for various noise levels. The shaded area corresponds to bootstrapped 95%-confidence intervals. Bottom panel: kernel density estimators for metrics and noise levels across seeds. Here *topo* stands for topographic similarity, *conf* for conflict count, *cont* for context independence, *pos* for positional disentanglement and *acc* for accuracy.

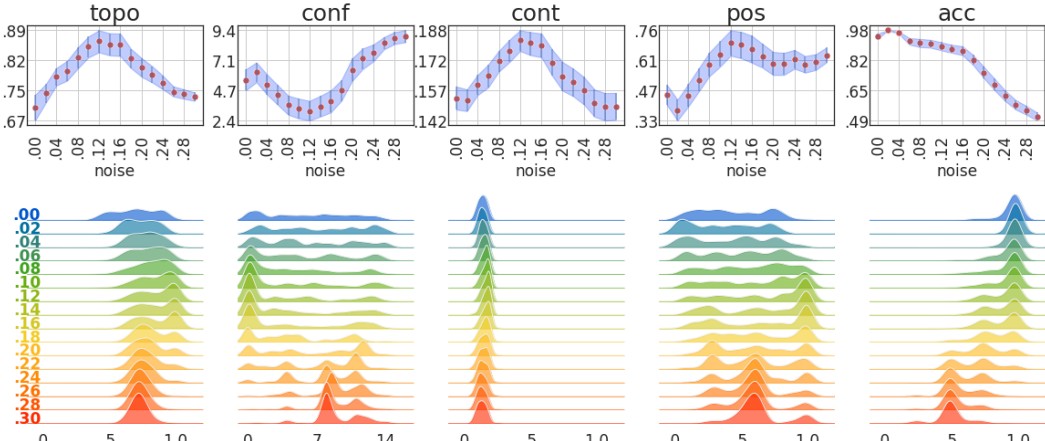

Figure 10: Obverter dataset. Top panel: average value of metrics for various noise levels. The shaded area corresponds to bootstrapped 95%-confidence intervals for this estimator. Bottom panel: kernel density estimators for metrics and noise levels across seeds. Bottom panel: kernel density estimators for metrics and noise levels across seeds. Here *topo* stands for topographic similarity, *conf* for conflict count, *cont* for context independence, *pos* for positional disentanglement and *acc* for accuracy.

As observed in Section 5.1, there is an interesting and non-trivial interplay between compositionality and accuracy. In Figure 11 we visualize topographic similarity behavior when conditioned on experiments with high accuracy (for shapes3d; for the obverter see Figure 12). We can see an increase in the metrics values across all noise levels with the increase of accuracy. For example, for noise 0.1 and threshold 0.85, topo equals 0.91, four percentage points higher than for unconditional case. We can see that increasing the threshold also strengthen the impact of noise on compositionality (which can be seen by increasing the profile of topo plots). Additionally, the count curves for smaller noises dominate the ones for higher noises. Notice, however, that the number of experiments exceeding some accuracy threshold declines as the threshold increases. For example, there are 80 experiments with noise level 0.1 exceeding the threshold of 0.85 accuracy, but only 4 with noise level 0.3. This implies that conclusions for high accuracy thresholds should be treated with care.

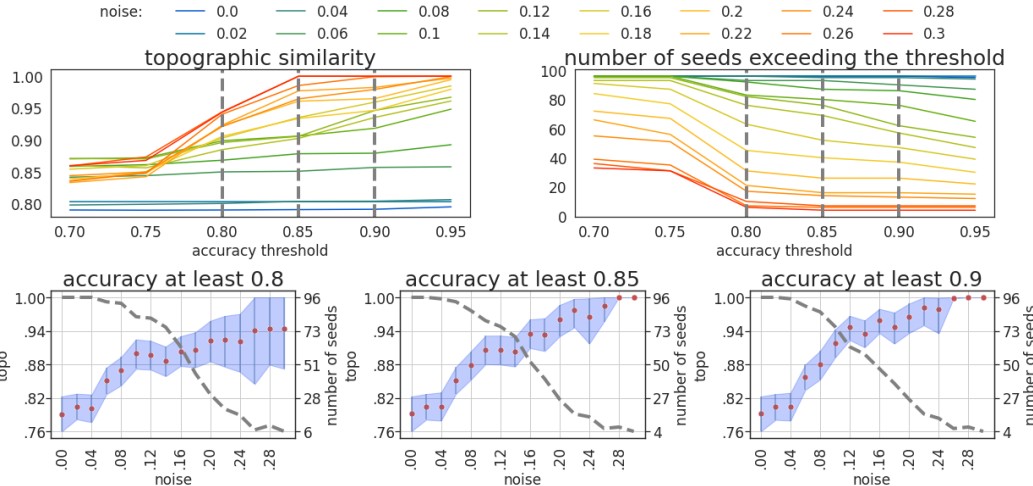

Figure 11: Shapes3d dataset. Top left: The values of topo computed for each noise level (hues) and on seeds exceeding a certain accuracy threshold ($x$-axis). Vertical dashed lines represent three cross-sections, visualized in the bottom panel. Top right: Similar to the left panel, but instead of topo we visualize the number of seeds with accuracy at least as a given threshold ($x$-axis). Vertical dashed lines represent three cross-sections, visualized in the bottom panel. Bottom: Each of the plots represents a cross-section of the plots in the top panel, taken at points 0.80, 0.85, and 0.90, respectively. On the left axis of each figure is the range of topo, whereas on the right axis is the number of seeds with accuracy exceed the corresponding level. On the $x$-axis are the noise levels. The scatter plot with 95%-confidence intervals represents the values of topo. The gray dashed line represents the number of seeds with accuracy exceeding a given threshold, for each of the noise levels.

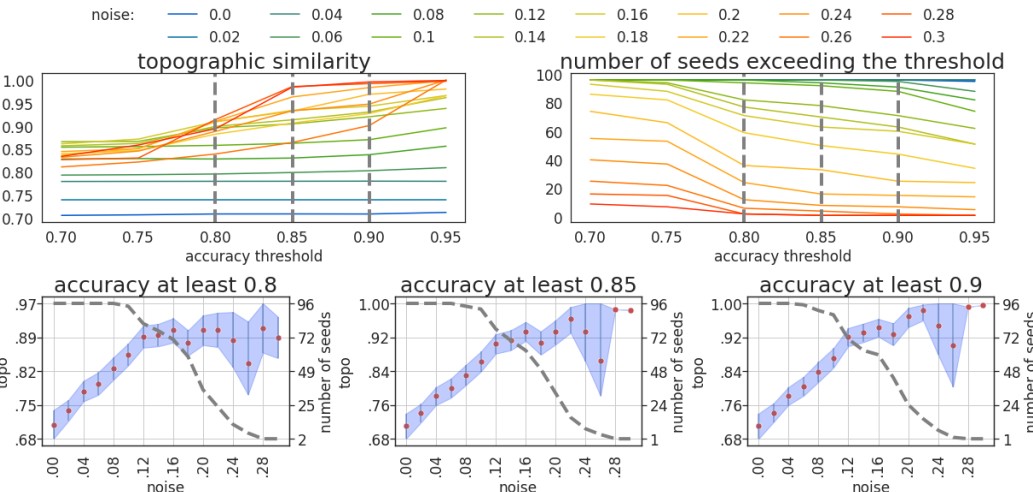

Figure 12: Obverter dataset. Top left: The values of topo computed for each noise level (hues) and on seeds exceeding a certain accuracy threshold ($x$-axis). Vertical dashed lines represent three cross-sections, visualized in the bottom panel. Top right: Similar to the left panel, but instead of topo we visualize the number of seeds with accuracy at least as a given threshold ($x$-axis). Vertical dashed lines represent three cross-sections, visualized in the bottom panel. Bottom: Each of the plots represents a cross-section of the plots in the top panel, taken at points 0.80, 0.85, and 0.90, respectively. On the left axis of each figure is the range of topo, whereas on the right axis is the number of seeds with accuracy exceed the corresponding level. On the $x$-axis are the noise levels. The scatter plot with 95%-confidence intervals represents the values of topo. The gray dashed line represents the number of seeds with accuracy exceeding a given threshold, for each of the noise levels.

For the main experiment, we also provide pair-plots for all metrics and three noise levels: 0.0, 0.1, and 0.2, see Figure 13 and Figure 14 for the shapes3d and the obverter datasets, respectively. It shows the Spearman correlation between metrics, broken down to noise levels (color-coded circles in the

upper triangle of the grid) as well as for the entire group (white circle in the upper triangle of the grid). We can see in Figure 13 that the metrics are highly correlated (the negative correlation with conflict count follows by definition of the metric, see Section 4.5). For zero-noise (blue color), we see that accuracy is high irrespective of the compositionality metrics, resulting in an almost vertical line. Looking at the topo-acc cell, we can see that for mediocre accuracy values, the noise level 0.2 (yellow color) tends to score higher in topo metrics than the noise level 0.1 (green color). This relation reverses for accuracy values closer to 1.0. For topo-conf and topo-pos cells, we see a visible linear correlation, with the effect weakening slightly for higher noise levels. At the topo-cont cell, a lot of the mass of all noise levels is occupied in the center, but the noise level 0.1 more frequently stays in the upper right corner. Similar observations can be done for Figure 14.

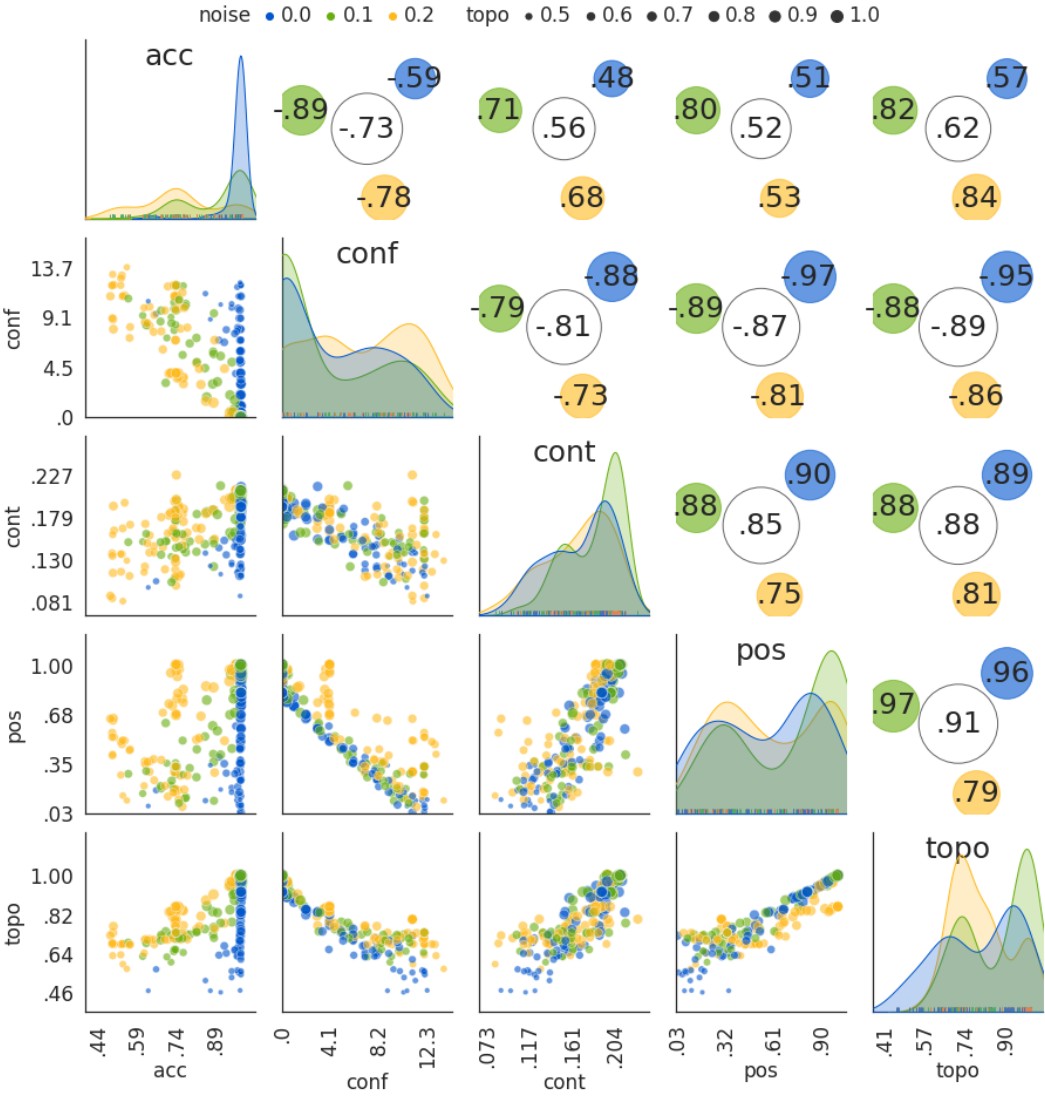

Figure 13: Shapes3d dataset. The lower triangle of the grid: color-coded scatter plot for experiments with different noise levels. The upper triangle of the grid: visualization of Spearman correlation between metrics. The large circle with white fill shows the correlation of metrics value without a split into noise levels. The smaller color-coded circles represent the in-group correlation. Diagonal: kernel density estimators for each metric and noise level. Here *topo* stands for topographic similarity, *conf* for conflict count, *cont* for context independence, *pos* for positional disentanglement and *acc* for accuracy.

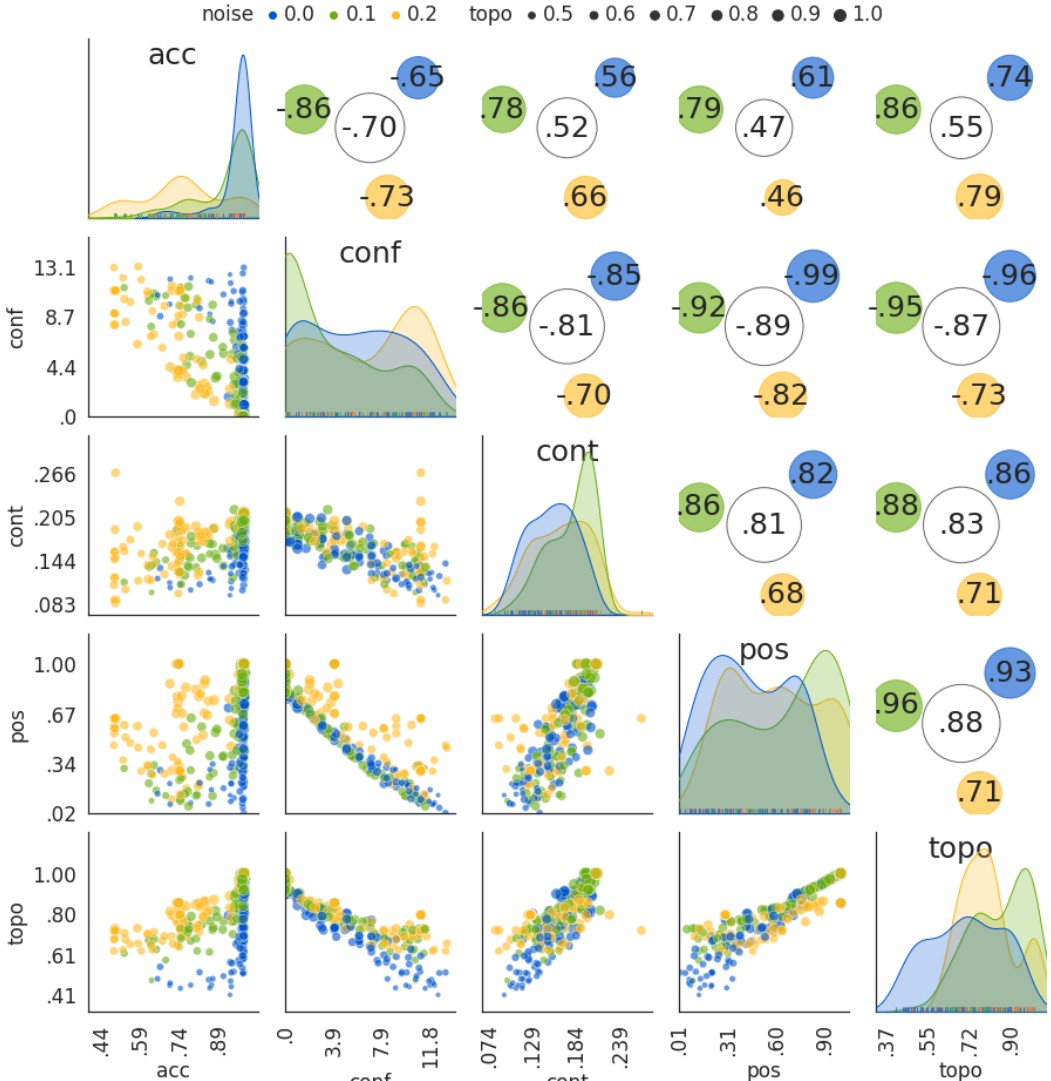

Figure 14: Obverter dataset. The lower triangle of the grid: color-coded scatter plot for experiments with different noise levels. The upper triangle of the grid: visualization of Spearman correlation between metrics. The large circle with white fill shows the correlation of metrics value without a split into noise levels. The smaller color-coded circles represent the in-group correlation. Diagonal: kernel density estimators for each metric and noise level. Here *topo* stands for topographic similarity, *conf* for conflict count, *cont* for context independence, *pos* for positional disentanglement and *acc* for accuracy.

| noise | topo | conf | cont | pos | acc |
|---|---|---|---|---|---|
| 0.00 | 0.79 [0.76, 0.82] | 4.08 [3.21, 4.94] | 0.168 [0.161, 0.175] | 0.57 [0.51, 0.63] | 0.97 [0.96, 0.98] |
| 0.02 | 0.80 [0.78, 0.82] | 4.26 [3.46, 5.08] | 0.166 [0.160, 0.172] | 0.52 [0.47, 0.58] | 0.98 [0.98, 0.98] |
| 0.04 | 0.80 [0.77, 0.82] | 4.78 [3.92, 5.64] | 0.165 [0.159, 0.171] | 0.51 [0.45, 0.57] | 0.95 [0.93, 0.96] |
| 0.06 | 0.84 [0.81, 0.86] | 3.31 [2.52, 4.15] | 0.173 [0.167, 0.179] | 0.62 [0.57, 0.68] | 0.93 [0.90, 0.95] |
| 0.08 | 0.86 [0.83, 0.88] | 3.22 [2.40, 4.06] | 0.179 [0.173, 0.184] | 0.66 [0.60, 0.73] | 0.91 [0.88, 0.93] |
| 0.10 | 0.87 [0.84, 0.90] | 3.49 [2.64, 4.41] | 0.183 [0.177, 0.188] | 0.69 [0.63, 0.76] | 0.89 [0.87, 0.92] |
| 0.12 | 0.87 [0.84, 0.89] | 3.58 [2.71, 4.47] | 0.184 [0.179, 0.189] | 0.69 [0.62, 0.75] | 0.88 [0.86, 0.90] |
| 0.14 | 0.85 [0.83, 0.88] | 3.99 [3.16, 4.87] | 0.180 [0.175, 0.186] | 0.67 [0.60, 0.73] | 0.87 [0.85, 0.89] |
| 0.16 | 0.84 [0.82, 0.87] | 4.49 [3.66, 5.36] | 0.180 [0.174, 0.185] | 0.67 [0.61, 0.73] | 0.84 [0.82, 0.87] |
| 0.18 | 0.81 [0.79, 0.84] | 5.63 [4.78, 6.52] | 0.171 [0.164, 0.177] | 0.62 [0.56, 0.68] | 0.80 [0.77, 0.82] |
| 0.20 | 0.80 [0.77, 0.82] | 6.32 [5.45, 7.22] | 0.166 [0.159, 0.173] | 0.60 [0.54, 0.66] | 0.75 [0.72, 0.78] |
| 0.22 | 0.78 [0.76, 0.81] | 6.78 [5.97, 7.64] | 0.165 [0.158, 0.172] | 0.62 [0.56, 0.68] | 0.71 [0.68, 0.74] |
| 0.24 | 0.77 [0.75, 0.79] | 7.01 [6.24, 7.83] | 0.163 [0.156, 0.169] | 0.62 [0.57, 0.68] | 0.68 [0.65, 0.71] |
| 0.26 | 0.74 [0.72, 0.76] | 7.97 [7.25, 8.71] | 0.158 [0.152, 0.164] | 0.60 [0.55, 0.66] | 0.62 [0.59, 0.65] |
| 0.28 | 0.75 [0.73, 0.77] | 7.58 [6.88, 8.32] | 0.161 [0.154, 0.168] | 0.65 [0.59, 0.70] | 0.62 [0.59, 0.65] |
| 0.30 | 0.74 [0.72, 0.76] | 7.80 [7.08, 8.54] | 0.161 [0.155, 0.168] | 0.67 [0.62, 0.73] | 0.59 [0.56, 0.62] |

Table 1: Shapes3d dataset. Results for the metrics for selected noise levels. Shown in square brackets are bootstrapped 95%-confidence intervals. Here *topo* stands for topographic similarity, *conf* for conflict count, *cont* for context independence, *pos* for positional disentanglement and *acc* for accuracy.

| noise | topo | conf | cont | pos | acc |
|---|---|---|---|---|---|
| 0.00 | 0.71 [0.67, 0.74] | 5.54 [4.73, 6.37] | 0.153 [0.147, 0.159] | 0.45 [0.40, 0.50] | 0.95 [0.94, 0.97] |
| 0.02 | 0.74 [0.72, 0.76] | 6.15 [5.36, 6.91] | 0.152 [0.146, 0.157] | 0.38 [0.33, 0.43] | 0.98 [0.98, 0.98] |
| 0.04 | 0.78 [0.76, 0.80] | 5.19 [4.42, 5.97] | 0.160 [0.155, 0.165] | 0.45 [0.39, 0.50] | 0.97 [0.97, 0.98] |
| 0.06 | 0.79 [0.77, 0.82] | 4.36 [3.61, 5.14] | 0.164 [0.158, 0.170] | 0.52 [0.46, 0.58] | 0.92 [0.90, 0.94] |
| 0.08 | 0.83 [0.80, 0.85] | 3.58 [2.84, 4.32] | 0.172 [0.166, 0.177] | 0.59 [0.54, 0.65] | 0.91 [0.89, 0.94] |
| 0.10 | 0.85 [0.82, 0.88] | 3.30 [2.54, 4.09] | 0.177 [0.171, 0.183] | 0.64 [0.58, 0.70] | 0.91 [0.89, 0.93] |
| 0.12 | 0.86 [0.84, 0.89] | 3.14 [2.39, 3.93] | 0.182 [0.177, 0.188] | 0.70 [0.64, 0.76] | 0.90 [0.87, 0.92] |
| 0.14 | 0.86 [0.83, 0.88] | 3.50 [2.69, 4.31] | 0.181 [0.175, 0.187] | 0.69 [0.62, 0.75] | 0.88 [0.86, 0.90] |
| 0.16 | 0.86 [0.83, 0.88] | 3.87 [3.02, 4.74] | 0.180 [0.174, 0.186] | 0.67 [0.60, 0.73] | 0.87 [0.85, 0.90] |
| 0.18 | 0.82 [0.80, 0.85] | 4.73 [3.93, 5.55] | 0.171 [0.164, 0.178] | 0.63 [0.58, 0.69] | 0.82 [0.79, 0.85] |
| 0.20 | 0.80 [0.78, 0.82] | 6.34 [5.46, 7.22] | 0.164 [0.157, 0.171] | 0.60 [0.54, 0.65] | 0.75 [0.72, 0.78] |
| 0.22 | 0.78 [0.76, 0.80] | 7.22 [6.39, 7.99] | 0.161 [0.154, 0.168] | 0.60 [0.54, 0.65] | 0.69 [0.65, 0.72] |
| 0.24 | 0.76 [0.75, 0.78] | 7.66 [6.98, 8.30] | 0.157 [0.150, 0.164] | 0.62 [0.57, 0.66] | 0.62 [0.59, 0.65] |
| 0.26 | 0.74 [0.73, 0.76] | 8.42 [7.91, 8.88] | 0.151 [0.144, 0.157] | 0.59 [0.55, 0.63] | 0.57 [0.55, 0.60] |
| 0.28 | 0.74 [0.73, 0.75] | 8.81 [8.33, 9.27] | 0.149 [0.142, 0.156] | 0.60 [0.57, 0.64] | 0.54 [0.52, 0.57] |
| 0.30 | 0.73 [0.72, 0.74] | 8.94 [8.47, 9.41] | 0.149 [0.142, 0.156] | 0.64 [0.60, 0.67] | 0.51 [0.49, 0.53] |

Table 2: Obverter dataset. Results for the metrics for selected noise levels. Shown in square brackets are bootstrapped 95%-confidence intervals. Here *topo* stands for topographic similarity, *conf* for conflict count, *cont* for context independence, *pos* for positional disentanglement and *acc* for accuracy.

## D.2 Different number of symbols

This section presents detailed results for the case when the message space has $16 = 4 \times 4$ elements (4-symbol alphabet; see Figure 15 and Table 3) and $64 = 8 \times 8$ elements (8-symbol alphabet; see Figure 16 and Table 4). In the former case, the results are more variable. For topo, this can be seen from a bimodal shape of its distribution and wide confidence intervals, particularly for small to medium values of noise. This makes it statistically hard to distinguish values of topo in this noise range. For larger values of noise (greater than $0.16$), topo starts to visibly decline. Similar behavior can be seen for other metrics.

Interestingly, for the latter experiment, with $64 = 8 \times 8$ messages, the story is different. While topographic similarity values for the small noise regime (up to $0.08$) do not improve over the baseline value, this behavior changes for medium to large values of noise. In this range, we can observe a visible increase in the topo, peaking at $0.88$ with a noise level of $0.24$.

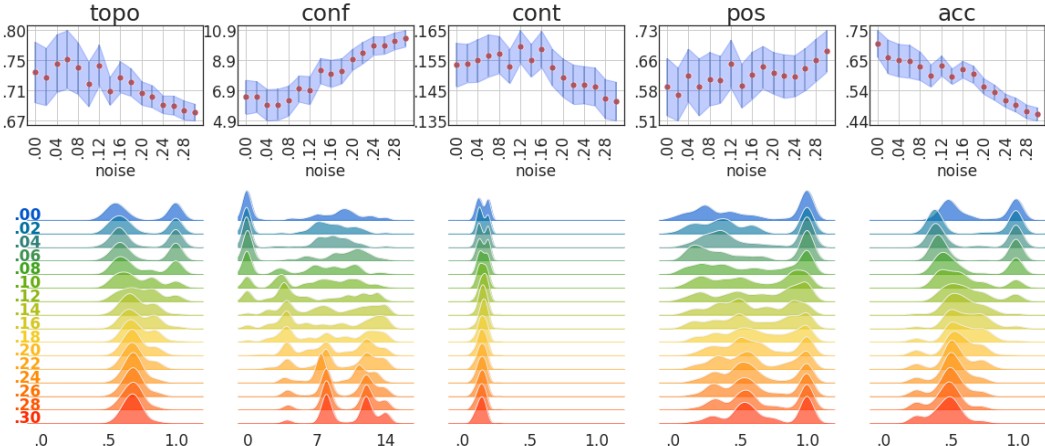

Figure 15: $4 \times 4$. Top panel: average value of metrics for various noise levels (on shapes3d dataset). The shaded area corresponds to bootstrapped $95\%$-confidence intervals for this estimator. Bottom panel: kernel density estimators for metrics and noise levels across seeds. Here *topo* stands for topographic similarity, *conf* for conflict count, *cont* for context independence, *pos* for positional disentanglement and *acc* for accuracy.

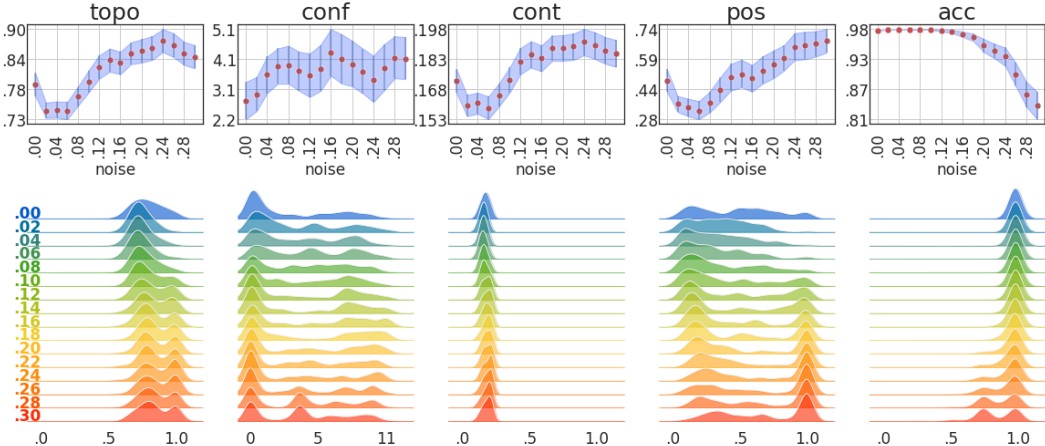

Figure 16: 8x8. Top panel: average value of metrics for various noise levels. The shaded area corresponds to bootstrapped $95\%$-confidence intervals for this estimator. Bottom panel: kernel density estimators for metrics and noise levels across seeds. Here *topo* stands for topographic similarity, *conf* for conflict count, *cont* for context independence, *pos* for positional disentanglement and *acc* for accuracy.

| noise | topo | conf | cont | pos | acc |
|-------|------|------|------|-----|-----|
| 0.00  | 0.74
[0.69, 0.78] | 6.45
[5.30, 7.57] | 0.154
[0.146, 0.161] | 0.59
[0.52, 0.67] | 0.71
[0.66, 0.75] |
| 0.02  | 0.73
[0.69, 0.77] | 6.48
[5.44, 7.48] | 0.154
[0.148, 0.160] | 0.57
[0.51, 0.64] | 0.66
[0.60, 0.72] |
| 0.04  | 0.75
[0.71, 0.79] | 5.93
[4.88, 6.96] | 0.155
[0.148, 0.162] | 0.62
[0.56, 0.69] | 0.65
[0.59, 0.71] |
| 0.06  | 0.76
[0.71, 0.80] | 5.94
[4.92, 6.99] | 0.156
[0.149, 0.163] | 0.59
[0.52, 0.66] | 0.65
[0.59, 0.70] |
| 0.08  | 0.74
[0.71, 0.78] | 6.19
[5.20, 7.19] | 0.157
[0.151, 0.163] | 0.61
[0.55, 0.68] | 0.63
[0.58, 0.68] |
| 0.10  | 0.72
[0.69, 0.75] | 7.00
[6.12, 7.87] | 0.153
[0.147, 0.159] | 0.61
[0.55, 0.67] | 0.60
[0.56, 0.63] |
| 0.12  | 0.75
[0.72, 0.78] | 6.90
[5.95, 7.82] | 0.159
[0.154, 0.165] | 0.65
[0.59, 0.71] | 0.63
[0.60, 0.67] |
| 0.14  | 0.71
[0.69, 0.73] | 8.20
[7.34, 9.02] | 0.155
[0.149, 0.160] | 0.60
[0.54, 0.65] | 0.59
[0.57, 0.62] |
| 0.16  | 0.73
[0.71, 0.75] | 8.01
[7.10, 8.91] | 0.159
[0.153, 0.165] | 0.62
[0.56, 0.68] | 0.62
[0.59, 0.65] |
| 0.18  | 0.72
[0.70, 0.74] | 8.18
[7.34, 9.01] | 0.153
[0.147, 0.159] | 0.64
[0.59, 0.70] | 0.60
[0.57, 0.63] |
| 0.20  | 0.71
[0.69, 0.72] | 8.93
[8.23, 9.63] | 0.149
[0.143, 0.155] | 0.63
[0.58, 0.68] | 0.56
[0.53, 0.59] |
| 0.22  | 0.70
[0.69, 0.72] | 9.38
[8.67, 10.09] | 0.147
[0.140, 0.153] | 0.62
[0.57, 0.67] | 0.54
[0.51, 0.56] |
| 0.24  | 0.69
[0.68, 0.70] | 9.86
[9.23, 10.46] | 0.147
[0.141, 0.153] | 0.62
[0.57, 0.67] | 0.51
[0.49, 0.53] |
| 0.26  | 0.69
[0.68, 0.70] | 9.87
[9.29, 10.45] | 0.146
[0.140, 0.153] | 0.64
[0.58, 0.69] | 0.49
[0.47, 0.52] |
| 0.28  | 0.68
[0.67, 0.70] | 10.15
[9.60, 10.72] | 0.142
[0.136, 0.149] | 0.66
[0.61, 0.71] | 0.47
[0.45, 0.49] |
| 0.30  | 0.68
[0.67, 0.69] | 10.32
[9.78, 10.86] | 0.141
[0.135, 0.148] | 0.68
[0.63, 0.73] | 0.46
[0.44, 0.48] |

Table 3: 4x4. Results for the metrics for selected noise levels. Shown in square brackets are bootstrapped 95%-confidence intervals. Here *topo* stands for topographic similarity, *conf* for conflict count, *cont* for context independence, *pos* for positional disentanglement and *acc* for accuracy.

| noise | topo | conf | cont | pos | acc |
|-------|------|------|------|-----|-----|
| 0.00 | 0.79 [0.77, 0.82] | 2.77 [2.17, 3.39] | 0.173 [0.166, 0.178] | 0.48 [0.42, 0.54] | 0.98 [0.98, 0.98] |
| 0.02 | 0.74 [0.73, 0.76] | 2.99 [2.43, 3.55] | 0.160 [0.155, 0.166] | 0.36 [0.32, 0.40] | 0.98 [0.98, 0.98] |
| 0.04 | 0.75 [0.73, 0.76] | 3.62 [3.02, 4.22] | 0.161 [0.156, 0.167] | 0.34 [0.30, 0.39] | 0.98 [0.98, 0.98] |
| 0.06 | 0.74 [0.73, 0.76] | 3.90 [3.33, 4.49] | 0.159 [0.153, 0.165] | 0.33 [0.28, 0.37] | 0.98 [0.98, 0.98] |
| 0.08 | 0.77 [0.75, 0.79] | 3.94 [3.33, 4.57] | 0.165 [0.160, 0.171] | 0.37 [0.32, 0.42] | 0.98 [0.98, 0.98] |
| 0.10 | 0.80 [0.78, 0.82] | 3.74 [3.10, 4.38] | 0.173 [0.167, 0.179] | 0.43 [0.37, 0.50] | 0.98 [0.98, 0.98] |
| 0.12 | 0.83 [0.80, 0.85] | 3.60 [2.90, 4.32] | 0.182 [0.176, 0.189] | 0.50 [0.43, 0.56] | 0.98 [0.98, 0.98] |
| 0.14 | 0.84 [0.82, 0.86] | 3.82 [3.09, 4.59] | 0.186 [0.179, 0.192] | 0.51 [0.44, 0.58] | 0.98 [0.98, 0.98] |
| 0.16 | 0.83 [0.81, 0.86] | 4.35 [3.59, 5.11] | 0.184 [0.177, 0.190] | 0.49 [0.42, 0.56] | 0.97 [0.97, 0.98] |
| 0.18 | 0.85 [0.83, 0.87] | 4.13 [3.34, 4.95] | 0.189 [0.182, 0.195] | 0.53 [0.46, 0.60] | 0.97 [0.96, 0.97] |
| 0.20 | 0.86 [0.84, 0.88] | 3.97 [3.24, 4.73] | 0.189 [0.182, 0.195] | 0.56 [0.49, 0.63] | 0.95 [0.94, 0.96] |
| 0.22 | 0.86 [0.84, 0.89] | 3.72 [3.01, 4.48] | 0.190 [0.183, 0.196] | 0.59 [0.53, 0.66] | 0.94 [0.93, 0.96] |
| 0.24 | 0.88 [0.85, 0.90] | 3.47 [2.73, 4.25] | 0.192 [0.185, 0.198] | 0.65 [0.58, 0.72] | 0.93 [0.91, 0.95] |
| 0.26 | 0.87 [0.85, 0.89] | 3.84 [3.11, 4.62] | 0.190 [0.183, 0.197] | 0.66 [0.59, 0.72] | 0.90 [0.87, 0.92] |
| 0.28 | 0.85 [0.83, 0.87] | 4.18 [3.51, 4.89] | 0.188 [0.181, 0.194] | 0.66 [0.60, 0.73] | 0.86 [0.83, 0.88] |
| 0.30 | 0.85 [0.82, 0.87] | 4.15 [3.48, 4.85] | 0.186 [0.179, 0.193] | 0.68 [0.62, 0.74] | 0.84 [0.81, 0.86] |

Table 4: 8x8. Results for the metrics for selected noise levels. Shown in square brackets are bootstrapped 95%-confidence intervals. Here *topo* stands for topographic similarity, *conf* for conflict count, *cont* for context independence, *pos* for positional disentanglement and *acc* for accuracy.

### D.3 Variable noise

In this section, we present detailed results for variable noise experiments. More precisely, at the beginning of training the noise is kept at some initial level $\epsilon_0 \in \{0.0, 0.15\}$ and after $T \in \{200, 700, 1200, 1700, 2200, 2700\}$ warmup network updates it is changed to a value, $\epsilon_T$, and kept there for the rest of the training. The results are presented in Figure 17 and Table 5 ($\epsilon_0 = 0.0, \epsilon_T = 0.15$), Figure 18 and Table 6 ($\epsilon_0 = 0.0, \epsilon_T = 0.1$), and Figure 19 and Table 7 ($\epsilon_0 = 0.15, \epsilon_T = 0.1$).

For $\epsilon_0 = 0.0, \epsilon_T = 0.15$ case, we see that topo increases from $0.85$ for $T = 200$, to $0.9$ for $T = 2700$, and the transition is reflected both in the density profile as well as the confidence intervals. The effect for $\epsilon_0 = 0.0, \epsilon_T = 0.1$ is weaker and the variance in the results is quite high. There seems to be a negligable effect for the case $\epsilon_0 = 0.15, \epsilon_T = 0.1$.

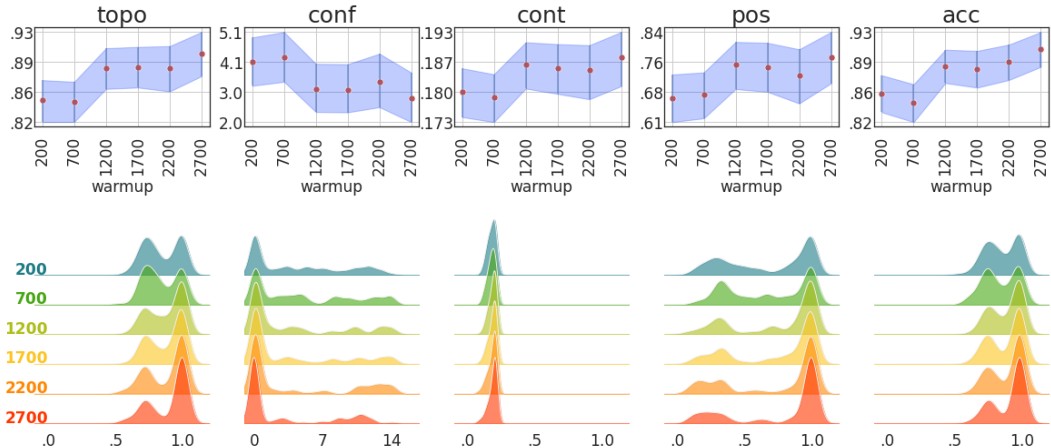

Figure 17: Variable noise with $\epsilon_0 = 0.0, \epsilon_T = 0.15$. Top panel: average value of metrics for various warmup levels ($T$). The shaded area corresponds to bootstrapped $95\%$-confidence intervals for this estimator. Bottom panel: kernel density estimators for metrics and noise levels across seeds. Here *topo* stands for topographic similarity, *conf* for conflict count, *cont* for context independence, *pos* for positional disentanglement and *acc* for accuracy.

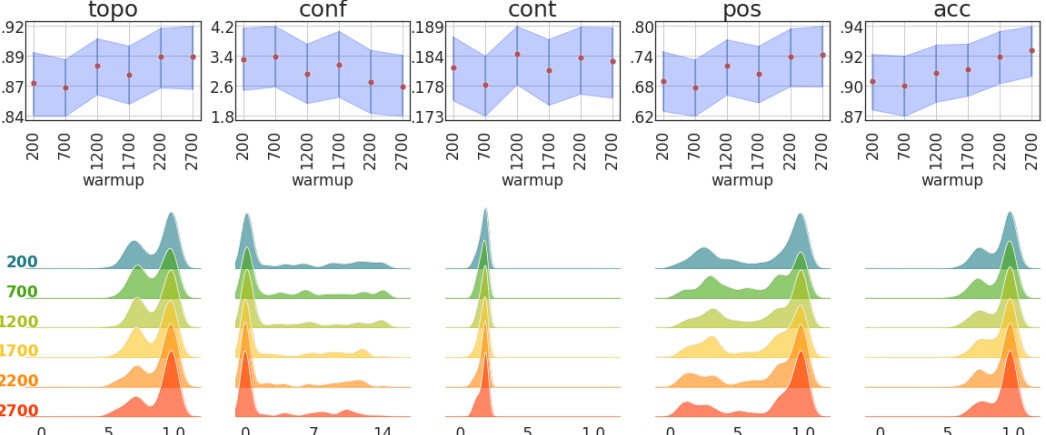

Figure 18: Variable noise $\epsilon_0 = 0.0, \epsilon_T = 0.1$. Top panel: average value of metrics for various warmup levels ($T$). The shaded area corresponds to bootstrapped $95\%$-confidence intervals for this estimator. Bottom panel: kernel density estimators for metrics and noise levels across seeds. Here *topo* stands for topographic similarity, *conf* for conflict count, *cont* for context independence, *pos* for positional disentanglement and *acc* for accuracy.

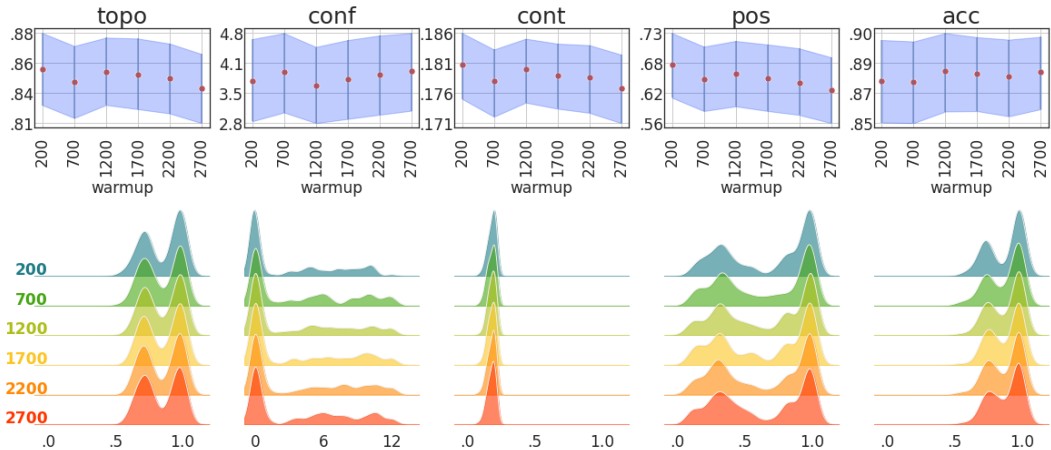

Figure 19: Variable noise $\epsilon_0 = 0.15, \epsilon_T = 0.1$. Top panel: average value of metrics for various warmup levels ($T$). The shaded area corresponds to bootstrapped 95%-confidence intervals for this estimator. Bottom panel: kernel density estimators for metrics and noise levels across seeds. Here *topo* stands for topographic similarity, *conf* for conflict count, *cont* for context independence, *pos* for positional disentanglement and *acc* for accuracy.

| warmup | topo | conf | cont | pos | acc |
|---|---|---|---|---|---|
| 200 | 0.85 [0.82, 0.87] | 4.06 [3.25, 4.89] | 0.180 [0.175, 0.185] | 0.67 [0.61, 0.73] | 0.85 [0.83, 0.88] |
| 700 | 0.85 [0.82, 0.87] | 4.21 [3.39, 5.07] | 0.179 [0.173, 0.184] | 0.68 [0.62, 0.73] | 0.84 [0.82, 0.87] |
| 1200 | 0.89 [0.86, 0.91] | 3.14 [2.37, 4.00] | 0.186 [0.181, 0.191] | 0.75 [0.69, 0.81] | 0.89 [0.87, 0.91] |
| 1700 | 0.89 [0.86, 0.91] | 3.12 [2.36, 3.99] | 0.185 [0.180, 0.190] | 0.75 [0.68, 0.81] | 0.88 [0.86, 0.91] |
| 2200 | 0.89 [0.86, 0.91] | 3.40 [2.53, 4.34] | 0.185 [0.178, 0.190] | 0.73 [0.65, 0.79] | 0.89 [0.87, 0.92] |
| 2700 | 0.90 [0.88, 0.93] | 2.82 [2.03, 3.70] | 0.188 [0.181, 0.193] | 0.77 [0.71, 0.84] | 0.91 [0.89, 0.93] |

Table 5: Variable noise with $\epsilon_0 = 0.0, \epsilon_T = 0.15$. Results for the metrics for various warmup levels ($T$). Shown in square brackets are bootstrapped 95%-confidence intervals. Here *topo* stands for topographic similarity, *conf* for conflict count, *cont* for context independence, *pos* for positional disentanglement and *acc* for accuracy.

| warmup | topo | conf | cont | pos | acc |
|---|---|---|---|---|---|
| 200 | 0.87 [0.84, 0.90] | 3.28 [2.47, 4.12] | 0.182 [0.176, 0.187] | 0.69 [0.63, 0.75] | 0.90 [0.88, 0.92] |
| 700 | 0.86 [0.84, 0.89] | 3.35 [2.56, 4.17] | 0.179 [0.173, 0.184] | 0.68 [0.62, 0.73] | 0.90 [0.87, 0.92] |
| 1200 | 0.88 [0.86, 0.91] | 2.89 [2.12, 3.70] | 0.184 [0.179, 0.189] | 0.72 [0.66, 0.78] | 0.91 [0.88, 0.93] |
| 1700 | 0.88 [0.85, 0.90] | 3.14 [2.28, 4.04] | 0.181 [0.175, 0.187] | 0.70 [0.65, 0.76] | 0.91 [0.89, 0.93] |
| 2200 | 0.89 [0.86, 0.92] | 2.69 [1.85, 3.54] | 0.183 [0.177, 0.189] | 0.74 [0.68, 0.80] | 0.92 [0.90, 0.94] |
| 2700 | 0.89 [0.86, 0.92] | 2.56 [1.78, 3.40] | 0.183 [0.176, 0.189] | 0.74 [0.68, 0.80] | 0.92 [0.90, 0.94] |

Table 6: Variable noise with $\epsilon_0 = 0.0, \epsilon_T = 0.1$. Results for the metrics for various warmup levels ($T$). Shown in square brackets are bootstrapped 95%-confidence intervals. Here *topo* stands for topographic similarity, *conf* for conflict count, *cont* for context independence, *pos* for positional disentanglement and *acc* for accuracy.

| warmup | topo | conf | cont | pos | acc |
|---|---|---|---|---|---|
| 200 | 0.86 [0.83, 0.88] | 3.73 [2.85, 4.63] | 0.181 [0.175, 0.186] | 0.67 [0.61, 0.73] | 0.88 [0.85, 0.90] |
| 700 | 0.85 [0.82, 0.87] | 3.91 [3.04, 4.76] | 0.178 [0.172, 0.183] | 0.65 [0.59, 0.71] | 0.88 [0.85, 0.90] |
| 1200 | 0.85 [0.83, 0.88] | 3.63 [2.81, 4.46] | 0.180 [0.175, 0.185] | 0.66 [0.59, 0.72] | 0.88 [0.86, 0.90] |
| 1700 | 0.85 [0.82, 0.88] | 3.77 [2.90, 4.61] | 0.179 [0.174, 0.184] | 0.65 [0.59, 0.71] | 0.88 [0.86, 0.90] |
| 2200 | 0.85 [0.82, 0.87] | 3.86 [2.99, 4.71] | 0.179 [0.173, 0.184] | 0.64 [0.58, 0.70] | 0.88 [0.85, 0.90] |
| 2700 | 0.84 [0.81, 0.87] | 3.94 [3.08, 4.76] | 0.177 [0.171, 0.183] | 0.62 [0.56, 0.69] | 0.88 [0.86, 0.90] |

Table 7: Variable noise with $\epsilon_0 = 0.15, \epsilon_T = 0.1$. Results for the metrics for various warmup levels ($T$). Shown in square brackets are bootstrapped 95%-confidence intervals. Here *topo* stands for topographic similarity, *conf* for conflict count, *cont* for context independence, *pos* for positional disentanglement and *acc* for accuracy.

## D.4 Sensitivity with respect to small architecture changes

This section provides details for experiments aiming to check the impact of small architecture change in CNN (Figure 20, Table 8) and the dense layers in the agents' network (Figure 21, Table 9). In the former experiment, we change the number of filters in the sender's CNN architecture from two layers with 8 filters, to two layers with 16 filters. This results in the slight change of topographic similarity profile with the highest average value of 0.86 for noise range 0.14 significantly outperforming the zero-noise case (0.79). For the second experiment, we added a dense layer (with 64 neurons) to the receiver's architecture. This again shows improvement of compositionality due to noise, although the noise in the range [0.08, 0.16] performs roughly the same.

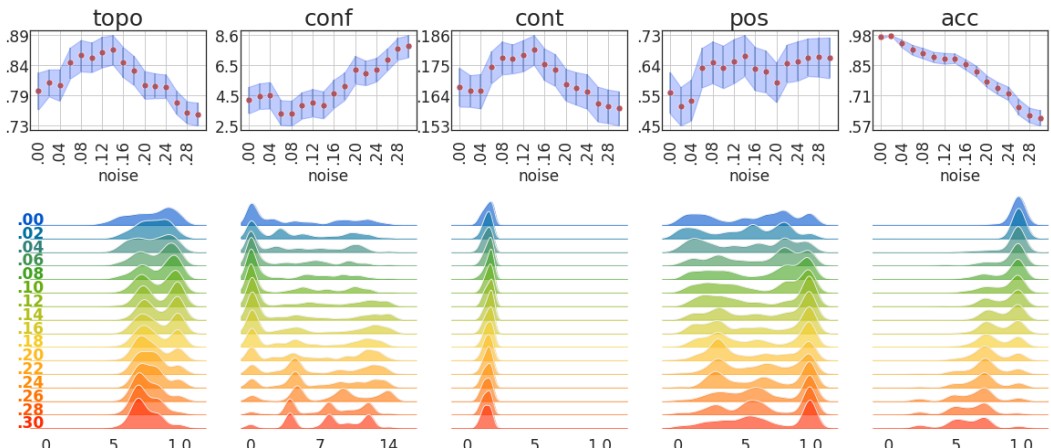

Figure 20: Bigger CNN. Top panel: average value of metrics for various noise levels. The shaded area corresponds to bootstrapped 95%-confidence intervals for this estimator. Bottom panel: kernel density estimators for metrics and noise levels across seeds. Here *topo* stands for topographic similarity, *conf* for conflict count, *cont* for context independence, *pos* for positional disentanglement and *acc* for accuracy.

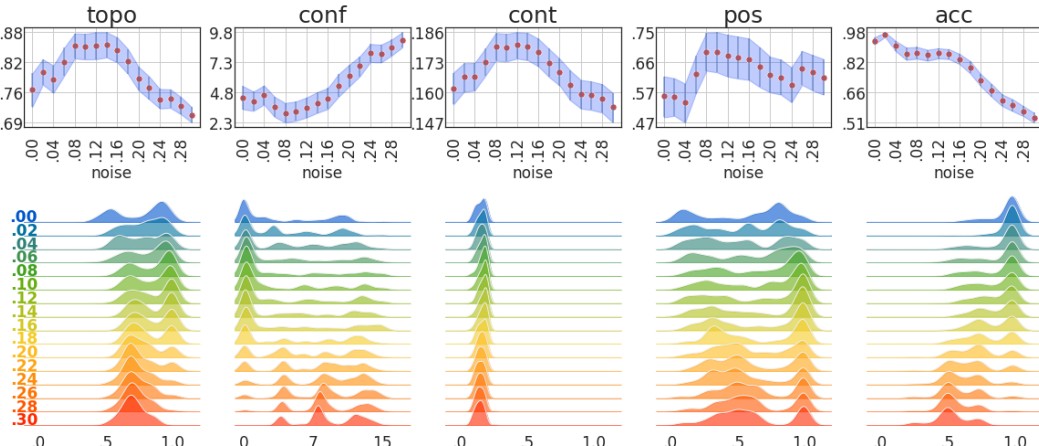

Figure 21: Bigger dense layer. Top panel: average value of metrics for various noise levels. The shaded area corresponds to bootstrapped 95%-confidence intervals for this estimator. Bottom panel: kernel density estimators for metrics and noise levels across seeds. Here *topo* stands for topographic similarity, *conf* for conflict count, *cont* for context independence, *pos* for positional disentanglement and *acc* for accuracy.

| noise | topo | conf | cont | pos | acc |
|---|---|---|---|---|---|
| 0.00 | 0.79 [0.76, 0.82] | 4.23 [3.34, 5.11] | 0.168 [0.160, 0.174] | 0.56 [0.49, 0.62] | 0.98 [0.97, 0.98] |
| 0.02 | 0.81 [0.78, 0.83] | 4.47 [3.63, 5.34] | 0.166 [0.160, 0.172] | 0.51 [0.45, 0.57] | 0.98 [0.98, 0.98] |
| 0.04 | 0.80 [0.78, 0.83] | 4.53 [3.66, 5.42] | 0.166 [0.160, 0.172] | 0.53 [0.47, 0.60] | 0.95 [0.93, 0.96] |
| 0.06 | 0.84 [0.81, 0.87] | 3.35 [2.56, 4.22] | 0.175 [0.169, 0.181] | 0.63 [0.57, 0.69] | 0.92 [0.89, 0.94] |
| 0.08 | 0.85 [0.83, 0.88] | 3.32 [2.52, 4.18] | 0.178 [0.172, 0.184] | 0.65 [0.58, 0.71] | 0.90 [0.88, 0.93] |
| 0.10 | 0.85 [0.83, 0.88] | 3.86 [2.99, 4.76] | 0.178 [0.172, 0.183] | 0.63 [0.57, 0.70] | 0.89 [0.86, 0.91] |
| 0.12 | 0.86 [0.83, 0.89] | 4.06 [3.13, 4.95] | 0.179 [0.173, 0.185] | 0.65 [0.58, 0.72] | 0.88 [0.85, 0.90] |
| 0.14 | 0.86 [0.84, 0.89] | 3.90 [3.00, 4.81] | 0.181 [0.176, 0.186] | 0.67 [0.61, 0.73] | 0.88 [0.86, 0.90] |
| 0.16 | 0.84 [0.82, 0.87] | 4.64 [3.70, 5.59] | 0.176 [0.170, 0.182] | 0.63 [0.56, 0.70] | 0.85 [0.83, 0.87] |
| 0.18 | 0.83 [0.80, 0.85] | 5.17 [4.22, 6.08] | 0.174 [0.168, 0.180] | 0.62 [0.56, 0.69] | 0.82 [0.80, 0.85] |
| 0.20 | 0.80 [0.78, 0.83] | 6.26 [5.32, 7.17] | 0.168 [0.162, 0.174] | 0.59 [0.53, 0.65] | 0.77 [0.75, 0.80] |
| 0.22 | 0.80 [0.78, 0.82] | 6.04 [5.21, 6.85] | 0.167 [0.161, 0.173] | 0.65 [0.59, 0.70] | 0.74 [0.72, 0.77] |
| 0.24 | 0.80 [0.78, 0.82] | 6.26 [5.46, 7.04] | 0.166 [0.160, 0.172] | 0.65 [0.59, 0.71] | 0.72 [0.69, 0.75] |
| 0.26 | 0.77 [0.75, 0.79] | 6.95 [6.19, 7.66] | 0.161 [0.155, 0.168] | 0.66 [0.61, 0.72] | 0.66 [0.62, 0.69] |
| 0.28 | 0.76 [0.74, 0.78] | 7.65 [6.91, 8.38] | 0.161 [0.154, 0.167] | 0.66 [0.60, 0.72] | 0.62 [0.59, 0.66] |
| 0.30 | 0.75 [0.73, 0.77] | 7.81 [7.08, 8.55] | 0.160 [0.153, 0.166] | 0.66 [0.60, 0.73] | 0.61 [0.57, 0.64] |

Table 8: Bigger CNN. Results for the metrics for selected noise levels. Shown in square brackets are bootstrapped 95%-confidence intervals. Here *topo* stands for topographic similarity, *conf* for conflict count, *cont* for context independence, *pos* for positional disentanglement and *acc* for accuracy.

| noise | topo | conf | cont | pos | acc |
|---|---|---|---|---|---|
| 0.00 | 0.76 [0.72, 0.80] | 4.42 [3.45, 5.38] | 0.162 [0.155, 0.169] | 0.56 [0.49, 0.62] | 0.93 [0.91, 0.95] |
| 0.02 | 0.80 [0.77, 0.83] | 4.10 [3.30, 4.92] | 0.167 [0.160, 0.173] | 0.55 [0.50, 0.61] | 0.96 [0.95, 0.98] |
| 0.04 | 0.78 [0.76, 0.81] | 4.65 [3.83, 5.42] | 0.167 [0.161, 0.173] | 0.54 [0.47, 0.59] | 0.90 [0.88, 0.93] |
| 0.06 | 0.82 [0.79, 0.85] | 3.62 [2.80, 4.48] | 0.173 [0.167, 0.179] | 0.62 [0.56, 0.68] | 0.86 [0.83, 0.90] |
| 0.08 | 0.86 [0.83, 0.88] | 3.14 [2.33, 3.97] | 0.180 [0.174, 0.185] | 0.69 [0.63, 0.75] | 0.87 [0.84, 0.90] |
| 0.10 | 0.85 [0.83, 0.88] | 3.27 [2.48, 4.09] | 0.179 [0.173, 0.185] | 0.69 [0.63, 0.75] | 0.86 [0.83, 0.88] |
| 0.12 | 0.86 [0.83, 0.88] | 3.56 [2.76, 4.42] | 0.180 [0.175, 0.186] | 0.68 [0.62, 0.74] | 0.87 [0.84, 0.89] |
| 0.14 | 0.86 [0.83, 0.88] | 3.95 [3.14, 4.87] | 0.180 [0.174, 0.186] | 0.67 [0.61, 0.74] | 0.86 [0.84, 0.89] |
| 0.16 | 0.85 [0.82, 0.87] | 4.31 [3.46, 5.21] | 0.177 [0.171, 0.183] | 0.67 [0.61, 0.73] | 0.84 [0.81, 0.86] |
| 0.18 | 0.82 [0.80, 0.85] | 5.42 [4.46, 6.36] | 0.173 [0.167, 0.179] | 0.65 [0.58, 0.71] | 0.79 [0.76, 0.83] |
| 0.20 | 0.79 [0.76, 0.81] | 6.25 [5.33, 7.14] | 0.169 [0.163, 0.175] | 0.62 [0.57, 0.68] | 0.73 [0.69, 0.76] |
| 0.22 | 0.77 [0.74, 0.79] | 7.04 [6.17, 7.85] | 0.163 [0.157, 0.170] | 0.61 [0.56, 0.67] | 0.67 [0.64, 0.71] |
| 0.24 | 0.74 [0.72, 0.76] | 8.14 [7.36, 8.87] | 0.159 [0.153, 0.166] | 0.59 [0.54, 0.64] | 0.62 [0.59, 0.65] |
| 0.26 | 0.74 [0.72, 0.76] | 8.08 [7.33, 8.82] | 0.159 [0.153, 0.165] | 0.64 [0.59, 0.69] | 0.60 [0.57, 0.63] |
| 0.28 | 0.73 [0.71, 0.75] | 8.58 [7.90, 9.24] | 0.158 [0.152, 0.164] | 0.63 [0.57, 0.68] | 0.57 [0.54, 0.59] |
| 0.30 | 0.71 [0.69, 0.73] | 9.21 [8.58, 9.85] | 0.154 [0.147, 0.160] | 0.61 [0.56, 0.67] | 0.53 [0.51, 0.56] |

Table 9: Bigger dense layer. Results for the metrics for selected noise levels. Shown in square brackets are bootstrapped 95%-confidence intervals. Here *topo* stands for topographic similarity, *conf* for conflict count, *cont* for context independence, *pos* for positional disentanglement and *acc* for accuracy.

## D.5 Sensitivity to noisy channel implementation

In this section, we provide details for experiments with alternative noisy channel implementation, see Figure 22 and Table 10. Notice that the range of noise values is different from the main experiment (smaller by the order of magnitude). The results suggest that the small values of noise can help, while the larger noise levels lead to a decline in compositionality.

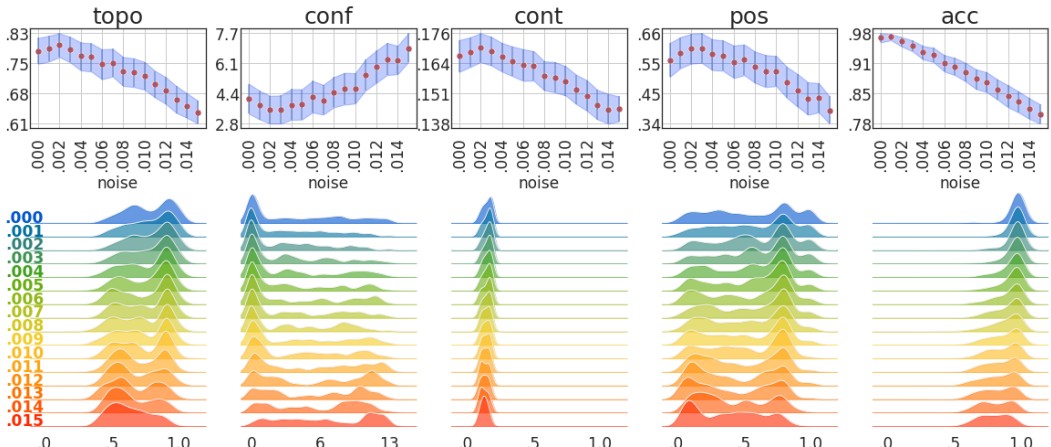

Figure 22: Alternative noise design. Top panel: average value of metrics for various noise levels. The shaded area corresponds to bootstrapped 95%-confidence intervals for this estimator. Bottom panel: kernel density estimators for metrics and noise levels across seeds. Here *topo* stands for topographic similarity, *conf* for conflict count, *cont* for context independence, *pos* for positional disentanglement and *acc* for accuracy.

| noise | topo | conf | cont | pos | acc |
|---|---|---|---|---|---|
| 0.000 | 0.79 [0.75, 0.82] | 4.13 [3.31, 4.95] | 0.167 [0.160, 0.173] | 0.56 [0.50, 0.62] | 0.97 [0.96, 0.98] |
| 0.001 | 0.79 [0.76, 0.82] | 3.77 [3.00, 4.57] | 0.168 [0.162, 0.174] | 0.59 [0.53, 0.64] | 0.97 [0.96, 0.98] |
| 0.002 | 0.80 [0.77, 0.83] | 3.51 [2.77, 4.29] | 0.170 [0.164, 0.176] | 0.60 [0.55, 0.66] | 0.96 [0.95, 0.97] |
| 0.003 | 0.79 [0.76, 0.82] | 3.52 [2.77, 4.31] | 0.169 [0.163, 0.175] | 0.61 [0.55, 0.66] | 0.95 [0.94, 0.96] |
| 0.004 | 0.77 [0.74, 0.81] | 3.80 [2.97, 4.66] | 0.166 [0.160, 0.172] | 0.58 [0.53, 0.64] | 0.94 [0.92, 0.95] |
| 0.005 | 0.77 [0.73, 0.80] | 3.81 [2.99, 4.65] | 0.165 [0.158, 0.171] | 0.58 [0.53, 0.63] | 0.93 [0.91, 0.95] |
| 0.006 | 0.75 [0.72, 0.79] | 4.21 [3.38, 5.11] | 0.163 [0.156, 0.170] | 0.56 [0.50, 0.61] | 0.91 [0.89, 0.93] |
| 0.007 | 0.76 [0.72, 0.79] | 4.05 [3.26, 4.89] | 0.163 [0.156, 0.169] | 0.56 [0.51, 0.62] | 0.91 [0.89, 0.92] |
| 0.008 | 0.74 [0.70, 0.77] | 4.48 [3.65, 5.31] | 0.158 [0.152, 0.164] | 0.54 [0.48, 0.59] | 0.89 [0.87, 0.91] |
| 0.009 | 0.73 [0.70, 0.77] | 4.68 [3.81, 5.55] | 0.158 [0.151, 0.164] | 0.52 [0.47, 0.58] | 0.88 [0.85, 0.90] |
| 0.010 | 0.72 [0.69, 0.76] | 4.70 [3.87, 5.53] | 0.156 [0.150, 0.162] | 0.52 [0.47, 0.58] | 0.87 [0.85, 0.89] |
| 0.011 | 0.70 [0.67, 0.74] | 5.43 [4.59, 6.30] | 0.153 [0.147, 0.159] | 0.48 [0.43, 0.54] | 0.85 [0.83, 0.88] |
| 0.012 | 0.69 [0.65, 0.72] | 5.87 [5.00, 6.76] | 0.150 [0.144, 0.157] | 0.46 [0.40, 0.51] | 0.84 [0.81, 0.86] |
| 0.013 | 0.67 [0.63, 0.70] | 6.29 [5.42, 7.19] | 0.146 [0.140, 0.152] | 0.43 [0.37, 0.48] | 0.83 [0.80, 0.85] |
| 0.014 | 0.65 [0.62, 0.68] | 6.23 [5.47, 7.03] | 0.144 [0.138, 0.151] | 0.43 [0.38, 0.48] | 0.81 [0.79, 0.83] |
| 0.015 | 0.63 [0.61, 0.66] | 6.91 [6.13, 7.72] | 0.145 [0.139, 0.150] | 0.39 [0.34, 0.44] | 0.80 [0.78, 0.82] |

Table 10: Alternative noise design. Results for the metrics for selected noise levels. Shown in square brackets are bootstrapped 95%-confidence intervals. Here *topo* stands for topographic similarity, *conf* for conflict count, *cont* for context independence, *pos* for positional disentanglement and *acc* for accuracy.

## D.6 Longer message

Here we provide details for longer message experiments, see Figure 23, Table 11 and Figure 24, Table 12 for message lengths three and four, respectively. The setup expands upon the main experiment by including floor color [11] for the former and, additionally, wall color [12] for the latter. The overall levels of compositionality metrics decline when compared with the main experiment, however, the general picture that noise improves compositionality remains intact.

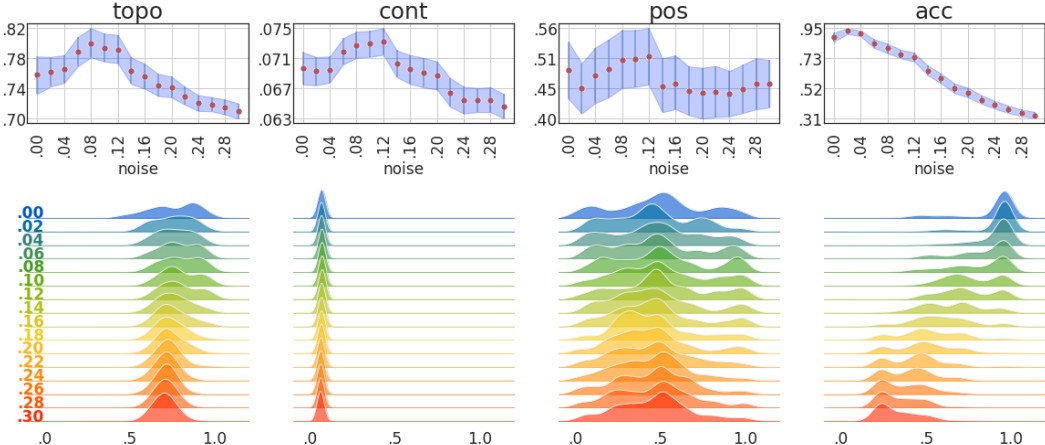

Figure 23: Message length equals 3. Top panel: average value of metrics for various noise levels. The shaded area corresponds to bootstrapped $95\%$-confidence intervals for this estimator. Bottom panel: kernel density estimators for metrics and noise levels across seeds. Here *topo* stands for topographic similarity, *pos* for positional disentanglement and *acc* for accuracy.

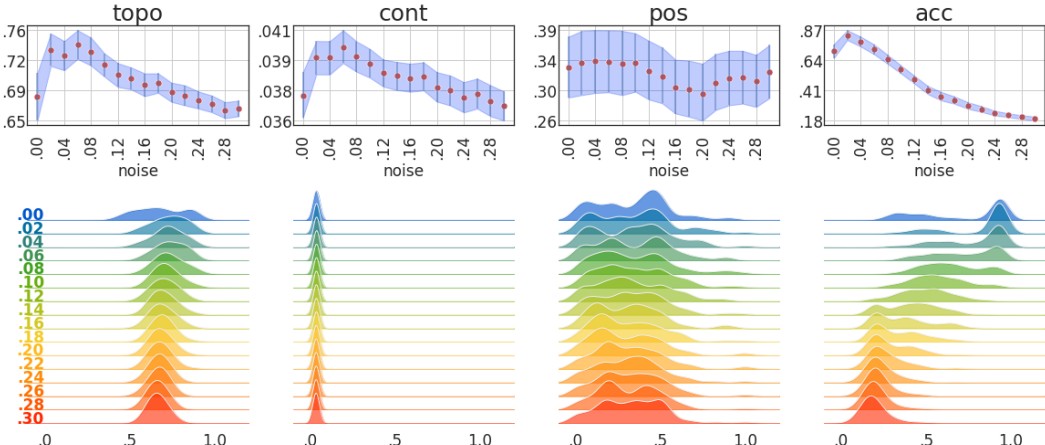

Figure 24: Message length equals 4. Top panel: average value of metrics for various noise levels. The shaded area corresponds to bootstrapped $95\%$-confidence intervals for this estimator. Bottom panel: kernel density estimators for metrics and noise levels across seeds. Here *topo* stands for topographic similarity, *pos* for positional disentanglement and *acc* for accuracy.

---

[11]We took the floor color feature to take values $0, 1, 2, 3$ (out of 10 possible). The resulting dataset contains 76800 images.

[12]We took the wall color feature to take values $0, 1, 2, 3$ (out of 10 possible). The resulting dataset contains 30720 images.

| noise | topo | cont | pos | acc |
|---|---|---|---|---|
| 0.00 | 0.76 [0.73, 0.78] | 0.069 [0.067, 0.072] | 0.49 [0.44, 0.54] | 0.89 [0.85, 0.92] |
| 0.02 | 0.76 [0.74, 0.78] | 0.069 [0.067, 0.071] | 0.45 [0.41, 0.50] | 0.93 [0.91, 0.95] |
| 0.04 | 0.77 [0.75, 0.79] | 0.069 [0.067, 0.071] | 0.48 [0.43, 0.53] | 0.91 [0.89, 0.93] |
| 0.06 | 0.79 [0.77, 0.81] | 0.072 [0.070, 0.074] | 0.49 [0.44, 0.54] | 0.84 [0.81, 0.87] |
| 0.08 | 0.80 [0.78, 0.82] | 0.073 [0.071, 0.074] | 0.51 [0.46, 0.56] | 0.81 [0.77, 0.84] |
| 0.10 | 0.80 [0.78, 0.82] | 0.073 [0.071, 0.075] | 0.51 [0.46, 0.56] | 0.76 [0.73, 0.79] |
| 0.12 | 0.79 [0.77, 0.81] | 0.073 [0.071, 0.075] | 0.51 [0.46, 0.56] | 0.74 [0.71, 0.78] |
| 0.14 | 0.76 [0.74, 0.78] | 0.070 [0.068, 0.072] | 0.46 [0.41, 0.51] | 0.65 [0.62, 0.68] |
| 0.16 | 0.76 [0.74, 0.77] | 0.069 [0.067, 0.071] | 0.46 [0.42, 0.51] | 0.59 [0.56, 0.63] |
| 0.18 | 0.74 [0.73, 0.76] | 0.069 [0.067, 0.071] | 0.45 [0.41, 0.49] | 0.52 [0.49, 0.56] |
| 0.20 | 0.74 [0.73, 0.76] | 0.069 [0.067, 0.070] | 0.45 [0.40, 0.49] | 0.49 [0.46, 0.53] |
| 0.22 | 0.73 [0.72, 0.74] | 0.066 [0.064, 0.068] | 0.45 [0.40, 0.50] | 0.44 [0.41, 0.47] |
| 0.24 | 0.72 [0.71, 0.73] | 0.065 [0.063, 0.067] | 0.45 [0.40, 0.49] | 0.41 [0.38, 0.43] |
| 0.26 | 0.72 [0.71, 0.73] | 0.065 [0.064, 0.067] | 0.45 [0.41, 0.50] | 0.37 [0.35, 0.40] |
| 0.28 | 0.71 [0.70, 0.72] | 0.065 [0.064, 0.067] | 0.46 [0.42, 0.51] | 0.35 [0.33, 0.37] |
| 0.30 | 0.71 [0.70, 0.72] | 0.064 [0.063, 0.066] | 0.46 [0.42, 0.51] | 0.33 [0.31, 0.36] |

Table 11: Message length equals 3. Results for the metrics for selected noise levels. Shown in square brackets are bootstrapped $95\%$-confidence intervals. Here *topo* stands for topographic similarity, *conf* for conflict count, *cont* for context independence, *pos* for positional disentanglement and *acc* for accuracy.

| noise | topo | cont | pos | acc |
|---|---|---|---|---|
| 0.00 | 0.68 [0.65, 0.71] | 0.038 [0.036, 0.039] | 0.33 [0.29, 0.38] | 0.71 [0.66, 0.76] |
| 0.02 | 0.74 [0.72, 0.76] | 0.040 [0.039, 0.040] | 0.34 [0.30, 0.39] | 0.83 [0.79, 0.87] |
| 0.04 | 0.73 [0.71, 0.75] | 0.040 [0.039, 0.040] | 0.34 [0.30, 0.39] | 0.78 [0.74, 0.82] |
| 0.06 | 0.74 [0.73, 0.76] | 0.040 [0.039, 0.041] | 0.34 [0.30, 0.39] | 0.72 [0.69, 0.76] |
| 0.08 | 0.73 [0.72, 0.75] | 0.040 [0.039, 0.040] | 0.34 [0.30, 0.39] | 0.65 [0.61, 0.68] |
| 0.10 | 0.72 [0.70, 0.73] | 0.039 [0.039, 0.040] | 0.34 [0.30, 0.38] | 0.57 [0.54, 0.60] |
| 0.12 | 0.71 [0.69, 0.72] | 0.039 [0.038, 0.039] | 0.33 [0.29, 0.37] | 0.49 [0.47, 0.52] |
| 0.14 | 0.70 [0.69, 0.72] | 0.039 [0.038, 0.039] | 0.32 [0.28, 0.36] | 0.41 [0.38, 0.45] |
| 0.16 | 0.69 [0.68, 0.71] | 0.038 [0.038, 0.039] | 0.31 [0.27, 0.35] | 0.37 [0.34, 0.40] |
| 0.18 | 0.70 [0.69, 0.71] | 0.039 [0.038, 0.039] | 0.30 [0.27, 0.34] | 0.34 [0.31, 0.37] |
| 0.20 | 0.69 [0.67, 0.70] | 0.038 [0.037, 0.039] | 0.30 [0.26, 0.34] | 0.30 [0.27, 0.32] |
| 0.22 | 0.68 [0.67, 0.69] | 0.038 [0.037, 0.039] | 0.31 [0.28, 0.35] | 0.27 [0.25, 0.29] |
| 0.24 | 0.68 [0.67, 0.69] | 0.038 [0.037, 0.038] | 0.32 [0.28, 0.36] | 0.24 [0.22, 0.26] |
| 0.26 | 0.67 [0.66, 0.68] | 0.038 [0.037, 0.038] | 0.32 [0.28, 0.36] | 0.22 [0.21, 0.24] |
| 0.28 | 0.66 [0.66, 0.67] | 0.037 [0.037, 0.038] | 0.32 [0.28, 0.35] | 0.21 [0.20, 0.23] |
| 0.30 | 0.67 [0.66, 0.68] | 0.037 [0.036, 0.038] | 0.33 [0.29, 0.37] | 0.20 [0.18, 0.21] |

Table 12: Message length equals 4. Results for the metrics for selected noise levels. Shown in square brackets are bootstrapped $95\%$-confidence intervals. Here *topo* stands for topographic similarity, *conf* for conflict count, *cont* for context independence, *pos* for positional disentanglement and *acc* for accuracy.

## D.7   Visual priors

In this section we give details for visual priors experiments, see Figure 25, Table 13 (tile 32), Figure 26, Table 14 (tile 16), and Figure 27, Table 15 (tile 8). The transition from coarser to finer tiles has a significant impact both on the metric's profiles and on their overall levels. The characteristic peak for some positive noise levels is still present.

We complement the picture with an analysis of the interplay between variable noise and accuracy, see Figure 28 (tile 32) and Figure 29 (tile 16). It shows that the overall metrics level, conditioned on seeds with high accuracy, increases significantly. Notice, however, that the number of experiments with high accuracy decrease as the threshold increases. In particular, we did not include visualization for a tile size 8, since there were too few experiments exceeding the accuracy threshold of $0.80$.

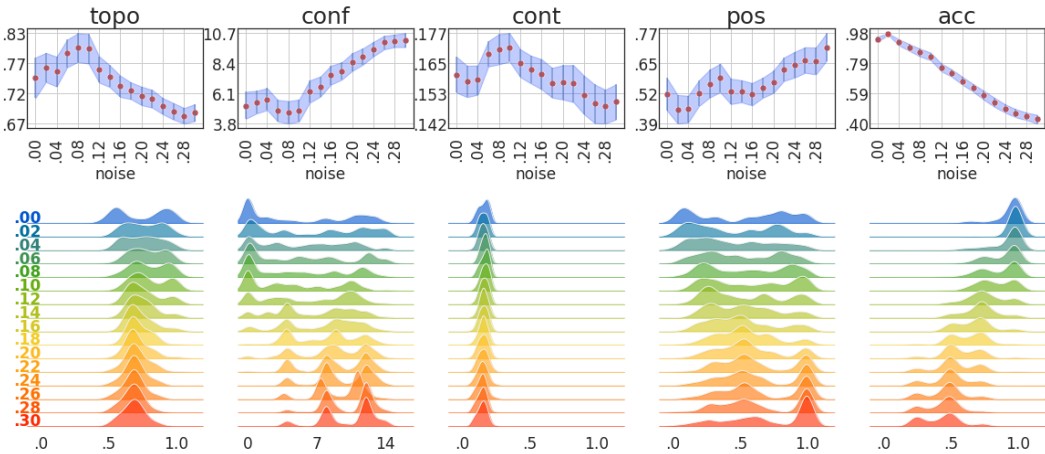

Figure 25: Scramble with tile 32. Top panel: average value of metrics for various noise levels. The shaded area corresponds to bootstrapped $95\%$-confidence intervals for this estimator. Bottom panel: kernel density estimators for metrics and noise levels across seeds. Here *topo* stands for topographic similarity, *conf* for conflict count, *cont* for context independence, *pos* for positional disentanglement and *acc* for accuracy.

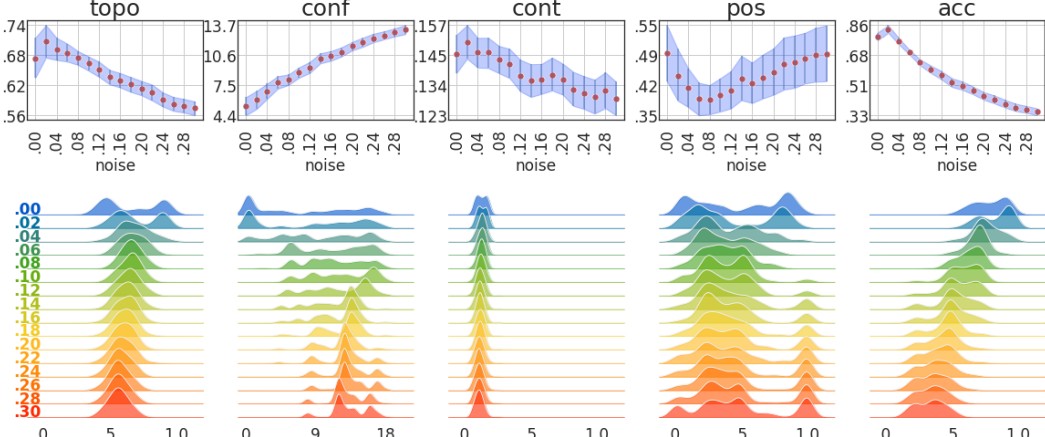

Figure 26: Scramble with tile 16. Top panel: average value of metrics for various noise levels. The shaded area corresponds to bootstrapped $95\%$-confidence intervals for this estimator. Bottom panel: kernel density estimators for metrics and noise levels across seeds. Here *topo* stands for topographic similarity, *conf* for conflict count, *cont* for context independence, *pos* for positional disentanglement and *acc* for accuracy.

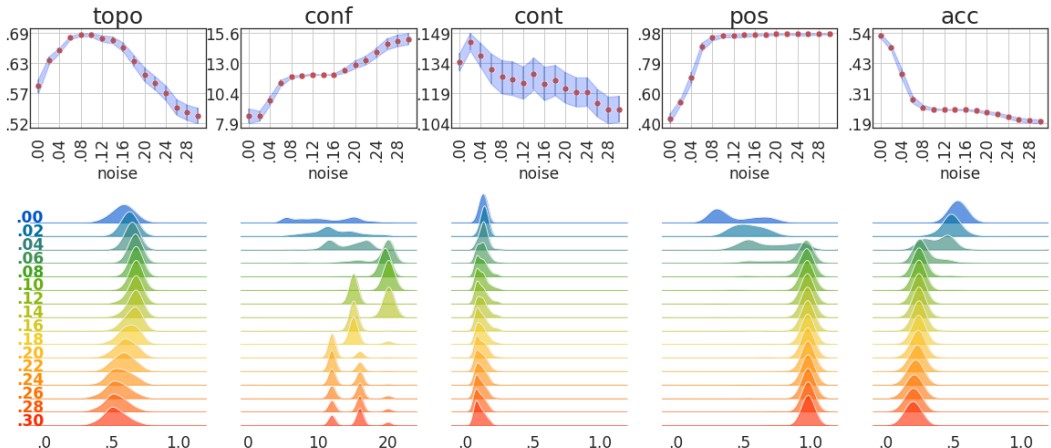

Figure 27: Scramble with tile 8. Top panel: average value of metrics for various noise levels. The shaded area corresponds to bootstrapped 95%-confidence intervals for this estimator. Bottom panel: kernel density estimators for metrics and noise levels across seeds. Here *topo* stands for topographic similarity, *conf* for conflict count, *cont* for context independence, *pos* for positional disentanglement and *acc* for accuracy.

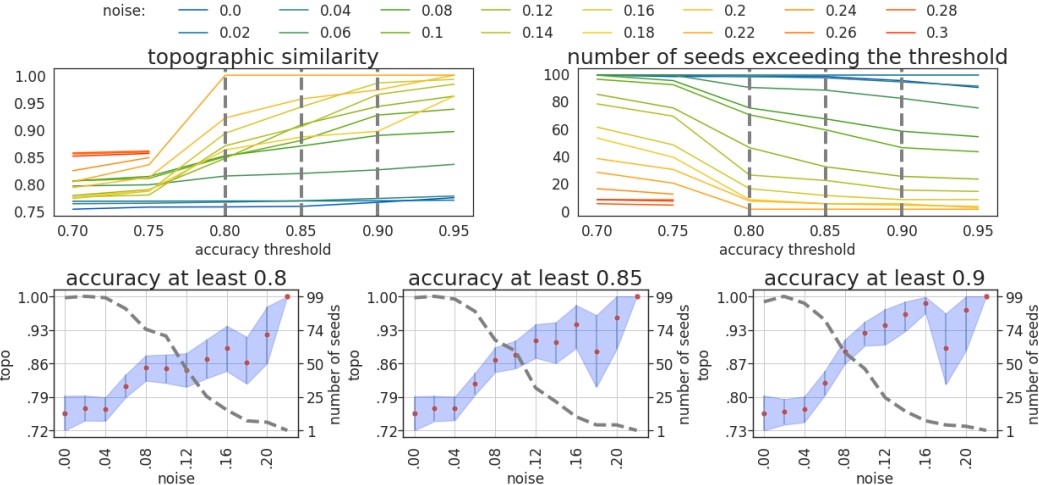

Figure 28: Tile size 32. Top left: The values of topo computed for each noise level (hues) and on seeds exceeding a certain accuracy threshold ($x$-axis). Vertical dashed lines represent three cross-sections, visualized in the bottom panel. Top right: Similar to the left panel, but instead of topo we visualize the number of seeds with accuracy at least as a given threshold ($x$-axis). Vertical dashed lines represent three cross-sections, visualized in the bottom panel. Bottom: Each of the plots represents a cross-section of the plots in the top panel, taken at points 0.80, 0.85, and 0.90, respectively. On the left axis of each figure is the range of topo, whereas on the right axis is the number of seeds with accuracy exceed the corresponding level. On the $x$-axis are the noise levels. The scatter plot with 95%-confidence intervals represents the values of topo. The gray dashed line represents the number of seeds with accuracy exceeding a given threshold, for each of the noise levels.

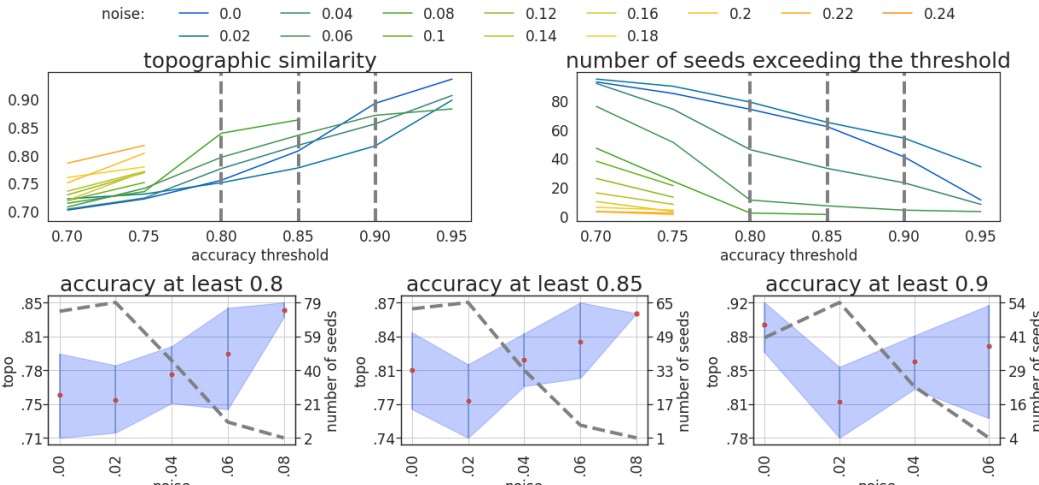

Figure 29: Tile size 16. Top left: The values of topo computed for each noise level (hues) and on seeds exceeding a certain accuracy threshold ($x$-axis). Vertical dashed lines represent three cross-sections, visualized in the bottom panel. Top right: Similar to the left panel, but instead of topo we visualize the number of seeds with accuracy at least as a given threshold ($x$-axis). Vertical dashed lines represent three cross-sections, visualized in the bottom panel. Bottom: Each of the plots represents a cross-section of the plots in the top panel, taken at points 0.80, 0.85, and 0.90, respectively. On the left axis of each figure is the range of topo, whereas on the right axis is the number of seeds with accuracy exceed the corresponding level. On the $x$-axis are the noise levels. The scatter plot with 95%-confidence intervals represents the values of topo. The gray dashed line represents the number of seeds with accuracy exceeding a given threshold, for each of the noise levels.

| noise | topo | conf | cont | pos | acc |
|---|---|---|---|---|---|
| 0.00 | 0.75 [0.71, 0.78] | 5.20 [4.21, 6.27] | 0.160 [0.154, 0.168] | 0.52 [0.45, 0.59] | 0.95 [0.93, 0.96] |
| 0.02 | 0.77 [0.74, 0.79] | 5.47 [4.61, 6.35] | 0.158 [0.152, 0.164] | 0.45 [0.39, 0.51] | 0.98 [0.97, 0.98] |
| 0.04 | 0.76 [0.73, 0.79] | 5.69 [4.87, 6.55] | 0.158 [0.153, 0.164] | 0.45 [0.40, 0.51] | 0.92 [0.90, 0.94] |
| 0.06 | 0.79 [0.77, 0.82] | 4.81 [4.01, 5.66] | 0.169 [0.164, 0.174] | 0.52 [0.47, 0.58] | 0.89 [0.87, 0.91] |
| 0.08 | 0.80 [0.78, 0.83] | 4.68 [3.84, 5.57] | 0.170 [0.165, 0.176] | 0.56 [0.50, 0.62] | 0.86 [0.83, 0.88] |
| 0.10 | 0.80 [0.77, 0.83] | 4.82 [4.00, 5.68] | 0.171 [0.166, 0.177] | 0.59 [0.53, 0.65] | 0.83 [0.81, 0.86] |
| 0.12 | 0.76 [0.74, 0.79] | 6.29 [5.44, 7.14] | 0.165 [0.160, 0.170] | 0.53 [0.48, 0.58] | 0.76 [0.74, 0.79] |
| 0.14 | 0.75 [0.73, 0.77] | 6.68 [5.90, 7.44] | 0.163 [0.157, 0.168] | 0.53 [0.48, 0.58] | 0.72 [0.70, 0.75] |
| 0.16 | 0.73 [0.72, 0.75] | 7.57 [6.82, 8.27] | 0.161 [0.155, 0.167] | 0.52 [0.47, 0.56] | 0.67 [0.65, 0.70] |
| 0.18 | 0.73 [0.71, 0.74] | 7.86 [7.19, 8.51] | 0.157 [0.152, 0.163] | 0.54 [0.50, 0.59] | 0.63 [0.60, 0.65] |
| 0.20 | 0.72 [0.70, 0.73] | 8.57 [7.87, 9.22] | 0.157 [0.151, 0.164] | 0.57 [0.52, 0.62] | 0.58 [0.55, 0.61] |
| 0.22 | 0.71 [0.70, 0.73] | 8.98 [8.34, 9.59] | 0.157 [0.150, 0.164] | 0.62 [0.57, 0.67] | 0.54 [0.51, 0.57] |
| 0.24 | 0.70 [0.69, 0.71] | 9.54 [8.98, 10.09] | 0.153 [0.145, 0.160] | 0.64 [0.59, 0.70] | 0.50 [0.47, 0.52] |
| 0.26 | 0.69 [0.68, 0.70] | 10.04 [9.51, 10.54] | 0.149 [0.142, 0.157] | 0.66 [0.61, 0.72] | 0.47 [0.44, 0.49] |
| 0.28 | 0.68 [0.67, 0.70] | 10.18 [9.69, 10.65] | 0.148 [0.142, 0.155] | 0.66 [0.60, 0.71] | 0.45 [0.42, 0.47] |
| 0.30 | 0.69 [0.67, 0.70] | 10.23 [9.68, 10.75] | 0.150 [0.143, 0.157] | 0.72 [0.66, 0.77] | 0.43 [0.40, 0.46] |

Table 13: Scramble with tile 32. Results for the metrics for selected noise levels. Shown in square brackets are bootstrapped 95%-confidence intervals. Here *topo* stands for topographic similarity, *conf* for conflict count, *cont* for context independence, *pos* for positional disentanglement and *acc* for accuracy.

| noise | topo | conf | cont | pos | acc |
|-------|------|------|------|-----|-----|
| 0.00 | 0.67 [0.64, 0.71] | 5.32 [4.36, 6.26] | 0.146 [0.139, 0.153] | 0.49 [0.43, 0.55] | 0.79 [0.77, 0.82] |
| 0.02 | 0.71 [0.67, 0.74] | 5.97 [5.01, 6.89] | 0.150 [0.144, 0.157] | 0.44 [0.39, 0.50] | 0.84 [0.81, 0.86] |
| 0.04 | 0.69 [0.67, 0.71] | 6.87 [6.07, 7.66] | 0.147 [0.141, 0.152] | 0.41 [0.37, 0.46] | 0.77 [0.75, 0.79] |
| 0.06 | 0.68 [0.67, 0.70] | 7.79 [7.13, 8.43] | 0.147 [0.141, 0.152] | 0.39 [0.35, 0.43] | 0.70 [0.69, 0.72] |
| 0.08 | 0.67 [0.66, 0.69] | 8.05 [7.50, 8.60] | 0.144 [0.139, 0.149] | 0.39 [0.36, 0.42] | 0.64 [0.62, 0.66] |
| 0.10 | 0.66 [0.65, 0.68] | 8.81 [8.27, 9.35] | 0.142 [0.137, 0.148] | 0.40 [0.36, 0.44] | 0.60 [0.58, 0.62] |
| 0.12 | 0.65 [0.64, 0.67] | 9.34 [8.75, 9.91] | 0.138 [0.132, 0.144] | 0.41 [0.37, 0.45] | 0.57 [0.54, 0.59] |
| 0.14 | 0.64 [0.62, 0.65] | 10.30 [9.76, 10.82] | 0.136 [0.130, 0.142] | 0.43 [0.39, 0.49] | 0.52 [0.50, 0.55] |
| 0.16 | 0.63 [0.62, 0.64] | 10.54 [10.03, 11.04] | 0.137 [0.131, 0.142] | 0.42 [0.38, 0.47] | 0.50 [0.48, 0.53] |
| 0.18 | 0.62 [0.61, 0.64] | 10.95 [10.47, 11.43] | 0.138 [0.132, 0.145] | 0.44 [0.39, 0.49] | 0.48 [0.45, 0.50] |
| 0.20 | 0.62 [0.60, 0.63] | 11.55 [11.10, 11.99] | 0.136 [0.131, 0.142] | 0.45 [0.40, 0.50] | 0.44 [0.42, 0.47] |
| 0.22 | 0.61 [0.59, 0.62] | 11.95 [11.47, 12.43] | 0.133 [0.126, 0.139] | 0.47 [0.41, 0.53] | 0.42 [0.40, 0.45] |
| 0.24 | 0.59 [0.58, 0.61] | 12.38 [11.94, 12.84] | 0.131 [0.125, 0.138] | 0.47 [0.41, 0.53] | 0.40 [0.38, 0.42] |
| 0.26 | 0.58 [0.57, 0.60] | 12.67 [12.26, 13.09] | 0.130 [0.124, 0.136] | 0.48 [0.42, 0.54] | 0.38 [0.36, 0.40] |
| 0.28 | 0.58 [0.57, 0.59] | 12.97 [12.55, 13.41] | 0.132 [0.126, 0.139] | 0.49 [0.43, 0.55] | 0.36 [0.34, 0.39] |
| 0.30 | 0.58 [0.56, 0.59] | 13.25 [12.79, 13.73] | 0.129 [0.123, 0.136] | 0.49 [0.43, 0.55] | 0.35 [0.33, 0.37] |

Table 14: Scramble with tile 16. Results for the metrics for selected noise levels. Shown in square brackets are bootstrapped 95%-confidence intervals. Here *topo* stands for topographic similarity, *conf* for conflict count, *cont* for context independence, *pos* for positional disentanglement and *acc* for accuracy.

| noise | topo | conf | cont | pos | acc |
|---|---|---|---|---|---|
| 0.00 | 0.59 [0.57, 0.60] | 8.51 [7.86, 9.13] | 0.134 [0.130, 0.139] | 0.44 [0.40, 0.47] | 0.53 [0.52, 0.54] |
| 0.02 | 0.64 [0.63, 0.64] | 8.48 [8.07, 8.92] | 0.144 [0.140, 0.149] | 0.54 [0.52, 0.56] | 0.49 [0.48, 0.50] |
| 0.04 | 0.65 [0.65, 0.66] | 9.82 [9.46, 10.18] | 0.138 [0.132, 0.144] | 0.70 [0.66, 0.73] | 0.38 [0.37, 0.40] |
| 0.06 | 0.68 [0.67, 0.68] | 11.33 [11.04, 11.58] | 0.130 [0.123, 0.138] | 0.90 [0.87, 0.92] | 0.29 [0.27, 0.30] |
| 0.08 | 0.68 [0.68, 0.69] | 11.86 [11.73, 11.96] | 0.127 [0.119, 0.135] | 0.96 [0.94, 0.97] | 0.26 [0.25, 0.27] |
| 0.10 | 0.68 [0.68, 0.69] | 11.95 [11.87, 12.00] | 0.126 [0.118, 0.135] | 0.97 [0.96, 0.98] | 0.25 [0.24, 0.25] |
| 0.12 | 0.68 [0.67, 0.68] | 12.04 [12.00, 12.12] | 0.124 [0.116, 0.132] | 0.97 [0.96, 0.98] | 0.25 [0.24, 0.25] |
| 0.14 | 0.67 [0.67, 0.68] | 12.00 [12.00, 12.00] | 0.128 [0.121, 0.136] | 0.97 [0.96, 0.98] | 0.25 [0.24, 0.25] |
| 0.16 | 0.66 [0.65, 0.67] | 12.03 [11.98, 12.12] | 0.123 [0.115, 0.132] | 0.98 [0.97, 0.98] | 0.25 [0.24, 0.25] |
| 0.18 | 0.63 [0.62, 0.65] | 12.36 [12.16, 12.60] | 0.125 [0.118, 0.133] | 0.98 [0.97, 0.98] | 0.24 [0.24, 0.25] |
| 0.20 | 0.61 [0.60, 0.62] | 12.88 [12.56, 13.24] | 0.121 [0.114, 0.129] | 0.98 [0.97, 0.98] | 0.24 [0.23, 0.24] |
| 0.22 | 0.59 [0.58, 0.61] | 13.24 [12.88, 13.64] | 0.119 [0.112, 0.127] | 0.98 [0.98, 0.98] | 0.23 [0.22, 0.24] |
| 0.24 | 0.57 [0.56, 0.59] | 13.93 [13.52, 14.34] | 0.120 [0.113, 0.127] | 0.98 [0.97, 0.98] | 0.22 [0.21, 0.23] |
| 0.26 | 0.55 [0.53, 0.56] | 14.65 [14.16, 15.13] | 0.114 [0.107, 0.121] | 0.98 [0.97, 0.98] | 0.21 [0.20, 0.22] |
| 0.28 | 0.54 [0.52, 0.55] | 14.92 [14.44, 15.44] | 0.111 [0.104, 0.117] | 0.98 [0.98, 0.98] | 0.20 [0.20, 0.21] |
| 0.30 | 0.53 [0.52, 0.55] | 15.08 [14.60, 15.56] | 0.111 [0.105, 0.118] | 0.98 [0.98, 0.98] | 0.20 [0.19, 0.21] |

Table 15: Scramble with tile 8. Results for the metrics for selected noise levels. Shown in square brackets are bootstrapped 95%-confidence intervals. Here *topo* stands for topographic similarity, *conf* for conflict count, *cont* for context independence, *pos* for positional disentanglement and *acc* for accuracy.

## D.8 Scrambled features

In these experiments, we aimed to understand if the architecture of the output is important. Our architecture has a separate output for 'color' ($\mathcal{F}_1$) and for 'shape' ($\mathcal{F}_2$) (for the architecture details, see Figure 8). We permute these features as follows. Let $k : \mathcal{F}_1 \times \mathcal{F}_2 \mapsto \{0, \dots, |\mathcal{F}_1 \times \mathcal{F}_2| - 1\} =: \tilde{\mathcal{F}}$ be any bijection from the color features and the shape features. Let $\pi : \tilde{\mathcal{F}} \mapsto \tilde{\mathcal{F}}$ be a random permutation, sampled at the beginning of the experiment. We assign new features via mapping: $k^{-1}(\pi(k((f_c, f_s))))$. Clearly, these are no longer colors and shapes (unless $\pi$ is an identity). However, they are still factorized along two dimensions and we still use the factorized output. We can see that behavior is similar to the 'standard' features (high levels of compositionality with the characteristic extremum point). This poses strong evidence that output architecture is a strong inductive bias for compositionality.

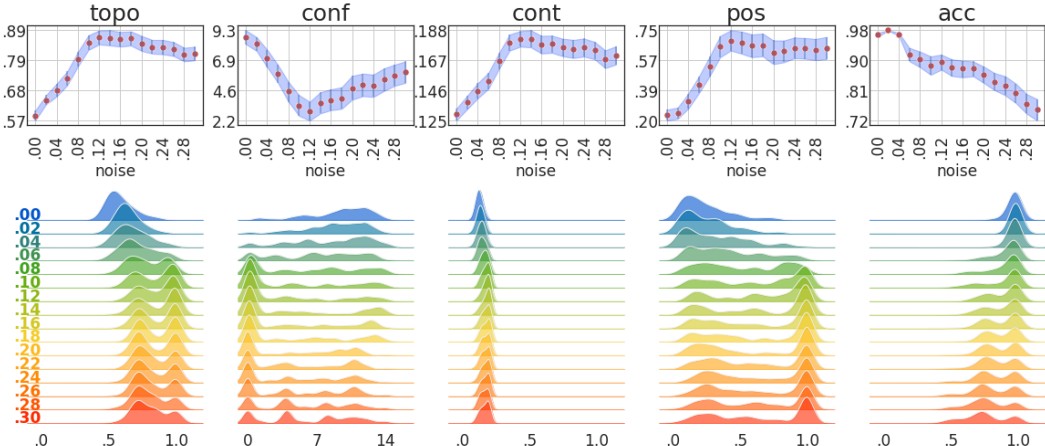

Figure 30: Scrambled features. Top panel: average value of metrics for various noise levels. The shaded area corresponds to bootstrapped 95%-confidence intervals for this estimator. Bottom panel: kernel density estimators for metrics and noise levels across seeds. Here *topo* stands for topographic similarity, *conf* for conflict count, *cont* for context independence, *pos* for positional disentanglement and *acc* for accuracy.

| noise | topo | conf | cont | pos | acc |
|-------|------|------|------|-----|-----|
| 0.00 | 0.59 [0.57, 0.61] | 8.72 [8.17, 9.26] | 0.129 [0.125, 0.133] | 0.24 [0.20, 0.27] | 0.97 [0.96, 0.98] |
| 0.02 | 0.64 [0.63, 0.66] | 8.21 [7.65, 8.73] | 0.138 [0.133, 0.142] | 0.25 [0.21, 0.28] | 0.98 [0.98, 0.98] |
| 0.04 | 0.68 [0.66, 0.70] | 7.07 [6.39, 7.75] | 0.145 [0.140, 0.151] | 0.32 [0.28, 0.37] | 0.97 [0.97, 0.98] |
| 0.06 | 0.72 [0.70, 0.75] | 5.88 [5.18, 6.58] | 0.153 [0.147, 0.158] | 0.42 [0.37, 0.47] | 0.91 [0.89, 0.93] |
| 0.08 | 0.79 [0.76, 0.82] | 4.48 [3.69, 5.24] | 0.166 [0.161, 0.172] | 0.53 [0.47, 0.59] | 0.90 [0.88, 0.92] |
| 0.10 | 0.85 [0.82, 0.88] | 3.40 [2.58, 4.21] | 0.179 [0.173, 0.185] | 0.65 [0.59, 0.72] | 0.88 [0.85, 0.91] |
| 0.12 | 0.87 [0.84, 0.89] | 2.93 [2.21, 3.69] | 0.182 [0.176, 0.187] | 0.69 [0.63, 0.75] | 0.89 [0.87, 0.91] |
| 0.14 | 0.87 [0.84, 0.89] | 3.57 [2.77, 4.42] | 0.182 [0.176, 0.188] | 0.68 [0.61, 0.74] | 0.88 [0.85, 0.90] |
| 0.16 | 0.86 [0.83, 0.89] | 3.81 [3.02, 4.65] | 0.178 [0.171, 0.184] | 0.66 [0.59, 0.73] | 0.87 [0.85, 0.90] |
| 0.18 | 0.86 [0.84, 0.89] | 3.95 [3.13, 4.82] | 0.178 [0.172, 0.184] | 0.66 [0.59, 0.73] | 0.87 [0.85, 0.89] |
| 0.20 | 0.85 [0.82, 0.87] | 4.75 [3.84, 5.71] | 0.176 [0.170, 0.182] | 0.62 [0.55, 0.69] | 0.85 [0.83, 0.88] |
| 0.22 | 0.83 [0.81, 0.86] | 4.99 [4.12, 5.88] | 0.175 [0.168, 0.181] | 0.62 [0.56, 0.69] | 0.83 [0.81, 0.86] |
| 0.24 | 0.83 [0.81, 0.86] | 4.92 [4.06, 5.81] | 0.176 [0.169, 0.182] | 0.64 [0.58, 0.71] | 0.82 [0.80, 0.85] |
| 0.26 | 0.83 [0.80, 0.85] | 5.41 [4.53, 6.30] | 0.174 [0.167, 0.180] | 0.64 [0.58, 0.70] | 0.80 [0.77, 0.83] |
| 0.28 | 0.81 [0.78, 0.83] | 5.73 [4.93, 6.58] | 0.168 [0.161, 0.174] | 0.63 [0.57, 0.70] | 0.77 [0.74, 0.80] |
| 0.30 | 0.81 [0.79, 0.83] | 5.98 [5.15, 6.83] | 0.170 [0.164, 0.176] | 0.64 [0.58, 0.71] | 0.75 [0.72, 0.78] |

Table 16: Scramble with tile 32. Results for the metrics for selected noise levels. Shown in square brackets are bootstrapped 95%-confidence intervals. Here *topo* stands for topographic similarity, *conf* for conflict count, *cont* for context independence, *pos* for positional disentanglement and *acc* for accuracy.

# E  Average topographical similarity for random languages

In this section, we compute the average performance of the topographic similarity metric for a random language when a message is of length $K = 2$. For simplicity, we assume that the feature space and the alphabet space are the same, and equal $\mathcal{F} = \{1, \ldots, n\}^2$. Then the topographic similarity for language $\ell \colon \mathcal{F} \to \mathcal{F}$ is defined as

$$\text{topo}(\ell) = corr(R(\rho(F_0, F_1)), R(\rho(\ell(F_0), \ell(F_1)))),$$

where $F_0, F_1$ are uniform random variables on $\mathcal{F}$ and $R$ is the rank function. The random variable $\rho(F_0, F_1)$ takes values $\{0, 1, 2\}$ with probabilities $p_0, p_1, p_2$. Since $\rho(F_0, F_1)$ is discrete, there are different conventions for defining function $R$. Typical choices for ranks are: (i) "min-ranks" $R(0) = 0, R(1) = p_0, R(2) = p_0 + p_1$; (ii) "max-ranks" $R(0) = p_0, R(1) = p_0 + p_1, R(2) = 1$; (iii) "average-ranks" $R(0) = p_0/2, R(1) = (p_0 + p_1)/2, R(2) = (p_0 + p_1 + 1)/2$. Lemma 1 gives the formula for $\mathbb{E}_{\ell \sim U}[\text{topo}(\ell)]$.

**Lemma 1.** *Let $R$ be a rank function. Then*

$$\mathbb{E}_{\ell \sim U}\left[topo(\ell)\right] = \frac{p_0 R(0)^2 + \frac{2p_1}{n+1} R(1)^2 + \frac{p_2(n-1)}{n+1} R(2)^2 + \frac{4p_2}{n+1} R(1)R(2) - (\sum_{i=0}^{2} p_i R(i))^2}{\sum_{i=0}^{2} p_i R(i)^2 - (\sum_{i=0}^{2} p_i R(i))^2},$$

*where $U$ is a uniform distribution among all bijective $\ell \colon \mathcal{F} \to \mathcal{F}$, $n_0 = n^2$, $n_1 = 2n^2(n-1)$, $n_2 = n^2(n-1)^2$, and $p_i = n_i/n^4$.*

*Proof.* Notice that $\rho(F_0, F_1)$ and $\rho(\ell(F_0), \ell(F_1))$ have the same distribution described by $p_i$. Define $\alpha = R(\rho(F_0, F_1))$ and $\alpha_\ell = R(\rho(\ell(F_0), \ell(F_1)))$. Consequently,

$$\mathbb{E}_{\ell \sim U}\left[\text{topo}(\ell)\right] = \frac{\mathbb{E}_{\ell \sim U}\mathbb{E}[\alpha \alpha_\ell] - (\mathbb{E}[\alpha])^2}{Var(\alpha)}.$$

Since, $\mathbb{E}[\alpha] = \sum_{i=0}^{2} p_i R(i)$ and $Var(\alpha) = \sum_{i=0}^{2} p_i R(i)^2 - (\mathbb{E}[\alpha])^2$, it remains to compute $\mathbb{E}_{\ell \sim U}\mathbb{E}[\alpha \alpha_\ell]$:

$$\mathbb{E}_{\ell \sim U}\mathbb{E}[\alpha \alpha_\ell] = \mathbb{E}\mathbb{E}_{\ell \sim U}[\alpha \alpha_\ell] = \frac{1}{n^4} \sum_{f_0, f_1 \in \mathcal{F}} R(\rho(f_0, f_1))\mathbb{E}_{\ell \sim U}[R(\rho(\ell(f_0), \ell(f_1)))]$$

$$= \frac{1}{n^4} \sum_{i=0}^{2} \sum_{f_0, f_1, \rho(f_0, f_1)=i} R(i)\mathbb{E}_{\ell \sim U}[R(\rho(\ell(f_0), \ell(f_1)))]$$

$$= p_0 R(0)^2 + \frac{1}{n^4} \sum_{i=1}^{2} \sum_{f_0, f_1, \rho(f_0, f_1)=i} R(i)\mathbb{E}_{\ell \sim U}[R(\rho(\ell(f_0), \ell(f_1)))]$$

$$= p_0 R(0)^2 + \frac{1}{n^4} \sum_{i=1}^{2} \sum_{f_0, f_1, \rho(f_0, f_1)=i} R(i)\left(R(1)\frac{2}{n+1} + R(2)\frac{n-1}{n+1}\right)$$

$$= p_0 R(0)^2 + (R(1)p_1 + R(2)p_2)\left(R(1)\frac{2}{n+1} + R(2)\frac{n-1}{n+1}\right).$$

The second to last equality follows from the fact that there $n_i(n^2 - 2)!$ bijections $\ell$ that map $f_0 \neq f_1$ to $\ell(f_0), \ell(f_1)$ such that $\rho(\ell(f_0), \ell(f_1)) = i$, for $i = 1, 2$. $\square$

# F  Optimality of compositional communication

When features are explicitly stated, we can write $\mathcal{F} = \prod_{i=1}^{K} \mathcal{F}_i$, where $\mathcal{F}_i$ is the space of values of $i$-th feature and $K$ is the number of features and a message length. We assume that $|\mathcal{F}_i| = |\mathcal{A}_s|$ and that $|\mathcal{A}_s| \geq 2$. We will assume that a language is a mapping $\ell : \mathcal{F} \to \mathcal{A}_s^K$. We said that the language is compositional if a change in one feature only impacts a corresponding index of the message. Formally, we say that $\ell$ is compositional if and only if for every $k = 0, \ldots, K$, and $f_0, f_1 \in \mathcal{F}$, $\rho(f_0, f_1) = k \iff \rho(\ell(f_0), \ell(f_1)) = k$. Here $\rho$ stands for the Hamming distance.

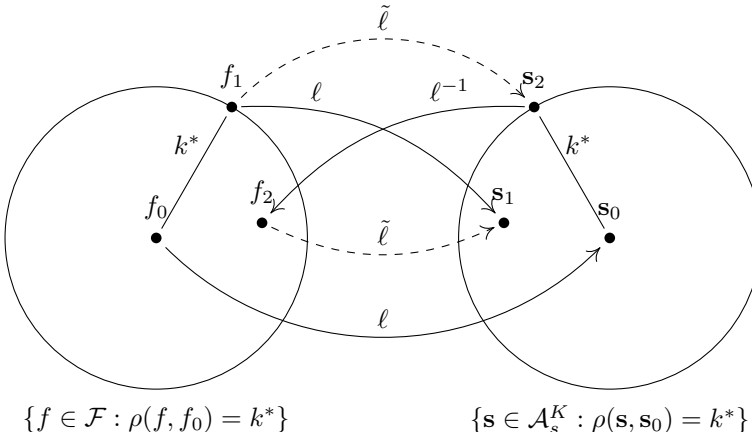

$$\{f \in \mathcal{F} : \rho(f, f_0) = k^*\} \qquad \{\mathbf{s} \in \mathcal{A}_s^K : \rho(\mathbf{s}, \mathbf{s}_0) = k^*\}$$

Figure 31: Illustration of the construction given in the proof of Theorem 3. Two circles represent the ball of radius $k^*$ given by $\rho$ (in $\mathcal{F}$ and $\mathcal{A}_s^K$, respectively). The solid arrow represents the language $\ell$, and the dashed arrows show the swap that is performed when defining $\tilde{\ell}$. $\tilde{\ell}$ improves upon $\ell$ by reducing the number of compositionality violations.

Recall that the corrupted message corresponding to $\mathbf{f} \in \mathcal{F}$ is denoted by $\mathbf{f}' := \ell^{-1}(\ell(\mathbf{f})')$, and the two loss function are defined as:

$$J_1(\ell, \mathbf{f}) = F(e_1, \ldots, e_K), \quad e_j := \mathbb{P}(\mathbf{f}'_j \neq \mathbf{f}_j),$$
$$J_2(\ell, \mathbf{f}) = \mathbb{E}[H(\rho(f', f))],$$

where $F$ is a non-negative function such that $F(\varepsilon, \ldots, \varepsilon) = 0$, and $H$ is a non-negative, increasing function.

The noisy channel transforms a message $\mathbf{s} \in \mathcal{A}_s^K$, into a corrupted message $\mathbf{s}' \in \mathcal{A}_s^K$, by replacing each symbol with a different symbol, independently and with probability $\varepsilon \in (0, 1)$. More formally, the conditional distribution of $\mathbf{s}'$, given $\mathbf{s}$, is expressed by the following formula:

$$\mathbb{P}(\mathbf{s}' = \hat{\mathbf{s}}|\mathbf{s}) = (1 - \varepsilon)^{K - \rho(\hat{\mathbf{s}}, \mathbf{s})} \varepsilon^{\rho(\hat{\mathbf{s}}, \mathbf{s})} \left( \frac{1}{|\mathcal{A}_s| - 1} \right)^{\rho(\hat{\mathbf{s}}, \mathbf{s})}, \qquad \text{for any } \hat{\mathbf{s}} \in \mathcal{A}_s^K. \qquad (5)$$

Indeed, there has to be $\rho(\hat{\mathbf{s}}, \mathbf{s})$ noise flips (hence the first two terms on the right-hand side equation 5), and each flip changes one coordinate of $\mathbf{s}$ to the corresponding coordinate of $\hat{\mathbf{s}}$ with probability $1/(|\mathcal{A}_s| - 1)$.

The following result is a more detailed version of Theorem 2.

**Theorem 3.** *Assume $\mathcal{F} = \mathcal{X}$, and $\varepsilon < (|\mathcal{A}_s| - 1)/|\mathcal{A}_s|$. A compositional language minimizes $J_1$ and $J_2$ over all one-to-one languages. Furthermore, for arbitrary $f \in \mathcal{F}$, $\min J_2(f, \ell) = \mathbb{E}[H(B_\varepsilon)]$, where $B_\varepsilon$ is a Binomial distribution with success probability $\varepsilon$. Moreover, $\ell$ is optimal for $J_2$ if and only if $\ell$ is compositional.*

Notice that, since the assertion holds for arbitrary $f \in \mathcal{F}$, the language $\ell$ is compositional if and only if it is optimal for $\mathbb{E}_{f \sim \nu}[J_2(f, \ell)]$, where $\nu \in \mathcal{P}(\mathcal{F})$ is any distribution such that $supp(\nu) = \mathcal{F}$.

*Proof.* We start by proving the claim for $J_2$. Fix $f_0 \in \mathcal{F}$ and denote $\mathbf{s}_0 = \ell(f_0)$. Suppose that $\ell$ is not compositional. Then, there exists $k > 0, f_1 \in \mathcal{F}, \mathbf{s}_2 \in \mathcal{A}_s^K$ such that $\rho(f_0, f_1) = k, \rho(\mathbf{s}_0, \mathbf{s}_2) = k$, $\rho(f_0, \ell^{-1}(\mathbf{s}_2)) \neq k$, and $\rho(\mathbf{s}_0, \ell(f_1)) \neq k$. Let $k^*$ be the biggest among mentioned $k$'s and denote $\mathbf{s}_1 = \ell(f_1), f_2 = \ell^{-1}(\mathbf{s}_2)$. By the definition of $k^*$, we have

$$\rho(\mathbf{s}_0, \mathbf{s}_1) < k^*, \qquad \rho(f_0, f_2) < k^*.$$

Since $\varepsilon < (|\mathcal{A}_s| - 1)/|A_s|$, the probability $\mathbb{P}(\mathbf{s}' = \hat{\mathbf{s}}|\mathbf{s})$ is a decreasing function of $\rho(\hat{\mathbf{s}}, \mathbf{s})$ (see equation equation 5). It follows that

$$\mathbb{P}(\mathbf{s}' = \mathbf{s}_1|\mathbf{s}_0) > \mathbb{P}(\mathbf{s}' = \mathbf{s}_2|\mathbf{s}_0).$$

We construct a new language $\tilde{\ell}$ (see Figure 31)

$$\tilde{\ell}(f) = \begin{cases} \mathbf{s}_1 & f = f_2, \\ \mathbf{s}_2 & f = f_1, \\ \ell(f) & \text{otherwise.} \end{cases}$$

As a result,

$$\rho\left(f_0, \tilde{\ell}^{-1}(\mathbf{s}_1)\right) - \rho\left(f_0, \ell^{-1}(\mathbf{s}_1)\right) = \rho\left(f_0, f_2\right) - \rho\left(f_0, f_1\right) < 0,$$

$$\rho\left(f_0, \tilde{\ell}^{-1}(\mathbf{s}_2)\right) - \rho\left(f_0, \ell^{-1}(\mathbf{s}_2)\right) = \rho\left(f_0, f_1\right) - \rho\left(f_0, f_2\right) > 0.$$

Putting things together and using the fact that $H$ is increasing, we get

$$J_2(f_0, \tilde{\ell}) - J_2(f_0, \ell) = \mathbb{E}\left[H\left(\rho(\tilde{\ell}^{-1}(\tilde{\ell}(f_0)'), f_0)\right)\right] - \mathbb{E}\left[H\left(\rho(\ell^{-1}(\ell(f_0)'), f_0)\right)\right]$$

$$= \mathbb{P}(\mathbf{s}' = \mathbf{s}_1|\mathbf{s}_0))\left\{H\left(\rho\left(f_0, \tilde{\ell}^{-1}(\mathbf{s}_1)\right)\right) - H\left(\rho\left(f_0, \ell^{-1}(\mathbf{s}_1)\right)\right)\right\}$$

$$+ \mathbb{P}(\mathbf{s}' = \mathbf{s}_2|\mathbf{s}_0)\left\{H\left(\rho\left(f_0, \tilde{\ell}^{-1}(\mathbf{s}_2)\right)\right) - H\left(\rho\left(f_0, \ell^{-1}(\mathbf{s}_2)\right)\right)\right\}$$

$$= \{H\left(\rho(f_0, f_1)\right) - H\left(\rho(f_0, f_2)\right)\}\left(\mathbb{P}(\mathbf{s}' = \mathbf{s}_2|\mathbf{s}_0) - \mathbb{P}(\mathbf{s}' = \mathbf{s}_1|\mathbf{s}_0)\right) < 0.$$

Consequently, for any non-compositional $\ell$ we can strictly decrease its loss value $J_2(f_0, \ell)$. Since the loss is nonnegative and there is a finite number of languages, optimal $\ell$ has to be compositional. For compositional language $\ell$,

$$J_2(f_0, \ell) = \mathbb{E}\left[H(\rho(\mathbf{s}_0', \mathbf{s}_0))\right] = \sum_{\hat{\mathbf{s}} \in \mathcal{A}_s^K} H(\rho(\hat{\mathbf{s}}, \mathbf{s}_0))\mathbb{P}(\mathbf{s}_0' = \hat{\mathbf{s}}|\mathbf{s}_0)$$

$$= \sum_{k=0}^{K} \sum_{\substack{\hat{\mathbf{s}} \in \mathcal{A}_s^K \\ \rho(\hat{\mathbf{s}}, \mathbf{s}_0) = k}} H(k)\mathbb{P}\left(\mathbf{s}' = \hat{\mathbf{s}}|\mathbf{s}_0\right) = \sum_{k=0}^{K} H(k)\binom{K}{k}\varepsilon^k(1-\varepsilon)^{K-k} = \mathbb{E}[H(B_\varepsilon)], \tag{6}$$

where $B_\varepsilon$ is a binomial random variable with success probability $\varepsilon$. This ends the proof for $J_2$.

To prove the assertion for $J_1$ it is enough to notice that a compositional language satisfies $\mathbb{P}(\mathbf{f}_j' \neq \mathbf{f}_j) = \varepsilon$. This follows by equation 6 and the fact that, by symmetry, $\mathbb{P}(\mathbf{f}_j' \neq \mathbf{f}_j)$ does not depend on $j$ (for compositional $\ell$). $\qquad\square$

As a corollary, we see that if $H(x) = x/K$, $J_2(f_0, \ell) = \varepsilon$ for any compositional language $\ell$.

Notice that Theorem 1 follows from the fact that any permutation $\pi : \mathcal{F} \to \mathcal{F}$ is a bijection.