# OpenReview forum: "Catalytic Role Of Noise And Necessity Of Inductive Biases In The Emergence Of Compositional Communication"
_NeurIPS.cc/2021/Conference — NeurIPS 2021 Poster_

### Official Review · Reviewer_PkWg · 2021-07-11

**Rating:** 6
**Confidence:** 3

**Summary:**

The community has been looking at compositional generalisation more recently, and while Fodor and Pyshylynn is an oft cited work, there is a lack of a good definition of compositionality that researchers can agree on.

This paper suggests that a noisy channel with an encoder decoder setup is crucial for the emergence of compositionality. The authors propose a simple definition of compositionality in a non-hierarchical case, and discuss how 1) if no assumptions or prior information about the data is incorporated into the training process, the resulting methods will be uncompositional, 2) if this inductive bias is injected through the use of a noisy channel, there are losses (namely J_1, J_2) that can be used to encourage the rise of a compositional encoding.

The basic setup involves communication through a noisy channel. They then discuss how, if the encoding is compositional, there are particular losses (J_1, and J_2) that will be optimal. More importantly, when J_2 is optimal, the encoding is compositional. In their experiments, they use J_1 primarily, and demonstrate compositional emergence (according to some metrics) under some circumstances.


**Limitations And Societal Impact:**


As mentioned in their conclusion, their work is largely theoretical, and does not directly lead to negative applications. However, improving our understanding of compositionality may lead to much better natural language understanding systems. As with all improvements in technology, these are fraught with positive and negative uses.


**Main Review:**

Originality: I am not aware of other work attempting to give a definition for compositionality, even in this limited form.

Quality:
 I find this definition of compositionality potentially useful, perhaps in future it can be extended to hierarchical cases, and therefore to domains like language. For now, there seems to be some similarity to the notion of equivariance, which can potentially be looked at for extending the current work.

1. The losses J_1 and J_2 are claimed to be optimal when the language is compositional. How J_2 works is much clearer to me. J_1 (as I understand it) requires the probability of P(f’!=f’) , which would require integration over the latent state and the output probabilities at the decoder. I’m not sure if this is being accounted for in the appendix, but this is perhaps due to a lack of understanding on my part.

2. Theorem 2: Am I right to say that minimising J_1 does not guarantee compositionality? The implication in the other direction only seems to apply to J_2.

3. Experiments: Given the nature of the theory here, it would be nearly impossible to test it on a “real” task. Still, I need to ask, is there a way to train a simple model on a simple task with J_1?

4. Experiments: are there other benchmarks on compositionality that can be compared with? I understand that the authors are demonstrating how varying particularly the noise parameters affect compositionality, but it would be useful to see how it stacks up against other methods .


Clarity: The paper is clearly written, but certain aspects of the encoder decoder and its details are hard to find. In fairness, some of these details are in the appendix, and it is hard to fit all details into the page budget.


**Time Spent Reviewing:**

5

---

> ### Author Response · Authors · 2021-08-10
> **Answer to Reviewer PkWg**
>
> We kindly thank the Reviewer for the comments.
>
> Concerning the $J_1$ loss, after reading this review and the Reviewer’s HGBu comments, we have decided to move the discussion of this loss from the main body of the paper to the appendix. $J_1$ can be computed using samples (i.e. via Monte-Carlo), which is a common way in deep learning to compute loss functions in practice. In the paper, we also provided an example of $J_1$ (line 125), which could be used to train a model. For compositional language, $e_j$ is equal to epsilon (since the factor j is changed only if the message on position $j$ was corrupted, which happens with probability epsilon) and by the definition of $J_1$, such a language has to be optimal.
>
> As to Theorem 2, the Reviewer is correct, it states a sufficient condition for $J_1$. It could happen that among minimizers of $J_1$ there are non-compositional languages (e.g. for $F$ mapping everything to 0, every 1-1 language is optimal).
> We are not aware of a dedicated benchmark for emergent communication. The datasets are spread across papers on the topic (see e.g. Korbak et al. [2019], Li and Bowling [2019], Brighton and Kirby [2006],  Bogin et al. [2018]). Our work deals with biases and studies their influence. The direct comparison with other methods would require explicit naming of all their biases, creating a common model and codebase. This is an interesting and worthwhile endeavor, but outside of the scope of this work.

---

> > ### Comment · Reviewer_PkWg · 2021-08-10
> > **Thanks for the clarification.**
> >
> > I would suggest, alternatively, modifying Theorem 2 so that it states both losses separately. "Conversely" in that statements sets up an expectation that makes it slightly more confusing when I get to the end of the sentence. This may be my individual failure in understanding of course.

---

> > > ### Author Response · Authors · 2021-08-18
> > > **Thank you**
> > >
> > > Thank you for your suggestion. This might be a good option. Perhaps, if, additionally, accompanied by our above explanation, the theorem will be more clear.

---

### Official Review · Reviewer_HGBu · 2021-07-16

**Rating:** 7
**Confidence:** 4

**Summary:**

The paper presents theoretical and empirical results showing that a specific type of utterance noise in a sender-receiver emergent communications architecture leads to compositional utterances, under certain constraints, and that the resulting loss function has all local minimum as global minimum (they don't state explicitly that all local minimum are global minimum, but there results strongly implies to me that they are). The following constraints need to be met in order to satisfy the theoretical proof:
- message length = number of attributes
- the input to the sender network should be a bag of attributes, not a non-symbolic, e.g. pixel, input (note that the paper calls attributes 'factors')
- the vocab size of the utterance should exactly match the number of attribute values
- the utterances are discretized using Gumbel, so we can backprop through them, and not using a discrete sampling, which would require REINFORCE
- a specific type of noise is added to the utterances, which randomly flips/corrupts each specific token with an identical probability $\epsilon$
- the noise should be more than 0, and less then (vocab size - 1) / vocab size

The paper presents theoretical results basd on Locatello et al that show that without sufficiently strong inductive biases, that one cannot the utterances to be disentangled.

The paper presents comprehensive empirical results for scenarios which both somewhat align with the theoretical constraints above, and also that deviate strongly from these constraints. For example, all the experiments use non-symbolic inputs, which is therefore not a bag of attributes input. Most of the experiments use a message length that does match the number of attributes, but some experiments are done using longer utterances, and show similar results to the shorter messages.

The paper evaluates the results using 4 existing compositional metrics: topographic similarity, conflict count, context independence, and positional disentanglement.


**Ethical Concerns:**

None noticed

**Limitations And Societal Impact:**

The paper has a 'limitations of the method' section, which is fairly comprehensive, and addresses most/all of the limitations of their theoretical work. The paper presents experiments which show empirically that violating many of these limitations does not cause the empirical results to deviate significantly from the conclusions of the theoretical results.

Overall, I think the paper does a good job of stating clearly the limitations of the work, (albeit, this is left until the section right at the end of the paper).

There are no obvious negative societal implications of the work, since this line of research is very theoretical, and does not involve classes of people, or similar. Any natural language utterances associated with the experiments are limited to very simple concepts such as `red square'.

**Main Review:**

# Main review

The paper provides a theoretical justification showing that, under fairly limitating constraints, noise introduced into the utterances of a sender-receiver emergent communications architecture leads to pure compositional utterances, and that all local minimums of the loss function are global minimum (they don't state that all local minimum are global minimum, but their theoretical results strongly imply this, I think). The constraints around this theoretical result are fairly strict. The message length is tiny. The inputs to the sender must be symbolic bags of attributes. A specific loss function must be used on the receiver, which receiver must predict the bag of attributes that was originally provided to the sender. These are I feel fairly strict constraints, that are not matched by many recent emergent communications works. Nevertheless, I really like that there is any theory at all. I feel that emergent communications could benefit from some theory, rather than just running some experiments, eye-balling the utterances, and saying 'look! compositional!'. I do think that the theoretical results of this paper are a useful baby-step towards more rigorous theoretical results for emergent communications as a whole.

The explanations given in section 3 are I feel fairly hard to understand. I would prefer that section 3 would be written to be much more intuitive and easier to understand. But I will probably let that slide in my own rating of this work, and approve this work for publication anyway.

The introduction and related works sections are I feel pretty solid and complete, and were easy for me to follow.

The experimental results section was I feel extremely comprehensive and nice. I really like the set of experiments and results provided. In addition, the experimental work probed what happens when one violates many of the constraints of the theoretical results, and showed that the empirical results aligned well with the theoretical results in any case under such circumstances. I suppose the main deviation from the theoretical results (which wasnt highlighted as such, and perhaps could/should be), is that the theoretical results hold for any noise $\epsilon$ that is greater than zero, and smaller than (vocab size - 1) / vocab size, which is a pretty broad range of values, extending to most of the range of $(0, 1)$. However, the empirical results using non-symbolic, pixel, inputs showed that the noise must be relatively small. This discrepancy between theory and practice was neither I feel pointed out clearly, nor was any explanation provided for why it is so. Still, I do feel that the provision of theoretical results, however tightly constrained, is an important step forward; and the empirical results did align fairly well with the theoretical results, even when the constraints of the theoretical results were violated somewhat.

Overall, whilst the explanations in section 3 are I feel hard to understand, and the constraints on the theoretical results of the work are quite strict, I feel that this work provides a nice baby step towards more rigorous theoretical understanding of emergent communications; and I feel the empirical work in the experiments section is very nice. At this point in the review lifecycle, my intent is to approve it for publication.

# Specific points I'd like to see addressed

- section 3 target audience
    - I feel that section 3 was written with a target audience of e.g. the person's advisor. Thus, any description of the setup of the task has been entirely skipped, and only the briefest description of the theorems is provided. I feel that a target audience that would make the work easier to understand for me would be, well me, but more broadly, perhaps a good target audience to keep in mind could be the author themselves, imagining that they only know eg question-answering domain, or machine translation domain, and knows nothing about emergent communications domain
- cut $J_1$ loss
    - No justification is provided for the inclusion of the $J_1$ loss, neither intuitive, nor theoretical (appendix F just states that a compositional language would reduce the loss to zero, but does not prove that all local minima are global minima, for example). Nor is any explanation provided anywhere as to how $J_1$ and $J_2$ are combined (i.e. some sort of $\lambda$ trade-off presumably). The explanations of $J_1$ loss are complicated, hard to understand, and not backed by any intuition insight/jusification. They take up lots of space that could be used to present the task etc instead. I feel that $J_1$ loss should be cut entirely. (Or else much more explanation should be provided to justify why it is needed)
- constraint on $\epsilon$ in Theorem 2
    - the constraint on $\epsilon$ in theorem 2, as stated in the main paper body, is misleading I feel. They state that a constraint is that "$\epsilon$ is small enough". This is technically true, but 'small enough' means 'smaller than (vocabsize - 1) / vocabsize', which is most of the range $(0, 1)$. It does not need e.g. $\epsilon \sim 0$, which I felt was implied here. I feel that the actual constraint on $\epsilon$ should be concretely given. I feel in hindsight that perhaps it is phrased as it is, because this makes it sound more similar to the empirical results reported later on, which show that $\epsilon$, when violating $\mathcal{X} = \mathcal{F}$, does need to be relatively small. However, I think it is important to highlight this difference in the constraint on $\epsilon$ for the theoretical results versus the optimal range of $\epsilon$ in the empirical results. This difference shows potential opportunites for future work perhaps

Any of these specific points could be cited by reviewers as sufficient and complete reasons to reject this work, for now, I feel, depending on what other reviewers feel about this work. I do feel that I'd strongly prefer that all three of these points are addressed prior to publication.

# Notes whilst reading

The following are notes I made whilst reading the paper. They are not filtered, or edited, in any way, but simply a stream of consciousness whilst I was writing. Make of them what you will :)

## after abstract:

- abstract looks interesting
- I think I've seen similar results before, possibly, something like "Learning to Communicate with Deep Multi-Agent Reinforcement Learning", Foerster et al, 2016 (but no, thats not quite teh same actually)

## 1. Introduction

- pretty comprehensive range of references
- conflict count Kucinski et al 2020 is new to me.

by the end of the second paragraph of the introduction, I feel the authors are very strong, and know what they are talking about.

[after checking kucinski et al reference] oh I have seen this paper. I just dont remember what is in kucinski et al paper. Oh, it is also about noisy channel, and is probably one of the papers I remember seeing about noisy channel, after reading the abstract.

Going back to the second paragraph of the paper under review, I like that it cites Locatello et al. This is I feel a key paper for anything involving disentanglement, which trying to emerge compositionality highly resembles, I feel.

"Such a result can be perceived as a discrete analog of Locatello et al. [2019], applicable in the communication context.". Hmmm, I worked out the maths for that once, a couple of years ago, but never did anything with it. /me kicks myself :P

"We then prove that adding an inductive bias in the loss function coupled with communication over a noisy channel leads to the spontaneous emergence of compositionality. This shows the catalytic role of noise in this process." Ok, so the authors are probably going further than the tiny baby step I took with discrete Locatello et al proof.

## 2. Related work

" Such an approach inevitably introduces noise into the learning process. This naturally leads to a question of whether the noise itself may be a sufficient mechanism of compositionality". Agreed. I have pondered the same thing, but again, done nothing useful with it :P

Nowak and Krakauer 1999 is new to me. I need to read that.

Good observation about noise allowing backpropagation through a discrete latent. An interesting way of looking at this, I feel.

The Foerster et al 2016 reference was cited too, which was the only reference I remembered after reading the abstract earlier. So, good.

By the end of Related work section, I'm pretty convinced this is going to be a very solid paper.

## 3. Noisy channel method

### 3.1 Language and compositionality

I like the use of the word 'factor' instead of 'attribute'. Although it's not clear to me after reading the first three sentences of section 3.1 what is the definition of a factor. Is it a single attribute, e.g. 'red', or a set of attributes for a single object, e.g. 'red square'? I find myself reading the sentence over and over to figure out the answer to this. You might consider helping me to understand which of these choices is the definition of a factor, in the paper text, perhaps. Ok, so, the fourth sentence implies strongly that a factor is a set of attributes for a single object, eg 'red square'. However, this doesn't align perfectly with the sentence 'the factor could represent shape, ..., and color'. Perhaps the sentence could be reworded slightly to 'the factor could represent a combination of shape, ..., and color'?

I like the definition of $\mathcal{X}$ and $\mathcal{F}$, and the description of the two-stage sampling process. Nice.


I like the definition of a language as mapping to $\mathcal{A}_s^*$. That's pretty compact and descriptive, I feel.

"We will say that l is compositional with respect to a given factor if a change in i-th factor only impacts
85 a corresponding j-th index of the message"

I'm a bit dubious about this definition of compositoinality. Unless you're using two letter utterances, I feel there's going to be more than a single index that is going to be impacted by changing a factor. (In practice, I'd expect most of the characters to be impacted to be honest; but assuming perfect compositionality, if we have 6 letter.... hmmm ... here factor is being defined as a single attribute I think, eg 'red'. Otherwise it doesn't make sense to talk about the effect of the $i$-th factor on the $j$-th index of the message. I think you might want to think carefully about what definition of factor you are using. I'm not sure you are being consistent :)

At this point, in the last two paragraphs, I feel less in the hands of an expert, and am ready to start to revert my earlier impressions on the paper.  I'm wondering if the paper was written by >1 person, and section 3 was written by a different person, perhaps?

By the way, what does the $_s$ in $\mathcal{A}_s$ mean? It has not been stated yet. I feel that either this should be stated, or the $_s$ should be dropped for now, for simplicity?

By the end of the second paragraph of 3.1, I mean, yes, such a language would be compositional, but using only two letters to represent two attributes (I'm using 'attributes' instead of 'factors', since attribute is widely used in the literature, and the definition is widely understood; whereas it's not clear to me which definition of 'factor' the authors intend here?)

3rd paragraph
nit: 'and denoted' => 'and denote'

So, I'm unsure why permuting $\mathcal{F}$ would affect the language. $\mathcal{F}$ is simply a set. It has no order. It was stated in the first sentence of 3.1 that $\mathcal{F}$ is a set. But even if it had order (which it does not, being a set), the language would not change simply because we had swapped the order of our ordered set, since the language is a mapping from sets of attributes/factors to utterances, and as long as $\mathcal{F}$ contains the same set of sets of attributes/factors, which it does, since we merely permuted it, then the language should be unchanged, I feel?

possibly, what is intended to be meant is that if one samples some elements $\{f_1, f_2, \dots, f_n}$ from $\mathcal{F}$, and permute those? I think that paragraph three needs to be carefully rethought/reworded. I don't feel it makes sense currently. At the very least, I feel it is hard to understand (e.g. perhaps I am misunderstanding what it is saying?).

### 3.2 Inductive biases and compositionality

I couldnt really follow 3.2, since $\mathcal{F}$ is a set, so I'm unclear how we can permute it? The other sentences in 3.2 don't really seem to clearly express their meaning to me either.

I'm a deep believer (if that's the right word for a mathematical proof) in Locatello et al. And I think Locatello et al have stated pretty clearly that you cannot in general disentangle representations, since any axis-aligned distribution can be projected to a uniform sphere, spun arbitrarily, then reprojected back into the original space, which is now entangled. Since the intermediate sphere is uniform, and can be rotated arbitrarily, there is no way of determining how to rotate it to obtain the original axis-aligned distribution. (I dont think I explained this really well). In any case, the point is that if one has a latent vector, in the absence of supervision, under very general conditions, one cannot expect e.g. the first element of the vector to become aligned with e.g. shape, and the second to become aligned with e.g. color. The vector will in general be entangled, so that all attribute are "smeared" across the entire vector. We would expect the same thing to hold true also for utterances: each attribute will be "smeared" across the entire utterance. How this relates to what is being stated in 3.2 is unclear to me. So, I feel that 3.2 needs to be reworded somewhat.

In addition, I'm not sure that 3.2 shows understanding of what Locatello et al states. I will wander off to find what they state:

... well they state "We theoretically prove that (perhaps unsurprisingly) the unsupervised learning of disentangled representations is fundamentally impossible without inductive biases both on the considered learning approaches and the data sets" (which aligns with what the paper under review states)

But I think they state something more strongly than that. Let's see...

well Locatello et al state in section 3 "Theorem 1 shows that unsupervised disentanglement learning is fundamentally impossible for arbitrary generative models".

They do however continue "this does not necessarily mean it is an impossible endeavour in practice. After all, real world generative models may have a certain structure that could be exploited through suitably chosen inductive biases. However, Theorem 1 clearly shows that inductive biases are required both for the models (so that we find a specific set of solutions) and for the data sets"

However, I'm not sure I agree with that. If they cannot be disentangled unsupervised, they cannot be: they must have supervision. Perhaps I am misunderstanding Locatello et al myself :P

However, going back to the introduction of Locatello et al, their empirical results align with my own point of view "We do not find any evidence that the considered models can be used to reliably learn disentangled representations in an unsupervised manner as random seeds and hyperparameters seem to matter more than the model choice. Furthermore, good trained models seemingly cannot be identified without access to ground-truth labels even if we are allowed to transfer good hyperparameter values across data sets."

In any case, going back to section 3.2 of the paper under review, the presentation doesn't really discuss any of the above points at all, but simply gives what I feel is a very hand waving sequence of statements, that involves permuting a set, which as far as I know is something that does not possess an ordering?

I googled 'set', doubting my maths. It seems it is possible to have ordered sets, and partially ordered sets, but the general concept of a set does not possess order, e.g. wikipedia says "in mathematics, a set is a collection of distinct elements", https://en.wikipedia.org/wiki/Set_(mathematics)#cite_note-JainAhmad1995-1

Looking back at the earlier 3.1, it says that $\mathcal{F}$ contains a 'combination' of values, and a combination is also unordered, c.f. a permutation. I think that section 3.1 and 3.2 should be at least rewritten to be more clearly understandable.

### 3.3 Compositionality and communication over a noisy channel

"a loss function that ... rewards for (partially) correct guesses". Since we don't have a task yet, I have no idea what the loss function is comparing. Is it comparing attributes? Utterances? Is it a referential task? If it is a referential task, what does it mean to have partially correct guesses. I feel that the presentation is out of order at this point: we need to know what task is being discussed, before we can be told about constraints on the loss function, I feel.

In addition, I mean, I have a pretty good idea of the architecture of a sender, receiver, intermediate utterances, however I feel that in general, a diagram and description should be provided at this point.

paragraph 3 of 3.3: starts to talk about factorized loss functions, but again, without presenting what is the task yet. I'm going to guess the task is a symbolic task, where the sender is provided a set of symbolic attributes, sends a message to the receiver, and the receiver attempts to reconstruct the original set of attribute. But this is basically a guess on my part, and should be stated explicitly in the paper.

Note that using symbolic tasks for experiments on compositionality is I feel a relatively weak approach to studying compositionality, since the inputs to the sender are already factorized/compositional, so I'm not incredibly impressed when the sender turns around and sends something somewhat compositional as a result. In fact, in general, I'm more stunned by how incredibly uncompositional the resulting utterance turns out to be. But that's off-topic: in general, I'm not a fan of compositionality experiments that use symbolic, structured inputs.

Ok, I scrolled ahead a bit, and there are pictures of 3d images in figure 1.

Ok, back to 3.3. I mean, I have no idea what task is being described. This needs to be fleshed out considerably I feel.

So, in equation 1, $l$ has been previously described to be a language, not an utterance, but a language (second paragraph of 3.1). So, is $l$ here being redefined to be an utterance? or does the loss function take as one of its inputs an entire language? Many languages might not even fit in memory on a standard computer, so it surprises me for the loss function to take as an input an entire language.

$\textbf{f}$ has not been defined. We had $f$ earlier (without bold), but not $\mathbf{f}$. I feel it should be defined.

oh, $l$ has been defined to mean 'the corrupted message corresponding to $\mathbf{f}$'. So, $l$ is being redefined. I feel that symbols should ideally be used consistently throughout the paper.

l maybe means the sender network, and $l^{-1}$ is the receiver network? This is not stated.

And maybe the loss is going to be a reconstruction loss between $l^{-1}(l(\mathbf{f})')$ ? I think the notation $'$ being used to mean a function applying noise is very compact, but I'm not sure that using `'` to denote a function is standard? I guess it is readable-ish though, and I'm not a mathematician, so I guess it is probably ok.

there is no intuition being supplied as to why we are taking the KL wrt a uniform distribution. I feel more intuition might be useful.

Ok, so working through this... $\mathbf{f}_j$ means presuably the $j$ attribute. and $e_j$ is the probability that passing the set of attribute through the sender-noise-receiver network has flipped the $j$ attribute. (note: I'm not sure I like the word 'flipped' here, which implies a binary choice. at least to me. Maybe simply 'changed'? I'm not sure). so anyway, $e_j$ is the probability that passing ... so wait... is $e_j$ the probability marginalized over all possible $j$-th attributes? Or is it the probability given some specific input set of attributes? I'm going to guess the latter, though it's kind of odd, since none of the $\mathbf{f}$ are indexed by e.g. an example index number or similar. Nor is it stated that $\mathbf{f}$ represents the attributes of a single example. But I guess it does. So, given some specific example, which is a set of attributes, e.g. a color and a shape, $e_j$ is the probability that after passing through the sender-noise-receiver, that the $j$-th attribute has been corrupted (I prefer 'corrupted' to 'flipped' I feel).

How is this probability being calculated? No indication is provided. It's just assumed that we can calculate this. Are we using Monte-Carlo sampling? Something else? Very unclear. Are we in fact marginalizing across all/many examples, or e.g. a mini-batch?

Then we take the KL divergence of a probability vector obtained by calculating $e_j$ for each of the attributes, w.r.t a uniform distribution.

Ok, so $x$ is not defined. I guess it's a draw from $\mathcal{X}$, so each $x$ is an object. But that's doesnt really make sense. Why are there $K$ objects? So does $x_k$ mean attribute $k$ of object $x$? But isn't $x$ an instance having certain attributes, e.g. a picture of a red ball?  It seems like $\mathbf{f}$ should be representing the attributes, and not $x$? I feel that $x$ needs to be defined. And $x_k$ needs to be defined.

So, let's assume that $x_k$ means attribute $k$. Now, KL is defined only between two probability distributions. Which means that in the expression $F(\epsilon, \dots, \epsilon)$, $\sum_{k=1}^K \epsilon = 1$, therefore $\epsilon = 1/K$. However, earlier $\epsilon$ appeared to be an arbitrary probability, configured as a hyper-parameter, something like dropout. It is not clear to me therefore whether $\epsilon$ is not $1/K$, and therefore $F(\epsilon, \dots, \epsilon)$ is not being calculated over a probability distribution, or else $\epsilon := 1/K$. I feel this needs to be explained more carefully please.

No intuition has been provided into what is intended by the mathematical equations.

For equation (2), I feel it is weird not to use $H$ to mean entropy, especially in this context. I feel that a symbol such as $g$ might be more standard perhaps?

Equation (2) is at least somewhat intuitively underestandable, but note that it does not define what the expectation is over. Is it over the language? A mini-batch? somethign else? I feel it should be defined what the expectation is being taken over.

Theorem 2: why is $\mathcal{F} = \mathcal{X}$? Previously, elements of $\mathcal{X}$ were being sampled on distributions conditioned on draws from $\mathcal{F}$ (start of section 3.1). It's unclear to me why these spaces are even comparable, let alone equal. I feel that more explanation is required here, please.

Ok, so.... stepping back, and thinking about this... the loss function assumes that we have a sender-noise-receiver network, that takes in a set of attributes, and outputs a prediction of the original attributes. the loss function then compares these predictions with the originals by ... calculating the probability of corrupting them. How this probability is being calculated/determined is as far as I can tell unstated/unclear. Let's assume that the authors are marginalizing over a mini-batch or so, i.e. Monte Carlo approximation. So, run a batch of examples through the network, measure empirically how many times each attribute gets flipped, at each position. calculate a loss based on this. and there is a second loss over how far away the prediction is from the ground truth original attributes. Over time, the predicted output will likely resemble the input attributes. But what does this say about the intermediate utterance $l(\mathbf{f})$? A theorem is asserted, but not proved, that states that a compositional language minimizes the losses. That doesnt sound implausible. And it states that any $l$ that minimizes $J_2$ is compositional. I don't have access to theorem 2, so am unable to judge this.

Ok, finally discovered the 'supplementary material' button in the reviewing page :P Downloading and reading...

## Appendix B.1

Ok, so the uttearnces are length 2. That seems ... unambitious... to me, but ok.

scanning forward for the theorems, wow there are a ton of experiments in the appendix :)

## Appendix F:

ok, so $\mathcal{F}$ is the space of all possible sets of attributes. $\mathcal{F}_i$ is the space of values for a single attribute. $K$ is the number of attributes. And the length of the utterances is set to equal the number of attributes, i.e. to $K$. The vocabulary size $|\mathcal{A}_s|$ is set to the number of attributes values $|\mathcal{F}_i|$. We assume at least 2 different letters.

The language maps from the space of all possible sets of attributes to the space of all possible K-length utterances, using vocabulary size $|\mathcal{A}_s|$. I note that the theorem is relatively constrained in only applying to scenarios where the utterance length is exactly identical to the number of attributes, and the vocabulary size is exactly equal to the number of attribute values.

I note that the definition of compositionality being used basically corresponds to that used in topographic similarity (e.g. Lazaridou et al 2018), albeit a stricter version. So, the authors are asserting that the L0 distance between two attribute sets should equal the L0 distance between teh resulting K-length utterances. Which makes sense to me.

$\mathbf{f} \in \mathcal{F}$ is presumably a specific set of attributes, eg 'red ball', drawn from the space of all possible sets of attributes.

eyeballing equation 7, I guess it might make sense. I think it would be good to explain the other two terms, on the left of the right hand side of the equation, just as you have explained the right most term. But they seem plausible-ish to me, eye-balling.

### Theorem 3:

I'm not sure why we are assuming $\mathcal{F} = \mathcal{X}$, but ok.

So, here the expectation is over $\mathbf{f} \sim \nu$, and the explanation makes sense to me.

Ok, far too much maths for me. I'm going to assume the maths is probably correct.

So, the maths assumes:
- $\mathcal{F} = \mathcal{X}$
- vocab size = number of attribute values
- utterance length = number of attributes
- $\epsilon < (vocabsize - 1) / vocabsize$

And the proof says that a loss function which is for example proportional to the L0 distance between the predicted attribute set and the input attribute set will drive the utterances to be perfectly compositional.

I wonder what assumptions it has on the noise? i.e. does it assume noise > 0. checking... well, I guess equation 7 would have a right hand side of 0 if $\epsilon = 0$. So, ok, makes sense.

Let's move back to the main body of the paper...

### section 3.4

ok, so the pipeline is:

pixel input => sender => utterance => noise => noised utterance => receiver => predicted attributes

As stated, I think this should be stated earlier, or at least provided as an example implementation, of a more general architecture.

I guess it makes intuitive sense that the utterances in this case might become compositional. Interesting that one needs noise for that to be the case. Interesting approach to noise. (I recognize that the noise design is not from this work, but I do understand the explanation of the noise better in this work tbh. Good :) )

The explanation of the noise matrix more or less makes sense to me. I do think that an example might be useful potentially.

### 4.3 Compositionality measures

Nit: the caption for figure 1 sort of blends in with the text, and it's hard to figure out where one ends and the next begins. Maybe separate these slightly more?

Figure 2 is interesting.

Nice comprehensive list of compositionality metrics :)

## 5. Experiment results

Figure 3 top row: interesting.

Figure 3 bottom row: very interesting. Interesting how the distributions are basically bimodal, but the mass shifts from the rightmost peak to the lowermost peak, ad add noise.

effect of variable noise is interesting to me

effect of changing the type of noise is interesting to me.

effect of visual priors is interesting

effect of scrambled labels is I feel a pretty nice experiment

network features generalization is interesting I feel

that compositionality is not sufficient to give good generalization is interesting I feel. Note that I'm not sure what is the goal of figure 5. I feel figure 5 could be cut, to make space for more explanations in section 3.

## 6 Limitations of the method

I like the presence of this section. This section seems fairly complete, and I like that.

# Thoughts after reading through the paper

At this point, I feel that:
- I like the theoretical contributions of the paper. Yes, they are for very constrained situations, but I feel this is a great start for more complete theoretical contributions by the authors and others in the future.
- I like the extremely comprehensive set of experiments, given these constraints, and that they use non-symbolic input
- section 3 is I feel extremely challenging to understand, but probably not enough to tip the balance towards rejection I feel. I do feel that if section 3 could be written more clearly, this would certainly have made it much easier for me to understand the paper

At this point, I will look (and not before) at the information provided about previous submissions:

# Thoughts after reading submission history

"The reviewers had concerns about the layout of the paper, the presentation of the results, and the clarity of the description of technical details. To address this, we thoroughly rewritten the paper, reworked the presentation of the experiment, and fixed a number of errors."
- so, the results presentation looks good to me
- the layout of the paper seems reasonable to me
- the clarity of the description of the technical details is, as stated above, still a major concern I feel

"We also run all experiments from scratch on 100 seeds (a 5x more than in the previous submission)"
- ok. I'm not sure that that was a concern of mine. ok.


"At the reviewers’ request, we added additional compositionality metrics: positional disentanglement."
- Ok. Good. I do like having multiple compositional metrics, yes

"Additionally, following the reviewers’ advice, we run experiments with a longer message length."
- technically this is true: there are experiments on a longer message length. And I did like this. The theoretical results of course are still only for the shorter messages, but: baby steps, have to start somewhere

"The reviewers wondered about the validity of Theorem’s 2 assertions when its assumptions are not met. We computationally verified that for small cases, it is indeed true."
- I have the same concern, but I do feel that the empirical results you've added help alleviate this concern, and I'm happy to see some small baby steps toward some actual theory in emergent communications


**Time Spent Reviewing:**

5-6 hours

---

> ### Author Response · Authors · 2021-08-10
> **Answer to Reviewer HGBu**
>
> We kindly thank the Reviewer for multiple valuable comments (the Reviewer’s ‘notes whilst reading’ were very helpful). They will greatly help to improve the publication.
>
>
> We address the Reviewer’s concerns as follows:
> * We will start this section with a new subsection, which clearly presents the task used in theoretical and empirical analysis. The subsection will also contain a discussion of the differences between the setups (such as the relation of $\mathcal X$ and $\mathcal F$, symbolic vs non-symbolic input, noise levels, alphabet size).
>
> * Section 3.1 will be rewritten to clarify points which we describe below:
>
>   * We write ‘factors could be identified with partitions of $\mathcal F$. We assumed the mathematical notion of the equivalence relations (partitions) for the sake of precision (perhaps at the cost of readability). In this view, a factor (or an attribute) is identified with an abstract equivalence class. For example, the notion of 'red' is identified with the set of 'red objects'. This could be perhaps best explained by an example: for color-shape factors, the partition {{red square, red circle}, {green square, green circle}} corresponds to color (‘red’ and ‘green’); while the partition {{red square, green square}, {red circle, green circle}} corresponds to a shape (‘square’ and ‘circle’). Such a formalism allows one to easily construct “non-intuitive” factors.
>
>   * In $A_s$, $s$ stands for ‘symbol’.
>
>   * As to the ‘permuting’ concerns, we agree that we should have been more precise with wording. The statement ‘we permute elements of $\mathcal F$ before generating elements of $\mathcal X$’ refers to a composition of functions $l$ (language) and $\pi$ (permutation), not to a ‘permutation of a set’. To our defense, the composition was used e.g. in Theorem 1.
>
> * Section 3.2:
>
>   * We will improve the clarity of this section and make sure that it doesn’t cause any formal confusion on the reader’s part.
>
>   * We argue that the finding in this section is in line with the findings in Section 3 of Locatello et al. [2019].
>
> * Section 3.3:
>
>   * We have decided to follow the Reviewer’s lead (as well as the Reviewer PkWg), and move the loss $J_1$ to the appendix. Both losses are related to whether the agents (the sender and the receiver) make mistakes. We will use the space in the Appendix to provide a better description of the loss.
>
>    * As to the inputs to the loss function, $l$ is indeed the loss function argument. The reason is that we want to show that compositionality is an optimal property of languages. This is a theoretical part, so we are not limited by memory constraints.
>
>   * Concerning the constraint on $\epsilon$ in Theorem 2, we will do as the Reviewer advises. Our goal here was to suggest that assumption on epsilon is a technical detail (and leave the exact details in the appendix).
>
>   * We will tie any loose ends when it comes to notation.

---

### Official Review · Reviewer_ecJL · 2021-07-17

**Rating:** 7
**Confidence:** 4

**Summary:**

The authors analyze the effect of a _noisy channel_ on an emergent communication protocol.  In particular, they show that languages that minimize losses involving this noise are compositional [in their sense of the term] and then conduct a very thorough set of experiments while varying noise in a standard emergent communication.  They find that compositionality tends to go up for some low levels of noise, but then goes down after that (and accuracy tends to go down consistently with more noise).  There is also a very thorough set of experiments manipulating myriad factors in the setup and finding that the overall patterns just discussed are robust.  The paper is a very interesting theoretical and experimental contribution to the emergent communication literature.  My only minor complaint would be that there are so many experimental results that it's easy for the reader to lose the forest for the trees; more synthesis of the overall trends would be welcome.

**Limitations And Societal Impact:**

Yes, the authors provided a very nice limitations section.

**Main Review:**

In addition to the summary above:

# Minor questions

* Why was a feed-forward instead of recurrent network?

* Fig 4(b): what about accuracy for this setting?  Does it still decline in higher noise values, or does it rise with higher topo even with more noise?


# Missing references

* It would be useful to discuss Steinert-Threlkeld 2019: https://www.journals.uchicago.edu/doi/10.1086/710628.  The notion of compositionality in the present paper is what he refers to as "trivial" compositionality.  This is still a hard and valuable target, but does not go all of the way to capturing the rich forms of composition that exist in natural language.  It would be good to call attention to this aspect of the definition of compositionality in the present paper.

* Pagin and Westerstahl 2010a/b show the following result: on their [relatively standard] definition of compositionality, all one-to-one mappings are compositional.  Is this at all connected to the one-to-one assumption made in the proof of your theorem?  It might be nice to discuss.


# Typographic comments

* page 3, line 98: "Theorem 1...." is a sentence fragment, with no verb.

* page 3, line 128: missing parenthesis after $e_j$

* page 5, line 214: "s^k" should be "s^f"

* page 6, line 233: maybe better to use "main experiment" instead of "master experiment"

**Time Spent Reviewing:**

1.5

---

> ### Author Response · Authors · 2021-08-10
> **Answer to Reviewer ecJL**
>
> We kindly thank the Reviewer for encouraging  comments.
>
> The reasons why we did not end up using recurrent neural networks are two-fold. First, we wanted to use simple models. Such an approach allows to limit the number of biases on the network side, making the setup easier and ‘cleaner’. Second, for the message length considered in the paper (two symbols), using more complex architecture could most likely have minor effects and we expect the results to be similar. However, in general, we could use a stronger model, such as recurrent networks, transformers, or graph-nets. This was highlighted in the ‘Limitations of the method’ section.
>
> As to Fig 4(b) or, more generally, experiments in sections 5.1-5.3, we have their dedicated sections in Appendix, where the detailed results for all the metrics (including accuracy) can be found. For this particular experiment see Appendix D.2. Figure 15 and Table 4. You can see that the accuracy stays at a similar level up to the noise level of 14% and starts to decline afterward.
>
> Concerning Steinert-Threlkeld 2019, we alluded to the notion of non-trivial compositionality when discussing limitations on lines 358-360 but didn’t refer to it explicitly; we will change that. It is important, however, to notice that the subject of non-trivial compositionality is relatively new, there is no definition agreed on by the community, and it is not fully understood theoretically. Additionally, Korbak et al. (2020) found that most compositionality metrics used in emergent communication fail to detect non-trivial compositionality, making it an elusive target. The scope of our paper is different: we investigate the fundamental questions concerning relations of compositionality with biases and the influence of noise.
>
> Thank you for pointing out the Pagin and Westerstahl 2010a/b; we will add it to the related work. It is indeed interesting. For comparison, our result could be interpreted as saying that every one-to-one language is compositional, with respect to *some* factors. However, when the factors have been fixed, only a narrow class of languages are compositional. We reckon such a definition makes the notion of compositionally meaningful (as it is related to the semantic category of factors).
>
> Korbak, T., Zubek, J. & Rączaszek-Leonardi, J. (2020). Measuring non-trivial compositionality in emergent communication. NeurIPS 2020 workshop on Emergent Communication.

---

### Official Review · Reviewer_H5rU · 2021-07-17

**Rating:** 6
**Confidence:** 3

**Summary:**

This paper adds results to the literature on learning compositional emergent communication protocols. Two results are theoretical: that inductive biases are required in order for compositional communication to emerge, and that adding a noisy channel with a specific loss function leads to compositional language emergence. The third result is a small-scale empirical corroboration of the latter theoretical result in a speaker-listener game.

**Limitations And Societal Impact:**

There is a specific section on limitations which is greatly appreciated. Societal impact is briefly mentioned

**Main Review:**

I thought this paper was fairly interesting. It adds to the many other papers that investigate factors that affect the emergence of compositional communication. This paper is unique in that the main contributions are theoretical, rather than empirical -- the paper proves that,

As far as I can tell, the results are technically sound. But it's a bit difficult for me to assess the significance of the paper:

- The first theorem, that inductive biases are needed for emergent communication, is interesting but also not very surprising (having agents policies' represented by neural networks is already a form of inductive bias?).

- The more consequential result in my view is the idea that adding noise to the communication channel promotes compositionality (Theorem 2 + empirical results). I find it surprising that nobody has published this before -- the closest that I can think of is the work by Foerster et al. (2016) who show that adding noise to a continuous communication channel leads to the formation of discrete symbols (which is different than how the work is described in this paper). If this is indeed true, I find the result pretty interesting (it's also simpler than many previously proposed strategies for attaining compositional emergent language).

- The empirical results are done at quite small scale, though this is fairly standard in the emergent communication literature.

- I'm uncertain as to the value of these kinds of theoretical results in the field of emergent communication, but this could just be a function of differing research interests / tastes.

As an aside, I greatly appreciate the section on the limitations of the paper, and see this as a big plus.

Overall, I lean slightly in favor of acceptance, but I'm uncertain in my assessment.

Minor points:
- The citation style seems non-standard for NeurIPS

**Time Spent Reviewing:**

2

---

> ### Author Response · Authors · 2021-08-10
> **Answer to Reviewer H5rU**
>
> We kindly thank the Reviewer for the comments.
>
> We agree with the Reviewer that the first theorem can be viewed perhaps as unsurprising in the light of Locatello, et al. [2019], however as far as we know, we are the first to point this out in the (discrete) communication context. The result itself is general and independent of a particular model class (e.g. neural networks). Furthermore, we aim to increase awareness of this fact, and the importance of explicit recognition of different kinds of biases chosen in the research.
>
> As to the scale of empirical results, the environment we use is quite standard in emergent communication literature  (see e.g. Korbak et al. [2019], Li and Bowling [2019], Brighton and Kirby [2006],  Bogin et al. [2018]). However, we feel that the scale of our experiments is quite extensive: 19 classes of experiments run on 16 noise levels (where applicable) and 100 seeds (with each seed running for around 5h on 5 cores). A detailed description of experiments can be found in Appendix.
> Regarding the value of theoretical results, the field of emergent communication (and cognitive science more broadly) is on the lookout for a theory that could explain what are the necessary and sufficient conditions for a compositional language to emerge. Recently, some building blocks towards such a theory were put forth by Ren et al. (2020, appendices C-D) who propose a mathematical argument for the learning speed advantage of compositional languages, and Andreas (2020, proposition 2) who prove that tree reconstruction error upper bounds topographic similarity. We consider our theoretical contributions as another, in our view important, step in this line of research: we proved necessary (inductive biases) and, under some assumptions, sufficient (noise) conditions for the emergence of compositionality.  We also bring the community’s attention to the role of biases, in the spirit of Locatello et al. [2019].
>
> Andreas, J. (2020). Measuring Compositionality in Representation Learning. ICLR.
>
> Ren, Y., Guo, S., Labeau, M., Cohen, S. B., & Kirby, S. (2020). Compositional languages emerge in a neural iterated learning model. ICLR.

---

> > ### Comment · Reviewer_H5rU · 2021-08-29
> > **Response**
> >
> > Thanks for the response. I agree that the fact that the environments considered in this paper are small-scale should not count against the paper (as it is common in the emergent communication literature), and I also agree that the number of experiments and ablations in this paper is extensive, which is one of the strongest points of the paper.
> >
> > I remain in favor of acceptance.

---

### Decision · Program_Chairs · 2021-09-27

**Decision:**

Accept (Poster)

**Comment:**

The paper studies the effect of a noisy channel in an emergent communication setting. They formally prove that, under fairly restrictive assumptions (type of noise, loss functions used, capacity of the channel, etc.), noise in the communication channel promotes emergence of compositional languages. The paper presents extensive empirical results studying the emergence of compositional languages in conditions that also diverge from the theoretical ones.

--

This paper offers to the literature of emergent communication a first, restrictive, but interesting, formal result on the emergence of compositional languages. All reviewers agree that the paper is interesting and the experiments are extensive. The section on the limitation of the current approach and the restrictive assumptions of the theorems have been found useful and important to inspire future work. I suggest the authors to incorporate the precious reviewers' feedback, and especially: (a) the sheer amount of experiments is hard to process, one possible direction for improvement would be to add a synthesis of the overall trends and most important observations; (b) improve writing and clarity of Section 3, expanding on the relevance and interpretation of the theorem and the task setup.

Overall, I think this paper brings a worthy contribution to the field and can inspire future theoretical work to relax the strict assumptions considered here. Therefore, I recommend this paper for acceptance.